# Electrophysiological dynamics of salience, default mode, and frontoparietal networks during episodic memory formation and recall revealed through multi-experiment iEEG replication

Anup Das[1]*, Vinod Menon[2,3,4]*

[1]Department of Biomedical Engineering, Columbia University, New York, United States; [2]Department of Psychiatry and Behavioral Sciences, Stanford University School of Medicine, Stanford, United States; [3]Department of Neurology and Neurological Sciences, Stanford University School of Medicine, Stanford, United States; [4]Wu Tsai Neurosciences Institute, Stanford University School of Medicine, Stanford, United States

**\*For correspondence:**
ad3772@columbia.edu (AD);
menon@stanford.edu (VM)

**Competing interest:** The authors declare that no competing interests exist.

## eLife Assessment

In this article, the authors present **valuable** findings on the apparent role of a salience network anterior insula node in directing frontoparietal and default mode network activity within a tripartite network during control of memory, drawn from an impressive invasive human neurophysiological dataset. Overall, the authors have presented a **convincing** set of analyses. We also commend the use of a large intracranial EEG dataset to approach this question.

**Abstract** Dynamic interactions between large-scale brain networks underpin human cognitive processes, but their electrophysiological mechanisms remain elusive. The triple network model, encompassing the salience network (SN), default mode network (DMN), and frontoparietal network (FPN), provides a framework for understanding these interactions. We analyzed intracranial electroencephalography (EEG) recordings from 177 participants across four diverse episodic memory experiments, each involving encoding as well as recall phases. Phase transfer entropy analysis revealed consistently higher directed information flow from the anterior insula (AI), a key SN node, to both DMN and FPN nodes. This directed influence was significantly stronger during memory tasks compared to resting state, highlighting the AI's task-specific role in coordinating large-scale network interactions. This pattern persisted across externally driven memory encoding and internally governed free recall. Control analyses using the inferior frontal gyrus (IFG) showed an inverse pattern, with DMN and FPN exerting higher influence on IFG, underscoring the AI's unique role. We observed task-specific suppression of high-gamma power in the posterior cingulate cortex/precuneus node of the DMN during memory encoding, but not recall. Crucially, these results were replicated across all four experiments spanning verbal and spatial memory domains with high Bayes replication factors. Our findings advance understanding of how coordinated neural network interactions support memory processes, highlighting the AI's critical role in orchestrating large-scale brain network dynamics during both memory encoding and retrieval. By elucidating the electrophysiological basis of triple network interactions in episodic memory, our study provides insights into neural

circuit dynamics underlying memory function and offer a framework for investigating network disruptions in memory-related disorders.

## Introduction

Dynamic interactions between large-scale brain networks are thought to underpin human cognitive processes, but the electrophysiological dynamics that underlie these interactions remain elusive. The triple network model, which includes the salience network (SN), default mode network (DMN), and frontoparietal network (FPN), offers a fundamental framework for understanding these complex interactions (*Cai et al., 2021*; *Menon, 2011*; *Menon, 2023*). These networks collaboratively manage tasks that require significant stimulus-driven and stimulus-independent attentional control, highlighting the integrated nature of brain function. Building on *Mesulam, 1990*' theory that all cognitive and memory systems operate within a complex architecture of interconnected brain regions, the triple network model articulates how these networks facilitate demanding cognitive tasks. However, despite the model's broad influence, the specific electrophysiological mechanisms that support these interactions during cognition remain poorly understood.

Episodic memory, the cognitive process of encoding, storing, and retrieving personally experienced events, is essential for a variety of complex cognitive functions and everyday activities (*Dickerson and Eichenbaum, 2010*; *Düzel et al., 2010*; *Moscovitch et al., 2016*; *Ranganath and Ritchey, 2012*; *Rugg and Vilberg, 2013*; *Rutishauser et al., 2021*; *Yonelinas et al., 2019*). Influential theoretical models of human memory posit a key role for control processes in regulating hierarchical processes associated with episodic memory formation (*Andermane et al., 2021*; *Atkinson and Shiffrin, 1968*; *Bastos et al., 2012*; *Kumaran and McClelland, 2012*; *Tulving, 2002*). Crucially, the formation of episodic memories relies on the intricate interplay between external stimulus-driven processes during encoding and internal recall processes during retrieval (*Buckner and DiNicola, 2019*; *Fornito et al., 2012*; *Mesulam, 1990*), making it an ideal cognitive process to investigate the triple network model's broader applicability and its underlying neurophysiological mechanisms. Elucidating these mechanisms is crucial not only for understanding basic brain functions but also for addressing neuropsychological disorders where these mechanisms may be disrupted (*Li et al., 2019*).

Each network in the triple network model plays a unique and critical role in regulating human cognition (*Menon, 2023*). The SN, anchored by the anterior insula (AI), identifies and filters salient stimuli, helping individuals focus on goal-relevant aspects of their environment (*Menon and Uddin, 2010*). In contrast, the DMN is typically engaged during internally focused cognitive processes and is implicated in the retrieval of past events and experiences (*Buckner et al., 2008*; *Fox and Raichle, 2007*; *Fox et al., 2005*; *Greicius et al., 2008*; *Greicius and Menon, 2004*; *Laufs et al., 2003*; *Raichle, 2015*; *Raichle et al., 2001*; *Smallwood et al., 2021*). The FPN is involved in the maintenance and manipulation of information within working memory and exerts top-down attentional control to regulate memory formation (*Badre et al., 2005*; *Badre and Wagner, 2007*; *Helfrich and Knight, 2016*; *Jin et al., 2010*; *Simons and Spiers, 2003*; *Uncapher and Wagner, 2009*; *Wagner et al., 2001*; *Wagner et al., 2005*).

Central to the functionality of this model is the AI, a pivotal node within the SN. Functional brain imaging studies have revealed the SN's critical role in regulating the engagement and disengagement of the DMN and FPN across diverse cognitive tasks (*Bressler and Menon, 2010*; *Cai et al., 2016*; *Cai et al., 2021*; *Chen et al., 2016*; *Kronemer et al., 2022*; *Raichle et al., 2001*; *Seeley et al., 2007*; *Sridharan et al., 2008*). The AI dynamically detects and filters task-relevant information, facilitating rapid and efficient switching between the DMN and FPN in response to shifting task demands (*Menon, 2015a*). However, how this process operates at the neurophysiological level remains unknown, underlining a significant gap in our understanding of directed network dynamics in memory formation.

While the tripartite network has been most extensively studied in the context of cognitive tasks requiring explicit cognitive control, growing evidence suggests its relevance to episodic memory as a domain-general control system. Brain imaging studies in both healthy individuals and clinical populations provide growing evidence for the involvement of the tripartite network in memory processes. In healthy adults, *Sestieri et al., 2014* found that the SN exhibited sustained activity across all phases of both episodic memory search and perceptual tasks. The SN was consistently activated across all task phases, from initiation to response, indicating its broad involvement in memory processes. Importantly,

the SN demonstrated flexible functional connectivity, linking with the DMN during memory search and dorsal attention network during perceptual search. These findings point to the SN's involvement in dynamically coordinating large-scale brain networks during episodic memory processes, supporting its characterization as a versatile, domain-general control network that adapts its connectivity patterns to meet diverse cognitive demands.

Further supporting this view, *Vatansever et al., 2021* demonstrated shared neural processes, centered on the AI, supporting the controlled retrieval of both semantic and episodic memories. They identified a common cluster of cortical activity centered on the AI and adjoining inferior frontal gyrus (IFG) for the retrieval of both weakly associated semantic and weakly encoded episodic memory traces. Moreover, they found that reduced functional interaction between this cluster and the ventro-medial prefrontal cortex, a key node of the DMN, was associated with better performance across both memory types. Higher pre-stimulus activity in the SN was associated with increased activity in temporal regions linked to encoding and reduced activity in regions associated with retrieval and self-referential processing (*Cohen et al., 2020*). This suggests that the SN may regulate memory by enhancing encoding and reducing interference from competing memory processes. Together, these findings not only reinforce the domain-general role of the SN in memory processes but also highlight the importance of investigating the interactions between the tripartite network components during memory tasks.

Clinical studies have also underscored aspects of the tripartite network in memory function. *Le Berre et al., 2017* found that disrupted insula connectivity was associated with unawareness of memory impairments in non-Korsakoff's syndrome alcoholism, highlighting the crucial role of the right insula in memory functioning. Additionally, alcoholics showed weaker connectivity between the right insula and the dorsal anterior cingulate cortex nodes of the SN, and stronger connectivity between the right insula and ventromedial prefrontal cortex, a key node of the DMN. Importantly, alcoholics who failed to desynchronize insula-ventromedial prefrontal cortex activity demonstrated greater overestimation of their memory predictions and poorer recognition performance. Similarly, *Xie et al., 2012* demonstrated that disrupted intrinsic connectivity of insula networks was associated with episodic memory deficits in patients with amnestic mild cognitive impairment. These studies suggest that disrupted insula connectivity may underlie the lack of awareness of memory impairments and highlights the crucial role of the SN in memory functioning.

Despite these advances, the electrophysiological basis and dynamic interactions of these networks during memory formation and retrieval remain poorly understood. Our understanding of dynamic network interactions during human cognition is primarily informed by fMRI studies, which are limited by their temporal resolution. This constraint impedes our understanding of real-time, millisecond-scale neural dynamics and underscores the need to explore network interactions at time scales more pertinent to neural circuit dynamics. However, the difficulties involved in acquiring human electrophysiological data from multiple brain regions have made it challenging to elucidate the precise neural mechanisms underlying the functioning of large-scale networks. These challenges obscure our understanding of the dynamic temporal properties and directed interactions between the AI and other large-scale distributed networks during memory formation.

To address these gaps, we leveraged intracranial electroencephalography (iEEG) data acquired during multiple memory experiments from the University of Pennsylvania Restoring Active Memory (UPENN-RAM) study (*Solomon et al., 2019*). This dataset provides an unprecedented opportunity to probe the electrophysiological dynamics of triple network interactions during both episodic memory encoding and recall, with depth recordings from 177 participants across multiple memory experiments. The UPENN-RAM dataset includes electrodes in the AI, the posterior cingulate cortex (PCC)/precuneus and medial prefrontal cortex (mPFC) nodes of the DMN, and the dorsal posterior parietal cortex (dPPC) and middle frontal gyrus (MFG) nodes of the FPN. By examining four diverse episodic memory tasks spanning verbal and spatial domains, we aimed to elucidate the neurophysiological underpinnings of the AI's dynamic network interactions with the DMN and FPN and assess the consistency of these interactions across tasks and stages of memory formation.

We investigated four episodic memory experiments spanning both verbal and spatial domains. The first experiment was a verbal free recall memory task (VFR) in which participants were presented with a sequence of words during the encoding period and asked to remember them for subsequent verbal recall. The second was a categorized verbal free recall task (CATVFR) in which participants

were presented with a sequence of categorized words during the encoding period and asked to remember them for subsequent verbal recall. The third involved a paired associates learning verbal cued recall task (PALVCR) in which participants were presented with a sequence of word-pairs during the encoding period and asked to remember them for subsequent verbal cued recall. The fourth was a water maze spatial memory task (WMSM) in which participants were shown objects in various locations during the encoding periods and asked to retrieve the location of the objects during a subsequent recall period. This comprehensive approach afforded a rare opportunity in an iEEG setting to examine network interactions between the AI and the DMN and FPN nodes during both encoding and recall phases across multiple memory domains.

A crucial test of the triple network model is whether the AI exerts a strong directed influence on the DMN and FPN. The AI is consistently engaged during attentional tasks, and dynamic causal modeling of fMRI data suggests that it exerts strong causal influences on the DMN and FPN in these contexts (*Cai et al., 2016*; *Cai et al., 2021*; *Chen et al., 2016*; *Sridharan et al., 2008*; *Wen et al., 2013*). However, it remains unknown whether the AI plays a causal role during both memory encoding and recall and whether such influences have a neurophysiological basis. To investigate the directionality of information flow between neural signals in the AI and DMN and FPN, we employed phase transfer entropy (PTE), a robust and powerful measure for characterizing information flow between brain regions based on phase coupling (*Hillebrand et al., 2016*; *Lobier et al., 2014*; *Wang et al., 2017*). Crucially, it captures linear and nonlinear intermittent and nonstationary dynamics in iEEG data (*Hillebrand et al., 2016*; *Lobier et al., 2014*; *Menon et al., 1996*). We hypothesized that the AI would exert higher directed influence on the DMN and FPN than the reverse.

To further enhance our understanding of the dynamic activations within the three networks during episodic memory formation, we determined whether high-gamma band power in the AI, DMN, and FPN nodes depends on the phase of memory formation. Memory encoding, driven primarily by external stimulation, might invoke different neural responses compared to memory recall, which is more internally driven (*Andrews-Hanna, 2012*; *Buckner et al., 2008*). We hypothesized that DMN power would be suppressed during memory encoding as it is primarily driven by external stimuli, whereas an opposite pattern would be observed during memory recall which is more internally driven. Based on the distinct functions of the DMN and FPN—internally oriented cognition and adaptive external response—we expected to observe differential modulations during encoding and recall phases. By testing these hypotheses, we aimed to provide a more detailed understanding of the dynamic role of triple network interactions in episodic memory formation, offering insights into the temporal dynamics and directed interactions within these large-scale cognitive networks.

Our final objective was to investigate the replicability of our findings across multiple episodic memory domains involving both verbal and spatial materials. Reproducing findings across experiments is a significant challenge in neuroscience, particularly in invasive iEEG studies where data sharing and sample sizes have been notable limitations. There have been few previous replicated findings from human iEEG studies across multiple task domains. Quantitatively rigorous measures are needed to address the reproducibility crisis in human iEEG studies. We used Bayesian analysis to quantify the degree of replicability (*Ly et al., 2019*; *Verhagen and Wagenmakers, 2014*). Bayes factors (BFs) are a powerful tool for evaluating evidence for replicability of findings across tasks and for determining the strength of evidence for the null hypothesis (*Verhagen and Wagenmakers, 2014*). Briefly, the replication BF is the ratio of marginal likelihood of the replication data, given the posterior distribution estimated from the original data, and the marginal likelihood for the replication data under the null hypothesis of no effect (*Ly et al., 2019*).

In summary, our study aims to elucidate the neurophysiological basis of the interactions between large-scale cognitive networks by leveraging a unique dataset of iEEG recordings across multiple memory experiments. By examining directed information flow, high-gamma band power modulation, and replicability across verbal and spatial memory domains, we sought to advance our understanding of the neural mechanisms underpinning human episodic memory. Our findings shed light on how the brain effectively integrates information from distinct networks to support memory formation, and cognition more broadly.

# Results

## AI response compared to PCC/precuneus during encoding and recall in the VFR task

We first examined neuronal activity in the AI and the PCC/precuneus and tested whether activity in the PCC/precuneus is suppressed compared to activity in the AI. Previous studies have suggested that power in the high-gamma band (80–160 Hz) is correlated with fMRI BOLD signals (*Hermes et al., 2017*; *Hutchison et al., 2015*; *Lakatos et al., 2019*; *Leopold et al., 2003*; *Mantini et al., 2007*; *Schölvinck et al., 2010*), and is thought to reflect local neuronal activity (*Canolty and Knight, 2010*). Therefore, we compared high-gamma band power (see 'Methods' for details) in the AI and PCC/precuneus electrodes during both encoding and recall and across the four episodic memory tasks. Briefly, in the VFR task, participants were presented with a sequence of words and asked to remember them for subsequent recall ('Methods', *Appendix 1—tables 1, 2 and 6*, *Figures 1a and 2*).

### Encoding
Compared to the AI, high-gamma power in PCC/precuneus was suppressed during almost the entire window 110–1600 ms during memory encoding (ps<0.05, *Figure 3a*).

### Recall
In contrast, suppression of high-gamma power in the PCC/precuneus was absent during the recall periods. Rather, high-gamma power in the PCC/precuneus was enhanced compared to the AI mostly during the 1390–1530 ms window prior to recall (ps<0.05, *Figure 3a*).

## AI response compared to PCC/precuneus during encoding and recall in the CATVFR task

We next examined high-gamma power in the CATVFR task. In this task, participants were presented with a list of words with consecutive pairs of words from a specific category (e.g., JEANS-COAT, GRAPE-PEACH, etc.) and subsequently asked to recall as many as possible from the original list ('Methods', *Appendix 1—tables 1, 3 and 7*, *Figure 1b*; *Qasim et al., 2023*).

### Encoding
High-gamma power in PCC/precuneus was suppressed compared to the AI during the 570–790 ms interval (ps<0.05, *Figure 3b*).

### Recall
High-gamma power mostly did not differ between AI and PCC/precuneus prior to recall (ps>0.05, *Figure 3b*).

## AI response compared to PCC/precuneus during encoding and recall in the PALVCR task

The PALVCR task also consisted of three periods: encoding, delay, and recall ('Methods', *Appendix 1—tables 1, 4 and 8*, *Figure 1c*). During encoding, a list of word-pairs was visually presented, and then participants were asked to verbally recall the cued word from memory during the recall periods.

### Encoding
High-gamma power in PCC/precuneus was suppressed compared to the AI during the memory encoding period, during the 470–950 ms and 2010–2790 ms windows (ps<0.05, *Figure 3c*).

### Recall
High-gamma power mostly did not differ between AI and PCC/precuneus prior to recall (ps>0.05, *Figure 3c*).

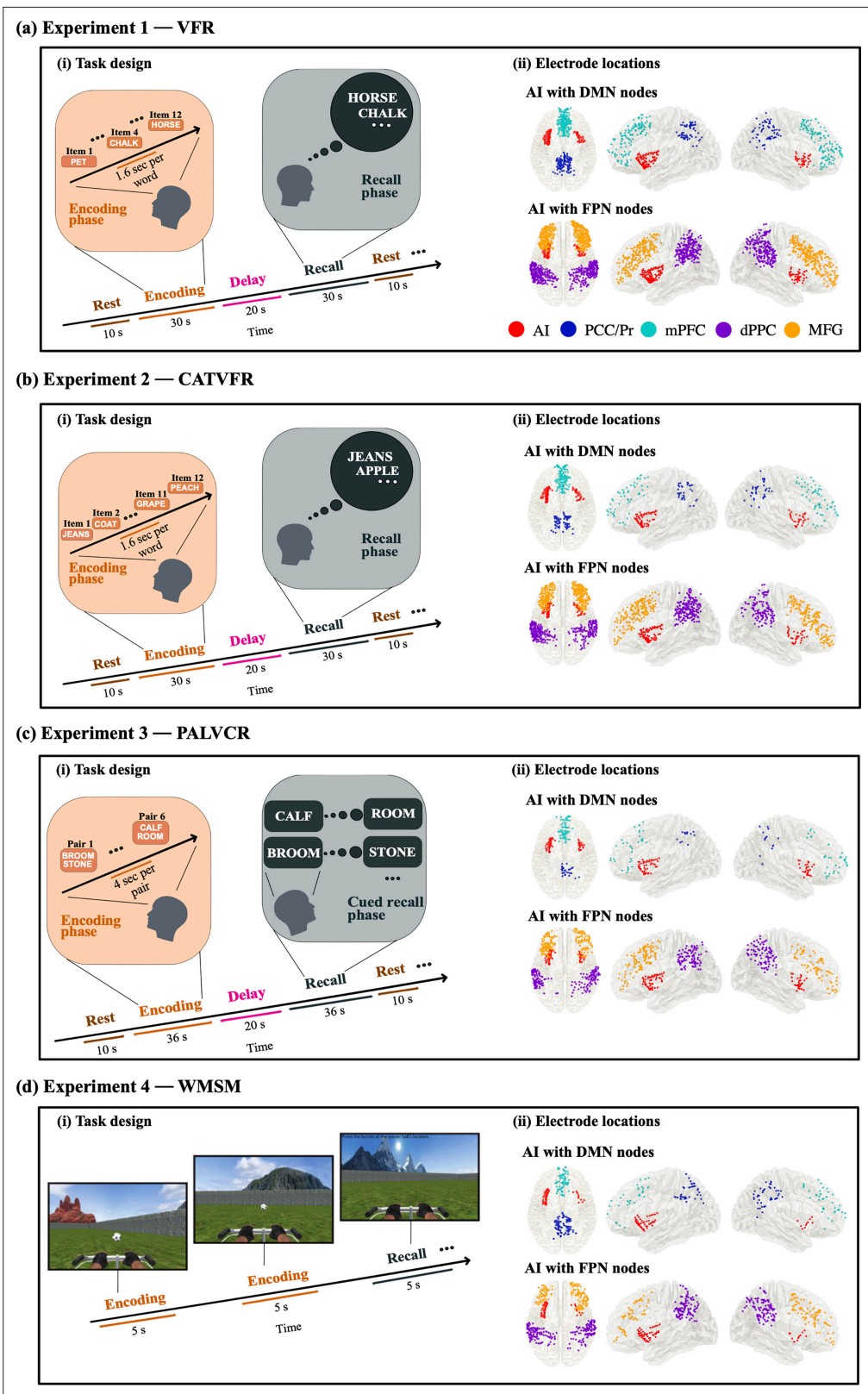

**Figure 1.** Task design of the encoding and recall periods of the memory experiments, and intracranial electroencephalography (iEEG) recording sites in AI, with DMN and FPN nodes. (**a**) Experiment 1: verbal free recall (VFR). (i) Task design of memory encoding and recall periods of the VFR experiment (see 'Methods' for details). Participants were first presented with a list of words in the encoding block and asked to recall as many as possible

*Figure 1 continued on next page*

*Figure 1 continued*

from the original list after a short delay. (ii) Electrode locations for AI with DMN nodes (top panel) and AI with FPN nodes (bottom panel), in the VFR experiment. Proportion of electrodes for AI, PCC/Pr, mPFC, dPPC, and MFG were 9, 8, 19, 32, and 32%, respectively, in the VFR experiment. (**b**) Experiment 2: categorized verbal free recall (CATVFR). (i) Task design of memory encoding and recall periods of the CATVFR experiment (see 'Methods' for details). Participants were presented with a list of words with consecutive pairs of words from a specific category (e.g., JEANS-COAT, GRAPE-PEACH, etc.) in the encoding block and subsequently asked to recall as many as possible from the original list after a short delay. (ii) Electrode locations for AI with DMN nodes (top panel) and AI with FPN nodes (bottom panel), in the CATVFR experiment. Proportion of electrodes for AI, PCC/Pr, mPFC, dPPC, and MFG were 10, 7, 11, 35, and 37%, respectively, in the CATVFR experiment. (**c**) Experiment 3: paired associates learning verbal cued recall (PALVCR). (i) Task design of memory encoding and recall periods of the PALVCR experiment (see 'Methods' for details). Participants were first presented with a list of six word-pairs in the encoding block, and after a short post-encoding delay, participants were shown a specific word-cue and asked to verbally recall the cued word from memory. (ii) Electrode locations for AI with DMN nodes (top panel) and AI with FPN nodes (bottom panel), in the PALVCR experiment. Proportion of electrodes for AI, PCC/Pr, mPFC, dPPC, and MFG were 14, 5, 13, 33, and 35%, respectively, in the PALVCR experiment. (**d**) Experiment 4: water maze spatial memory (WMSM). (i) Task design of memory encoding and recall periods of the WMSM experiment (see 'Methods' for details). Participants were shown objects in various locations during the encoding period and asked to retrieve the location of the objects during the recall period. (ii) Electrode locations for AI with DMN nodes (top panel) and AI with FPN nodes (bottom panel), in the WMSM experiment. Proportion of electrodes for AI, PCC/Pr, mPFC, dPPC, and MFG were 10, 15, 13, 38, and 24%, respectively, in the WMSM experiment. Overall, proportion of electrodes for VFR, CATVFR, PALVCR, and WMSM experiments were 43, 27, 15, and 15%, respectively. AI: anterior insula, DMN: default mode network, FPN: frontoparietal network; PCC: posterior cingulate cortex, Pr: precuneus, mPFC: medial prefrontal cortex, dPPC: dorsal posterior parietal cortex, MFG: middle frontal gyrus.

## AI response compared to PCC/precuneus during encoding and recall in the WMSM task

We next examined high-gamma power in the WMSM task. Participants performed multiple trials of a spatial memory task in a virtual navigation paradigm (*Goyal et al., 2018*; *Jacobs et al., 2016*; *Lee et al., 2018*) similar to the Morris water maze (*Morris, 1984*; 'Methods', *Appendix 1—tables 1, 5 and 9*, *Figure 1d*). Participants were shown objects in various locations during the encoding periods and asked to retrieve the location of the objects during the recall period.

### Encoding
High-gamma power in PCC/precuneus was suppressed compared to the AI, mostly during the 1390–2030 ms and 3150–4690 ms window (ps<0.05, *Figure 3d*).

### Recall
High-gamma power mostly did not differ between AI and PCC/precuneus (ps>0.05, *Figure 3d*).

## Replication of increased high-gamma power in AI compared to PCC/precuneus across four memory tasks

We next used replication BF analysis to estimate the degree of replicability of high-gamma power suppression of the PCC/precuneus compared to the AI during the memory encoding periods of the four tasks. We used the posterior distribution obtained from the VFR (primary) dataset as a prior distribution for the test of data from the CATVFR, PALVCR, and WMSM (replication) datasets (*Ly et al., 2019*; see 'Methods' for details). We used the encoding time windows for which we most consistently observed decrease of PCC/precuneus high-gamma power compared to the AI. These correspond to 110–1600 ms during the VFR task, 570–790 ms in the CATVFR task, 2010–2790 ms in the PALVCR task, and 3150–4690 ms in the WMSM task. We first averaged the high-gamma power across these strongest time windows for each task and then used replication BF analysis to estimate the degree of replicability of high-gamma power suppression of the PCC/precuneus compared to the AI.

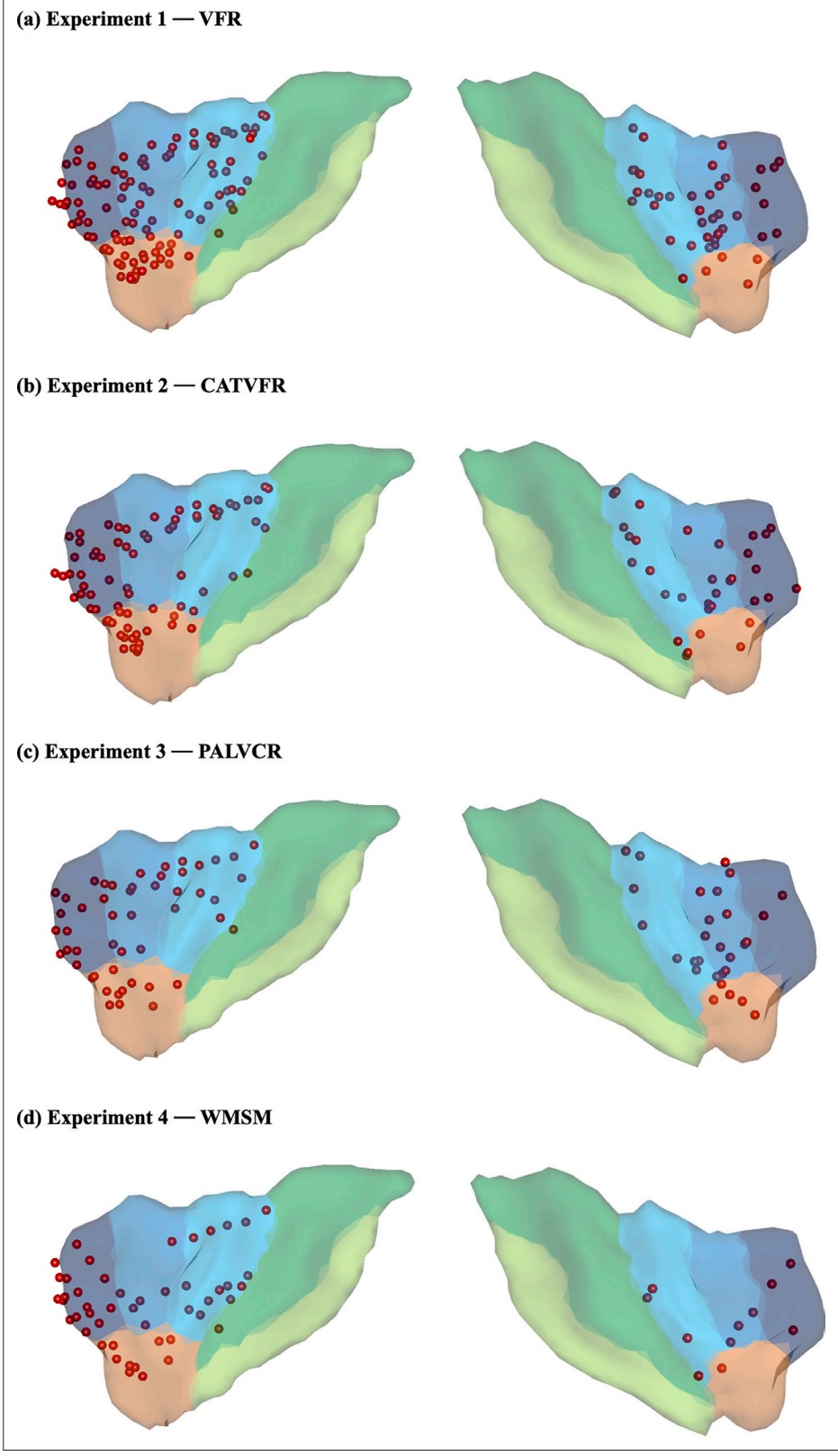

**Figure 2.** Anterior insula electrode locations (red) visualized on insular regions based on the atlas by *Faillenot et al., 2017*. Anterior insula (AI) is shown in blue, and posterior insula (PI) mask is shown in green (see 'Methods' for details). This atlas is based on probabilistic analysis of the anatomy of the insula with demarcations of the AI based on three short dorsal gyri and the PI, which encompasses two long and ventral gyri.

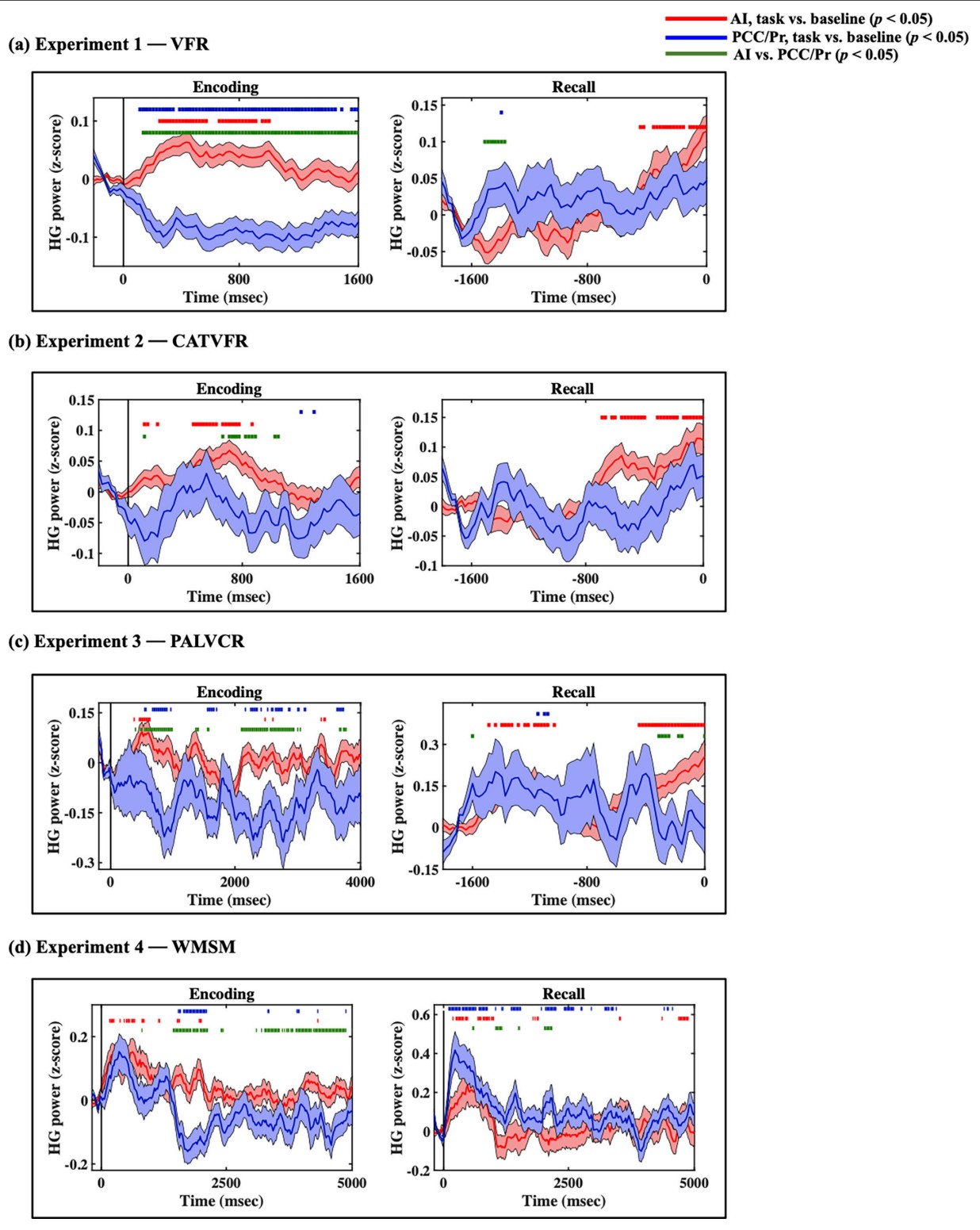

**Figure 3.** Intracranial electroencephalography (iEEG)-evoked response, quantified using high-gamma (HG) power, for anterior insula (AI) (red) and posterior cingulate cortex (PCC)/precuneus (blue) during (**a**) verbal free recall (VFR), (**b**) categorized verbal free recall (CATVFR), (**c**) paired associates learning verbal cued recall (PALVCR), and (**d**) water maze spatial memory (WMSM) experiments. Green horizontal lines denote greater HG power for AI compared to PCC/precuneus (ps<0.05). Red horizontal lines denote increase of AI response compared to the resting baseline during the encoding and recall periods (ps<0.05). Blue horizontal lines denote decrease of PCC/precuneus response compared to the baseline during the encoding periods and increase of PCC/precuneus response compared to the baseline during the recall periods (ps<0.05).

Findings corresponding to the high-gamma power suppression of the PCC/precuneus compared to AI were replicated in the PALVCR (BF 5.16e+1) and WMSM (BF 2.69e+8) tasks. These results demonstrate very high replicability of high-gamma power suppression of the PCC/precuneus compared to AI during memory encoding. The consistent suppression effect was localized only to the PCC/precuneus, but not to the mPFC node of the DMN or the dPPC and MFG nodes of the FPN (*Appendix 1—figures 1–3*).

In contrast to memory encoding, a similar analysis of high-gamma power did not reveal a consistent pattern of increased high-gamma power in AI and suppression of the PCC/precuneus across the four tasks during memory recall (*Figure 3*).

## AI and PCC/precuneus response during encoding and recall compared to resting baseline

We examined whether AI and PCC/precuneus high-gamma power response during the encoding and recall periods are enhanced or suppressed compared to the baseline periods. High-gamma power in the AI was increased compared to the resting baseline during both the encoding and recall periods, and across all four tasks (ps<0.05, *Figure 3*). This suggests an enhanced role for the AI during both memory encoding and recall compared to resting baseline.

In contrast, high-gamma power in the PCC/precuneus was reduced compared to the resting baseline in three tasks—VFR, PALVCR, and WMSM—providing direct evidence for PCC/precuneus suppression during memory encoding (*Figure 3*). We did not find any increased high-gamma power activity in the PCC/precuneus, compared to the baseline, during memory retrieval (*Figure 3*). These results provide evidence for PCC/precuneus suppression compared to both the AI and resting baseline during externally triggered stimuli during encoding.

High-gamma power for other brain areas compared to resting baseline were not consistent across tasks (*Appendix 1—figures 1–3*).

## Directed information flow from the AI to the DMN during encoding

We next examined directed information flow from the AI to the PCC/precuneus and mPFC nodes of the DMN during the memory encoding periods of the VFR task. We used PTE (*Lobier et al., 2014*) to evaluate directed influences from the AI to the PCC/precuneus and mPFC and vice versa. Informed by recent electrophysiology studies in nonhuman primates, which suggest that broadband field potentials activity, rather than narrowband, governs information flow in the brain (*Davis et al., 2020*; *Davis et al., 2022*), we examined PTE in a 0.5–80 Hz frequency spectrum to assess dynamic-directed influences of the AI on the DMN.

Directed information flow from the AI to the PCC/precuneus ($F(1, 264) = 59.36$, p<0.001, Cohen's $d = 0.95$) and mPFC ($F(1, 208) = 13.96$, p<0.001, Cohen's $d = 0.52$) was higher than the reverse (*Figure 4a*).

### Replication across three experiments with BF

We used replication BF analysis to estimate the degree of replicability of direction of information flow across the four experiments (*Table 1a*, *Figure 4b–d*, also see Appendix Results for detailed stats related to the CATVFR, PALVCR, and WMSM experiments). Findings corresponding to the direction of information flow between the AI and the PCC/precuneus during memory encoding were replicated all three tasks (BFs 9.31e+5, 1.44e+4, and 1.68e+18 for CATVFR, PALVCR, and WMSM respectively). Findings corresponding to the direction of information flow between the AI and mPFC during memory encoding were also replicated across all three tasks (BFs 4.10e+1, 8.78e+0, and 5.34e+5 for CATVFR, PALVCR, and WMSM respectively). This highly consistent pattern of results was not observed in any other frequency band (delta-theta [0.5–8 Hz], alpha [8–12 Hz], beta [12–30 Hz], gamma [30–80 Hz], and high-gamma [80–160 Hz]; results not shown). These results demonstrate very high replicability of directed information flow from the AI to the DMN nodes during memory encoding.

These results demonstrate robust directed information flow from the AI to the PCC/precuneus and mPFC nodes of the DMN during memory encoding.

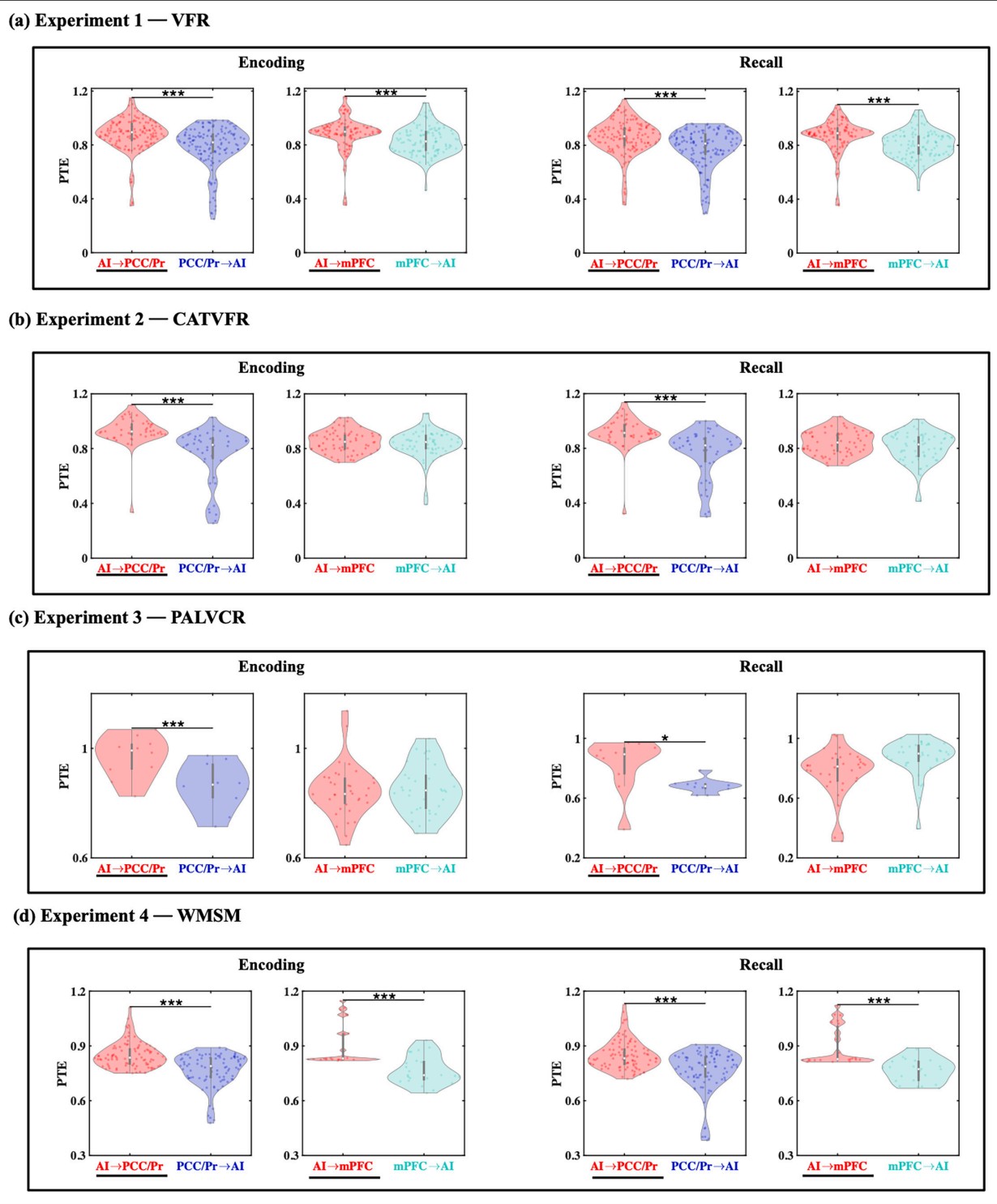

**Figure 4.** Directed information flow between the anterior insula (AI) and the posterior cingulate cortex (PCC)/precuneus and medial prefrontal cortex (mPFC) nodes of the default mode network (DMN), across verbal and spatial memory domains, measured using phase transfer entropy (PTE). (**a**) Experiment 1: verbal free recall (VFR). The AI showed higher directed information flow to the PCC/precuneus (AI → PCC/Pr) compared to the reverse direction (PCC/Pr → AI) (n = 142) during both encoding and recall. The AI also showed higher directed information flow to the mPFC (AI → mPFC) compared to the reverse direction (mPFC → AI) (n = 112) during both memory encoding and recall. (**b**) Experiment 2: categorized verbal free recall (CATVFR). The AI showed higher directed information flow to the PCC/precuneus (AI → PCC/Pr) compared to the reverse direction (PCC/Pr → AI) (n = 46) during both encoding and recall. (**c**) Experiment 3: paired associates learning verbal cued recall (PALVCR). The AI showed higher directed information flow to the PCC/precuneus (AI → PCC/Pr) compared to the reverse direction (PCC/Pr → AI) (n = 10) during both encoding and recall.

*Figure 4 continued on next page*

*Figure 4 continued*

(**d**) Experiment 4: water maze spatial memory (WMSM). The AI showed higher directed information flow to PCC/precuneus (AI → PCC/Pr) than the reverse (PCC/Pr → AI) (n = 91) during both spatial memory encoding and recall. The AI also showed higher directed information flow to mPFC (AI → mPFC) than the reverse (mPFC → AI) (n = 23) during both spatial memory encoding and recall. In each panel, the direction for which PTE is higher is underlined. White dot in each violin plot represents median PTE across electrode pairs. ***p<0.001, * p<0.05.

## Directed information flow from the AI to the DMN during recall

Next, we examined directed influences of the AI on PCC/precuneus and mPFC during the recall phase of the verbal episodic memory task. During memory recall, directed information flow from the AI to the PCC/precuneus ($F(1, 264) = 43.09$, p<0.001, Cohen's $d = 0.81$) and mPFC ($F(1, 211) = 21.94$, p<0.001, Cohen's $d = 0.65$) was higher than the reverse (*Figure 4a*).

### Replication across three experiments with BF

We next repeated the replication BF analysis for the recall periods of the memory tasks (*Table 1b*, *Figure 4b–d*, also see Appendix Results for detailed stats related to the CATVFR, PALVCR, and WMSM experiments). Findings corresponding to the direction of information flow between the AI and the PCC/precuneus during memory recall were replicated across all three tasks (BFs 1.30e+5, 6.74e+0, and 2.54e+10 for CATVFR, PALVCR, and WMSM respectively). Findings corresponding to the direction of information flow between the AI and the mPFC during memory recall were also replicated across the CATVFR and WMSM tasks (BFs 2.02e+1 and 1.32e+4 respectively).

These results demonstrate very high replicability of directed information flow from the AI to the DMN nodes across verbal and spatial memory tasks during both memory encoding and recall.

**Table 1.** Replicability of findings of directed interactions of the AI with the DMN and FPN nodes for different memory experiments during (a) memory encoding and (b) memory recall.

The verbal free recall (VFR) task was considered the original dataset and the categorized verbal free recall (CATVFR), paired associates learning verbal cued recall (PALVCR), and water maze spatial memory (WMSM) tasks were considered replication datasets, and Bayes factor (BF) for replication was calculated for pairwise tasks (VFR vs. T, where T can be CATVFR, PALVCR, or WMSM task). Significant BF results (BF > 3) are indicated in bold. AI: anterior insula, DMN: default mode network; FPN: frontoparietal network, PCC: posterior cingulate cortex, Pr: precuneus, mPFC: medial prefrontal cortex, dPPC: dorsal posterior parietal cortex, MFG: middle frontal gyrus.

**(a) Memory encoding**

| Finding | BF for VFR-CATVFR replication | BF for VFR-PALVCR replication | BF for VFR-WMSN replication |
|---|---|---|---|
| AI→PCC/Pr >PCC/Pr→AI | 9.31E+05 | 1.44E+04 | 1.68E+18 |
| AI→mPFC>mPFC→AI | 4.10E+01 | 8.78E+00 | 5.34E+05 |
| AI→dPPC>dPPC→AI | 3.95E+43 | 2.33E+26 | 3.25E+40 |
| AI→MFG>MFG→AI | 1.49E+51 | 1.61E+33 | 2.35E+27 |

**(b) Memory recall**

| Finding | BF for VFR-CATVFR replication | BF for VFR-PALVCR replication | BF for VFR-WMSN replication |
|---|---|---|---|
| AI→PCC/Pr>PCC/Pr→AI | 1.30E+05 | 6.74E+00 | 2.54E+10 |
| AI→mPFC>mPFC→AI | 2.02E+01 | 3.52E-05 | 1.32E+04 |
| AI→dPPC>dPPC→AI | 7.04E+38 | 2.98E+45 | 4.51E+27 |
| AI→MFG>MFG→AI | 1.74E+54 | 5.72E+52 | 6.90E+27 |

## Directed information flow from AI to FPN nodes during memory encoding

We next probed directed information flow between the AI and FPN nodes during the encoding periods of the VFR task. Directed information flow from the AI to the dPPC ($F$(1, 1143) = 11.69, p<0.001, Cohen's $d$ = 0.20) and MFG ($F$(1, 1245) = 21.69, p<0.001, Cohen's $d$ = 0.26) was higher than the reverse during memory encoding of the VFR task (*Figure 5a*).

### Replication across three experiments with BF

We used replication BF analysis for the replication of AI-directed influences on FPN nodes during the encoding phase of the memory tasks (*Table 1a*, *Figure 5b–d*, Appendix Results). Similarly, we also obtained very high BFs for findings corresponding to the direction of information flow between the AI and dPPC (BFs > 2.33e+26) and also between the AI and MFG (BFs > 2.35e+27), across all three tasks.

These results demonstrate that the AI has robust directed information flow to the dPPC and MFG nodes of the FPN during memory encoding.

## Directed information flow from AI to FPN nodes during memory recall

Directed influences from the AI to the dPPC ($F$(1, 1143) = 17.47, p<0.001, Cohen's $d$ = 0.25) and MFG ($F$(1, 1246) = 42.75, p<0.001, Cohen's $d$ = 0.37) were higher than the reverse during memory recall of the VFR task (*Figure 5a*).

### Replication across three experiments with BF

We also found very high BFs for findings corresponding to the direction of information flow between the AI and the dPPC (BFs > 4.51e+27) and MFG (BFs > 6.90e+27) nodes of the FPN across the CATVFR, PALVCR, and WMSM tasks during the memory recall period (*Table 1b*, *Figure 5b–d*, Appendix Results).

These results demonstrate very high replicability of directed information flow from the AI to the FPN nodes across multiple memory experiments during both memory encoding and recall.

## Comparison of directed information flow: AI vs. IFG

To examine the specificity of the AI-directed information flow to the DMN and FPN, we conducted a control analysis using electrodes implanted in the IFG (BA 44). The IFG serves as an ideal control region due to its anatomical adjacency to the AI, its involvement in a wide range of cognitive control functions including response inhibition (*Cai et al., 2014*), and its frequent co-activation with the AI in fMRI studies. Furthermore, the IFG has been associated with controlled retrieval of memory (*Badre et al., 2005*; *Badre and Wagner, 2007*; *Wagner et al., 2001*), making it a compelling region for comparison.

Our analysis revealed a striking contrast between the AI and IFG in their patterns of directed information flow. While the AI exhibited strong directed influences on both the DMN and FPN, the IFG showed the opposite pattern. Specifically, both the DMN and FPN demonstrated higher influence on the IFG than the reverse during both encoding and recall periods, and across all four memory experiments (*Appendix 1—figures 4 and 5*).

To quantify this difference more precisely, we calculated the net outflow for both regions, defined as the difference (PTE(out) – PTE(in), see 'Methods' for details). This analysis revealed that the AI's net outflow was significantly higher than that of the IFG during both encoding and recall phases, a finding replicated across all four experiments (all ps<0.001) (*Appendix 1—figure 6*).

These results not only highlight the unique role of the AI in orchestrating large-scale network dynamics during memory processes but also demonstrate the specificity of this function compared to an anatomically adjacent and functionally relevant region. The consistent pattern across diverse memory tasks and experimental phases underscores the robustness of the AI's role as an outflow hub during memory formation and retrieval.

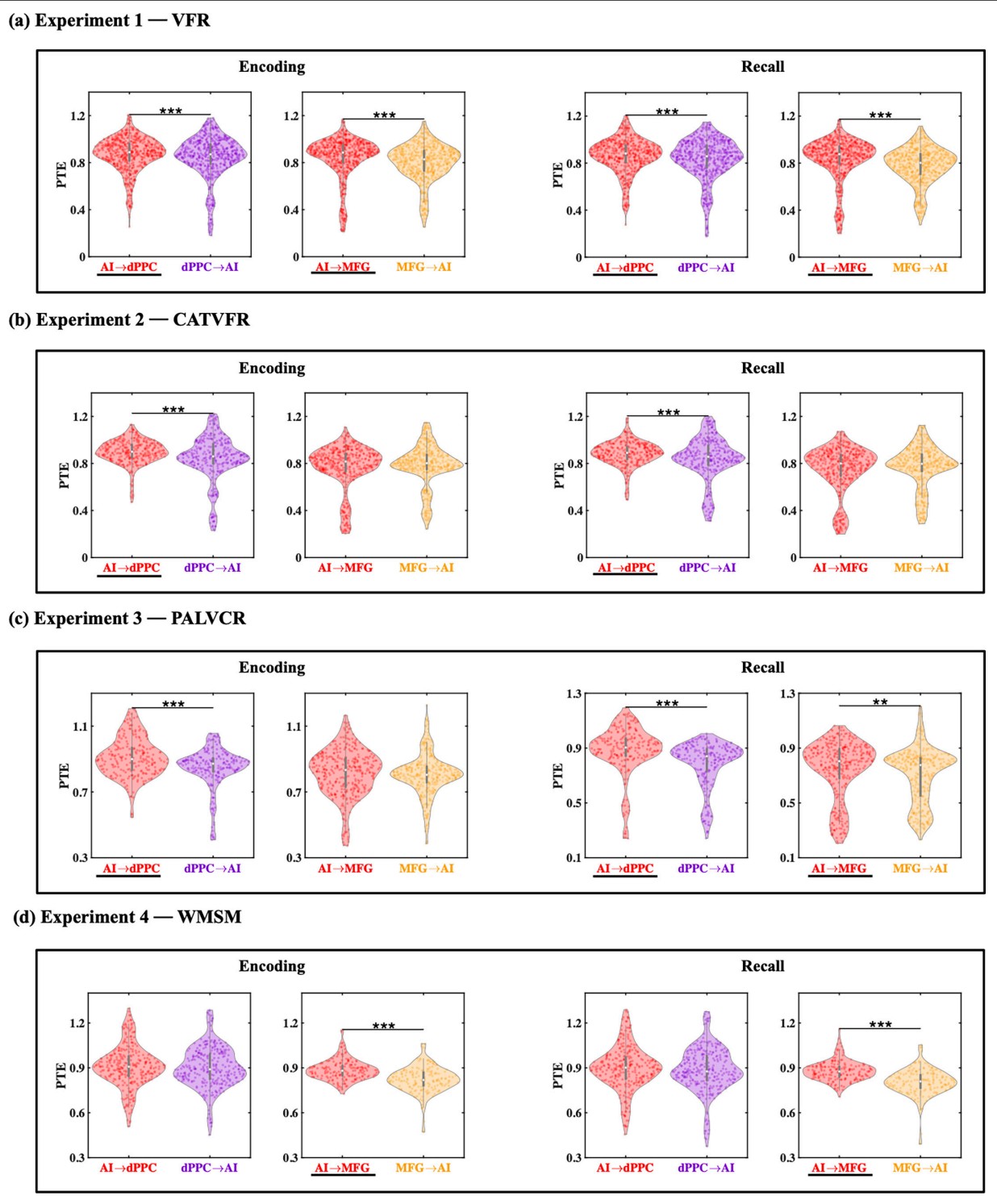

**Figure 5.** Directed information flow between the anterior insula (AI) and the dorsal posterior parietal cortex (dPPC) and middle frontal gyrus (MFG) nodes of the frontoparietal network (FPN) across verbal and spatial memory domains. (**a**) Experiment 1: verbal free recall (VFR). The AI showed higher directed information flow to the dPPC (AI → dPPC) compared to the reverse direction (dPPC → AI) (n = 586) during both encoding and recall. The AI also showed higher directed information flow to the MFG (AI → MFG) compared to the reverse direction (MFG → AI) (n = 642) during both memory encoding and recall. (**b**) Experiment 2: categorized verbal free recall (CATVFR). The AI showed higher directed information flow to the dPPC (AI → dPPC) compared to the reverse direction (dPPC → AI) (n = 327) during both encoding and recall. (**c**) Experiment 3: paired associates learning verbal cued recall (PALVCR). The AI showed higher directed information flow to the dPPC (AI → dPPC) compared to the reverse direction (dPPC → AI) (n = 242) during both encoding and recall. The AI also showed higher directed information flow to the MFG (AI → MFG) compared to the reverse direction (MFG → AI)

*Figure 5 continued on next page*

*Figure 5 continued*

(n = 362) during memory recall. (**d**) Experiment 4: water maze spatial memory (WMSM). The AI showed higher directed information flow to MFG (AI → MFG) than the reverse (MFG → AI) (n = 177) during both spatial memory encoding and recall. In each panel, the direction for which PTE is higher is underlined. ***p<0.001, **p<0.01.

## Enhanced information flow from the AI to the DMN and FPN during episodic memory processing compared to resting-state baseline

We next examined whether directed information flow from the AI to the DMN and FPN nodes during the memory tasks differed from the resting-state baseline. Resting-state baselines were extracted immediately before the start of the task sessions and the duration of task and rest epochs were matched to ensure that differences in network dynamics could not be explained by differences in duration of the epochs. Directed information flow from the AI to both the DMN and FPN was higher during both the memory encoding and recall phases and across the four experiments compared to baseline in all but two cases (*Appendix 1—figures 7 and 8*).

To further elucidate the task-specific role of the AI, we compared its net outward directed influence during memory tasks to that observed during resting state. We quantified this influence as the difference between outgoing and incoming information flow (PTE(out) − PTE(in)). This analysis revealed that the AI's net outflow was significantly enhanced during both encoding and recall phases of memory tasks compared to resting state in all but one case (ps<0.05) (*Appendix 1—figure 9*). This pattern was consistently observed across all four experiments. These findings provide strong evidence for enhanced role of AI-directed information flow to the DMN and FPN during memory processing compared to the resting state.

## Differential information flow from the AI to the DMN and FPN for successfully recalled and forgotten memory trials

We examined memory effects by comparing PTE between successfully recalled and forgotten memory trials. However, this analysis did not reveal differences in directed influence from the AI on the DMN and FPN or the reverse between successfully recalled and forgotten memory trials during the encoding as well as recall periods in any of the memory experiments (all ps>0.05) (*Appendix 1—figures 10 and 11*).

## Outflow hub during encoding and recall

fMRI studies have suggested that the AI acts as an outflow hub with respect to interactions with the DMN and FPN (*Sridharan et al., 2008*). To test the potential neural basis of this finding, we calculated net outflow (PTE(out) − PTE(in)) as the difference between the total outgoing information and total incoming information.

### Encoding

This analysis revealed that the net outflow from the AI is positive and higher than the PCC/precuneus ($F(1, 3319)$ = 154.8, p<0.001, Cohen's $d$ = 0.43) node of the DMN in the VFR task (*Figure 6a*).

This analysis also revealed that the net outflow from the AI is higher than both the dPPC ($F(1, 5346)$ = 67.87, p<0.001, Cohen's $d$ = 0.23) and MFG ($F(1, 6920)$ = 132.74, p<0.001, Cohen's $d$ = 0.28) nodes of the FPN in the VFR task (*Figure 6a*).

Findings in the VFR task were also replicated across the CATVFR, PALVCR, and WMSM tasks, where we found that the net outflow from the AI is higher than the PCC/precuneus and mPFC nodes of the DMN and the dPPC and MFG nodes of the FPN (*Figure 6b–d*, also see Appendix Results for detailed stats related to the CATVFR, PALVCR, and WMSM experiments).

### Recall

Net outflow from the AI is positive and higher than both PCC/precuneus ($F(1, 3287)$ = 151.21, p<0.001, Cohen's $d$ = 0.43) and mPFC ($F(1, 4694)$ = 7.81, p<0.01, Cohen's $d$ = 0.08) during the recall phase of the VFR task (*Figure 6a*).

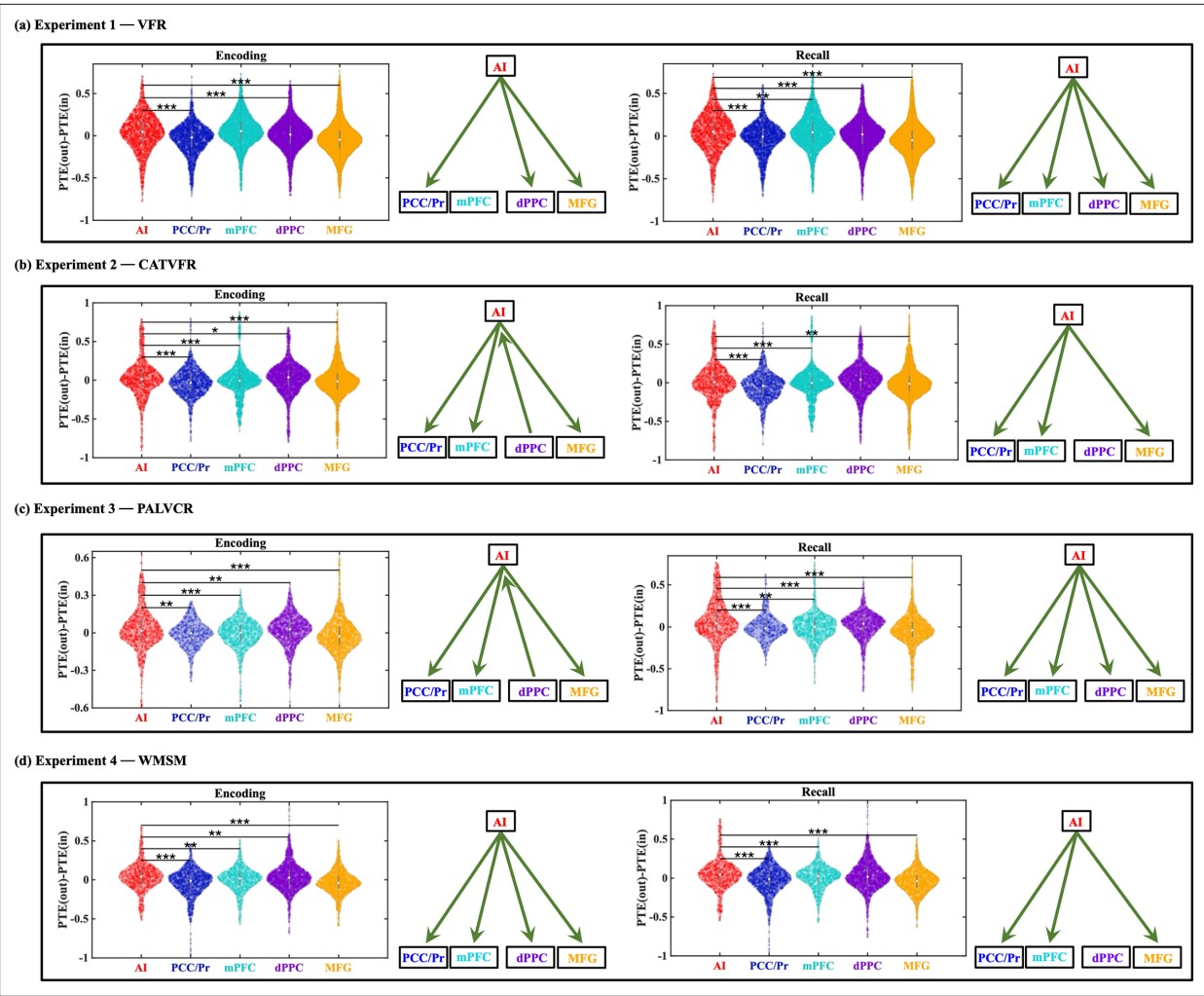

**Figure 6.** The anterior insula (AI) is an outflow hub in its interactions with the default mode network (DMN) and frontoparietal network (FPN) during encoding and recall periods, and across memory experiments. In each panel, the net direction of information flow between the AI and the DMN and FPN nodes is indicated by green arrows on the right. \*\*\*p<0.001, \*\*p<0.01, \*p<0.05.

Net outflow from the AI is also higher than both the dPPC ($F$(1, 5388) = 90.71, p<0.001, Cohen's $d$ = 0.26) and MFG ($F$(1, 6945) = 167.14, p<0.001, Cohen's $d$ = 0.31) nodes of the FPN during recall (*Figure 6a*).

Crucially, these findings were also replicated across the CATVFR, PALVCR, and WMSM tasks and during both encoding and recall periods (*Figure 6b–d*, also see Appendix Results for detailed stats related to the CATVFR, PALVCR, and WMSM experiments). Together, these results demonstrate that the AI is an outflow hub in its interactions with the PCC/precuneus and mPFC nodes of the DMN and also the dPPC and MFG nodes of the FPN, during both verbal and spatial memory encoding and recall.

## Narrowband phase synchronization between the AI and the DMN and FPN during encoding and recall compared to resting baseline

We next directly compared the phase locking values (PLVs) (see 'Methods' for details) between the AI and the PCC/precuneus and mPFC nodes of the DMN and also the dPPC and MFG nodes of the FPN for the encoding and the recall periods compared to resting baseline. However, narrowband PLV values did not significantly differ between the encoding/recall vs. rest periods in any of the delta-theta (0.5–8 Hz), alpha (8–12 Hz), beta (12–30 Hz), gamma (30–80 Hz), and high-gamma (80–160 Hz)

frequency bands. These results indicate that PTE, rather than phase synchronization, more robustly captures the AI dynamic interactions with the DMN and the FPN.

## Discussion

Our study investigated the electrophysiological underpinnings of large-scale brain network interactions during episodic memory processes, focusing on the dynamic interplay between the SN, DMN, and FPN as conceptualized in the triple network model (*Cai et al., 2021*; *Menon, 2011*; *Menon, 2023*). This model has been primarily investigated in the context of cognitive control tasks (*Cai et al., 2021*; *Menon, 2011*; *Menon, 2023*). However, its applicability to memory processes remains less explored, particularly at the electrophysiological level. We elucidated how these three networks interact during different phases of memory processing, focusing on the directed information flow between key cortical nodes. The triple network model posits distinct roles for each network: the SN, anchored by the AI, is thought to detect behaviorally relevant stimuli and orient attention toward information that needs to be encoded; the DMN is implicated in internally driven processes and memory recall; and the FPN contributes to the maintenance and manipulation of information in working memory, processes critical for both encoding and recall (*Badre et al., 2005*; *Badre and Wagner, 2007*; *Wagner et al., 2001*; *Wagner et al., 2005*). By leveraging intracranial EEG data from a large cohort of participants across four diverse memory tasks, we sought to provide a comprehensive, high-temporal resolution account of these network dynamics.

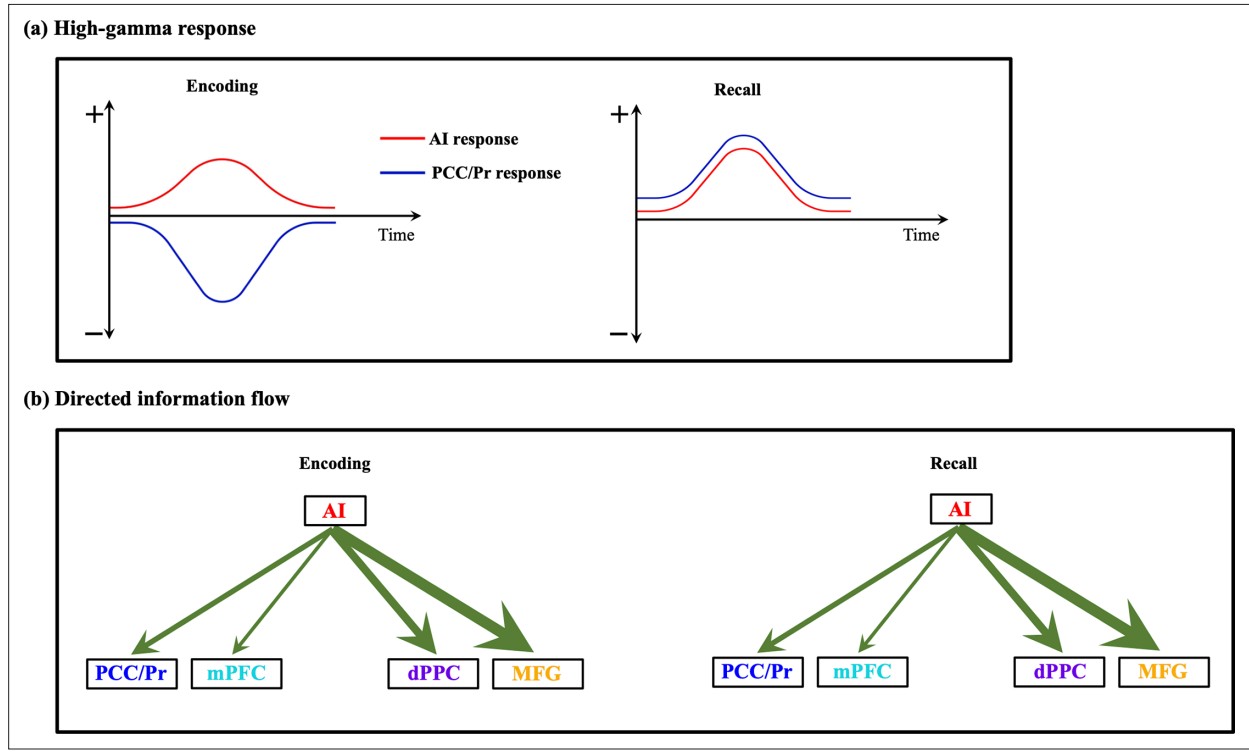

**Figure 7.** Schematic illustration of key findings related to the intracranial electrophysiology of the triple network model in human episodic memory. (**a**) High-gamma response. Our analysis of local neuronal activity revealed consistent suppression of high-gamma power in the posterior cingulate cortex (PCC)/precuneus compared to the anterior insula (AI) during encoding periods across all four episodic memory experiments. We did not consistently observe any significant differences in high-gamma band power between AI and the medial prefrontal cortex (mPFC) node of the default mode network (DMN) or the dorsal PPC (dPPC) and middle frontal gyrus (MFG) nodes of the frontoparietal network (FPN) during the encoding periods across the four episodic memory experiments. In contrast, we detected similar high-gamma band power in the PCC/precuneus relative to the AI during the recall periods. (**b**) Directed information flow. Despite variable patterns of local activation and suppression across DMN and FPN nodes during memory encoding and recall, we found stronger directed influence (denoted by green arrows, thickness of arrows denotes degree of replicability across experiments, see *Table 1*) by the AI on both the DMN as well as the FPN nodes than the reverse across all four memory experiments, and during both encoding and recall periods.

We discovered that the AI, a crucial node of the SN, exerts strong directed influence on both the DMN and FPN during both memory encoding and recall. This finding was consistently observed across multiple experiments spanning verbal and spatial memory domains, highlighting the robustness and generalizability of our results. Importantly, our study extends the applicability of the triple network model beyond cognitive control tasks to episodic memory processes, thus broadening its explanatory power in the context of memory formation. Furthermore, we observed a distinctive suppression of high-gamma power in the PCC/precuneus node of the DMN compared to the AI during memory encoding, suggesting a task-specific functional down-regulation of this region. Our findings significantly advance the understanding of the SN's role in modulating large-scale brain networks during episodic memory formation and underscore the importance of the triple network model in domain-general coordination of brain networks (*Figure 7*).

## Investigating directed inter-network interactions using iEEG and PTE

Dynamic interactions between the AI and the DMN and FPN are hypothesized to shape human cognition (*Cai et al., 2016*; *Cai et al., 2014*; *Dosenbach et al., 2008*; *Dosenbach et al., 2006*; *Menon, 2015b*; *Menon and Uddin, 2010*). Although fMRI research has suggested that the AI plays a pivotal role in the task-dependent engagement and disengagement of the DMN and FPN across diverse cognitive tasks (*Menon and Uddin, 2010*; *Sridharan et al., 2008*), the neuronal basis of these results or the possibility of their being artifacts arising from slow dynamics and regional variation in the hemodynamic response inherent to fMRI signals remained unclear. To address these ambiguities, our analysis focused on casual interactions involving the AI and leveraged the high temporal resolution of iEEG signals. By investigating the directionality of information flow, we aimed to overcome the temporal resolution limitations of fMRI signals, providing a more mechanistic understanding of the AI's role in modulating the DMN and FPN during memory formation. To assess reproducibility, we scrutinized network interactions across four different episodic memory tasks involving VFR, CATVFR, PALVCR, and WMSM tasks (*Solomon et al., 2019*).

We employed PTE, a robust metric of nonlinear and nonstationary dynamics, to investigate dynamic interactions between the AI and four key cortical nodes of the DMN and FPN. PTE assesses the ability of one time series to predict future values of another, estimating time-delayed directed influences, and is superior to methods like phase locking or coherence as it captures nonlinear and nonstationary interactions (*Bassett and Sporns, 2017*; *Hillebrand et al., 2016*; *Lobier et al., 2014*). PTE offers a robust and powerful tool for characterizing information flow between brain regions based on phase coupling (*Hillebrand et al., 2016*; *Lobier et al., 2014*; *Wang et al., 2017*) and has been successfully utilized in our previous studies (*Das et al., 2022c*; *Das and Menon, 2020*; *Das and Menon, 2021*; *Das and Menon, 2022b*; *Das and Menon, 2023*).

## Broadband-directed influences of the AI on DMN and FPN

Informed by recent electrophysiology studies in nonhuman primates, which suggest that broadband field potentials activity, rather than narrowband, governs information flow in the brain (*Davis et al., 2020*; *Davis et al., 2022*), we first examined PTE in a 0.5–80 Hz frequency spectrum to assess dynamic-directed influences of the AI on the DMN and FPN. Our analysis revealed that AI exerts stronger influences on the PCC/precuneus and mPFC nodes of the DMN than the reverse. A similar pattern also emerged for FPN nodes, with the AI displaying stronger directed influences on the dPPC and MFG than the reverse. Crucially, this asymmetric pattern of directed information flow was replicated across all four memory tasks. Moreover, this pattern also held during the encoding and recall of memory phases of all four tasks.

## Replicability across memory tasks

Replication, a critical issue in all of systems neuroscience, is particularly challenging in the field of intracranial EEG studies, where data acquisition from patients is inherently difficult. Compounding this issue is the virtual absence of data sharing and the substantial complexities involved in collecting electrophysiological data across distributed brain regions (*Das and Menon, 2022b*). Consequently, one of our study's major objectives was to reproduce our findings across multiple experiments, bridging verbal and spatial memory domains and task phases. To quantify the degree of replicability of our findings across these domains, we employed replication BF analysis (*Ly et al., 2019*; *Verhagen and*

*Wagenmakers, 2014*). Our analysis revealed very high replication BFs related to replication of information flow from the AI to the DMN and FPN (*Table 1*). Specifically, the BFs associated with the replication of direction of information flow between the AI and the DMN and FPN were decisive (BFs > 100), demonstrating consistent results across various memory tasks and contexts.

## Task-specific enhancement of AI's directed influence: Contrasts with IFG and resting state

Our analysis revealed a striking contrast between the AI and IFG in their patterns of directed information flow. While the AI exhibited strong directed influences on both the DMN and FPN, the IFG demonstrated an inverse relationship. Specifically, both the DMN and FPN exerted higher influence on the IFG than vice versa, a pattern that held consistent across both encoding and recall periods, and throughout all four memory experiments (*Appendix 1—figures 4 and 5*). Our analysis also revealed that the AI's net outflow was significantly higher than that of the IFG during both encoding and recall phases, a finding replicated across all four experiments.

Furthermore, we compared the AI's net outward directed influence during memory tasks to that observed during resting state. This analysis showed that the AI's net outflow was significantly enhanced during both encoding and recall phases of memory tasks compared to resting state, consistently across all four experiments. This task-specific enhancement suggests that the AI's role in coordinating large-scale network dynamics is specifically amplified during memory processes.

These results not only highlight the unique role of the AI in orchestrating large-scale network dynamics during memory processes but also demonstrate the specificity of this function compared to an anatomically adjacent and functionally relevant region implicated in cognitive control (*Badre et al., 2005*; *Badre and Wagner, 2007*; *Cai et al., 2014*; *Wagner et al., 2001*). The consistent pattern across diverse memory tasks and experimental phases underscores the robustness of the AI's role in memory-related network interactions.

## High-gamma power suppression in the PCC/precuneus during encoding, but not recall

Our analysis of local neuronal activity revealed a consistent and specific pattern of high-gamma power suppression in the PCC/precuneus compared to the AI during memory encoding across all four episodic memory tasks. This finding aligns with the typical deactivation of DMN nodes during attention-demanding tasks (*Wen et al., 2013*), while also extending our understanding of the DMN's role in episodic memory formation (*Buckner et al., 2008*; *Menon, 2023*).

Importantly, this suppression effect was confined to the PCC/precuneus within the DMN, with no parallel reductions observed in the mPFC. Moreover, suppression of the PCC/precuneus was stronger compared to the dPPC and MFG nodes of the FPN (*Appendix 1—figures 12 and 13*). Bayesian replication analysis substantiated the high degree of replicability of this PCC/precuneus suppression effect across tasks (BFs > 5.16e+1). These findings extend previous fMRI studies reporting DMN suppression during attention to external stimuli (*Bressler and Menon, 2010*; *Raichle et al., 2001*; *Seeley et al., 2007*) and complement optogenetic research in rodents' brains demonstrating AI-induced suppression of DMN regions (*Menon, 2023*).

High-gamma activity (80–160 Hz) is a reliable indicator of localized, task-related neural processing, often associated with synchronized activity of local neural populations and elevated neuronal spiking (*Canolty and Knight, 2010*). High-gamma activity (typically ranging from 80 to 160 Hz) has been reliably implicated in various cognitive tasks across sensory modalities, including visual (*Lachaux et al., 2005*; *Tallon-Baudry et al., 2005*), auditory (*Crone et al., 2001*; *Edwards et al., 2005*), and across cognitive domains, including working memory (*Canolty et al., 2006*; *Mainy et al., 2007*) and episodic memory (*Daitch and Parvizi, 2018*; *Sederberg et al., 2007*). The suppression we observed during encoding likely reflects functional down-regulation of the PCC/precuneus, potentially to minimize interference from internally oriented processes during the encoding of external information.

In contrast, during memory recall, we observed different patterns of activity. In the three verbal tasks (VFR, CATVFR, and PALVCR), PCC/precuneus activity showed enhanced responses compared to the AI in the 1–1.6 s window prior to word production. However, it is crucial to note that our analysis was time-locked to word production rather than the onset of internal retrieval processes. In the spatial memory task WMSM, the PCC/precuneus exhibited an earlier onset and enhanced activity compared

to the AI. This task may provide a clearer window into recall processes: findings align with the view that DMN nodes may play a crucial role in triggering internal recall processes. However, the precise timing of internal retrieval initiation remains a challenge in verbal tasks, potentially limiting our ability to capture the full dynamics of regional activity, and its replicability, during early stages of recall.

The observed high-gamma suppression in the PCC/precuneus during encoding, but not recall, likely reflects the distinct cognitive demands of these memory phases. Encoding primarily involves externally driven processes, requiring attention to and processing of incoming stimuli. In contrast, recall is predominantly internally driven, relying on the retrieval and reconstruction of stored information. This dissociation in PCC/precuneus activity aligns with its known role in the DMN, which typically shows deactivation during externally oriented tasks and activation during internally directed cognition. This pattern of activity underscores the flexible and context-dependent functioning of brain regions within large-scale networks, adapting their engagement to support different aspects of memory processing.

## Broadband- vs. high-gamma-directed influences

Notably, our findings reveal a robust and consistent directed influence exerted by the AI on all nodes of both the DMN and the FPN, extending across all four memory tasks and both memory encoding and recall phases. These directed influences were prominently manifested in broadband signals. Interestingly, such directed influences were not observed in the high-gamma frequency range (80–160 Hz). This absence aligns with current models positing that high-gamma activity is more likely to reflect localized processing, while lower-frequency bands are implicated in longer-range network communication and coordination (*Bastos et al., 2015*; *Das et al., 2022c*; *Das and Menon, 2020*; *Das and Menon, 2021*; *Das and Menon, 2023*; *Miller et al., 2007*). More generally, our findings emphasize that it is crucial to differentiate between high-gamma activity ($f > 80$ Hz) and sub-high-gamma ($f < 80$ Hz) fluctuations as these signal types are indicative of different underlying physiological processes, each with distinct implications for understanding neural network dynamics.

## Successful and unsuccessful memory effects engage similar AI-directed circuits

Our analysis revealed no significant differences in directed connectivity between successfully recalled and forgotten memory trials, suggesting that the reported effects may not be specific to successful memory formation and may be related to attentional or other general cognitive processing rather than memory processing per se. While our study provides valuable insights into the interactions between the AI and the DMN and FPN during cognitive tasks involving verbal and spatial information processing during memory tasks, it is crucial to acknowledge that these interactions may not be unique to memory processes. The AI's directed influence on the DMN and FPN could reflect a more general role in coordinating attentional resources, which are essential for various cognitive functions, including memory formation (*Menon and Uddin, 2010*; *Uddin, 2015*). To disentangle the specific contributions of memory recall and attention, future studies should incorporate carefully designed control tasks that do not involve memory components. It is also important to note that successful memory recall likely involves the coordinated activity of multiple brain systems beyond the triple network model investigated here. For instance, the medial temporal lobe, including the hippocampus and adjacent cortical regions, plays a crucial role in episodic memory formation and retrieval (*Burgess et al., 2002*; *Moscovitch et al., 2016*). Future studies will need to investigate a broader set of brain areas during successful and unsuccessful memory trials to gain a more comprehensive understanding of the neural circuits supporting distinctions between successfully recalled and forgotten memory trials.

## Externally triggered vs. internally driven memory processes

Our results reveal a consistent pattern of directed information flow from the AI to both the DMN and FPN, persisting across externally triggered encoding and internally driven free recall. This pattern underscores the AI's robust and versatile role in modulating large-scale brain networks across diverse task contexts, aligning with the triple network model's conceptualization of the AI as a critical hub for attentional and cognitive control (*Menon, 2011*; *Menon, 2023*). However, the persistence of AI-driven information flow during internally triggered free recall was unexpected, given the view of the DMN's dominance in internal cognition. This reproducible pattern, observed across both externally and

internally driven tasks in all four experiments, reinforces the AI's crucial role in orchestrating network dynamics over extended time periods.

We did not detect an opposing pattern of greater directed influences from the DMN during recall, as might be expected given the internally driven nature of free recall. Several factors may contribute to this unexpected result. First, in the three verbal recall tasks, our PTE analysis was time-locked to word production onset, which may not capture the dynamics of network interactions during recall, particularly in the early retrieval initiation stage whose precise onset is unknown. This limitation is especially relevant for understanding the DMN's role, which might be more prominent in the initiation of recall rather than the selection of verbal output. Secondly, the PTE method requires relatively long time series for robust estimation of information flow. The brief windows associated with the initiation of individual recall events may not provide sufficient data for detecting subtle shifts in network dynamics, potentially masking transient increases in DMN influence.

Moreover, the consistent AI-driven information flow during recall might reflect the SN's ongoing role in monitoring and evaluating retrieved information, even during internally driven processes. This interpretation aligns with Sestieri and colleagues' observation of sustained SN activity across all phases of memory search tasks (*Sestieri et al., 2014*) and suggests a more complex view of the AI's function in both externally driven and internal cognitive processes.

Intriguingly, as noted above, while we observed PCC/precuneus suppression during encoding and enhancement during recall, the AI maintained its directed influence on this DMN node during encoding and recall. This apparent discrepancy between local activity (suppression) and network-level communication highlights the complex nature of brain network dynamics. It is likely that PTE-based network interactions examined in this study at the time scale of about 2 s miss subtle changes in directed interactions that occur during internally driven initiation of memory recall. Furthermore, our directed connectivity analysis used broadband signals (0.5–80 Hz), while power analysis of local neuronal activity focused on the high-gamma band (80–160 Hz). These different frequency ranges may capture distinct aspects of neural processing, with broadband connectivity reflecting more general, sustained network interactions.

To further elucidate these dynamics, future studies should consider employing techniques that can capture rapid changes in directed network interactions, investigating the temporal evolution of network interactions leading up to and following recall events, exploring the relationship between different frequency bands in connectivity and local activity measures, and developing methods to better estimate the onset of internal retrieval processes in verbal tasks. These approaches could provide valuable insights into the transition between externally driven and internally driven processes and offer a more precise understanding of the AI and PCC/precuneus's differential roles in coordinating network dynamics across different memory phases.

## AI as an outflow hub and a novel perspective on theoretical models of memory

Beyond information flow along individual pathways linking the AI with the DMN and FPN, our PTE analysis further revealed that the AI is an outflow hub in its interactions with the DMN and the FPN regardless of stimulus materials. As a central node of the SN (*Menon and Uddin, 2010*; *Seeley et al., 2007*; *Sridharan et al., 2008*), the AI is known to play a crucial role in influencing other networks (*Menon and Uddin, 2010*; *Uddin, 2015*). Our results align with findings based on control theory analysis of brain networks during a working memory task. Specifically, *Cai et al., 2021* found higher causal outflow and controllability associated with the AI compared to DMN and FPN nodes during an n-back working memory task. Controllability refers to the ability to perturb a system from a given initial state to other configuration states in finite time by means of external control inputs. Intuitively, nodes with higher controllability require lower energy for perturbing a system from a given state, making controllability measures useful for identifying driver nodes with the potential to influence overall state dynamics. By virtue of its higher controllability relative to other brain areas, the AI is well-positioned to dynamically engage and disengage with other brain areas. These findings expand our understanding of the AI's role, extending beyond attention and working memory tasks to incorporate two distinct stages of episodic memory formation. Our study, leveraging the temporal precision of iEEG data, substantially enhances previous fMRI findings by unveiling the neurophysiological

mechanisms underlying the AI's dynamic regulation of network activity during memory formation and cognition more generally.

Our findings bring a novel perspective to the seminal model of human memory proposed by *Atkinson and Shiffrin, 1968*. This model conceptualizes memory as a multistage process, with control mechanisms regulating the transition of information across these stages. The observed suppression of high-gamma power in the PCC/precuneus and enhancement in the AI during the encoding phase may be seen as one neurophysiological manifestation of these control processes. The AI's role as a dynamic switch, modulating activity between the DMN and FPN, aligns with active processing and control needed to encode sensory information into short-term memory. On the other hand, the transformations observed during the recall phase, particularly the discernible lack of DMN suppression patterns, may correspond to the retrieval processes where internally generated cues steer the reactivation of memory representations during recall. These results provide a novel neurophysiological model for understanding the complex control processes underpinning human memory functioning.

## Limitations and future work

Our study, while revealing important insights into network dynamics during memory processes, has several limitations that provide avenues for further investigation. Although our computational methods suggest directed influences, direct causal manipulations, such as targeted brain stimulation during memory tasks, are needed to establish definitive causal relationships between network nodes. The PTE method, while powerful, cannot reliably capture rapid shifts in network dynamics. Subsequent research should employ techniques with higher temporal precision to map these changes.

To determine whether our observed network dynamics are memory-specific or reflect more general cognitive processes, additional work should compare directed connectivity patterns across memory and non-memory tasks. Our analysis approach, necessitated by limited multi-task participation, precluded robust within-subject analyses. Future studies should aim for more consistent multi-task participation to enable individual-level analyses of network dynamics across tasks.

In the free recall verbal tasks, precisely timing the onset of internal retrieval processes remains challenging. Experimental designs with cued recall similar to the WMSM task could provide crucial insights into early stages of memory retrieval. This approach could help clarify the roles of different networks, especially the DMN, during the initiation of recall versus the execution of verbal output. The dissociation we observed between local activity and network-level communication warrants further investigation. Further studies are needed to determine the relationship between different frequency bands in connectivity and local activity measures to better understand how these distinct aspects of neural processing contribute to memory formation and retrieval.

Despite these limitations, our findings provide a robust foundation for investigations into the electrophysiological basis of large-scale brain network interactions during memory formation and recall. By addressing these limitations, subsequent studies can further refine our understanding of how these networks dynamically coordinate to support episodic memory and other cognitive functions. Such investigations may reveal a more dynamic interplay between the SN, DMN, and FPN, where their relative influences shift rapidly depending on the specific cognitive demands of the task.

## Conclusions

Our study provides novel insights into the neural dynamics underpinning episodic memory processes across four diverse memory experiments. We discovered that the AI, a key node of the SN, exerts a strong and consistent directed influence on both the DMN and FPN during memory encoding and recall. This finding extends the applicability of the triple network model to episodic memory processes in both verbal and spatial domains, highlighting the AI's crucial role as an outflow hub that modulates information flow within and between these cognitive networks.

Importantly, we observed a dissociation between local activity and network-level communication in the PCC/precuneus node of the DMN. The suppression of high-gamma power in this region during encoding, but not during recall, suggests a context-specific functional regulation that varies across memory phases. This finding reveals the intricate and dynamic interplay between local neural activity and large-scale network communication, and highlights the multifaceted nature of brain mechanisms underlying human memory processing.

The robust replicability of our findings across multiple memory tasks and modalities enhances the reliability and generalizability of our results, addressing a critical need in human intracranial EEG research. Our results reinforce the concept that memory operations rely on the concerted action of widely distributed brain networks (*Mesulam, 1990*), extending beyond traditional memory-specific regions.

By elucidating the electrophysiological basis of directed information flow within the triple network model, our study advances the understanding of neural circuit dynamics in human memory and cognition. Our findings provide a template for understanding the neural basis of memory impairments in neurological and psychiatric disorders. For instance, the disruption of these network interactions could contribute to memory deficits in conditions such as Alzheimer's disease, where dysfunctions in the SN, DMN, and FPN are now being increasingly documented (*Bonthius et al., 2005*; *Guzmán-Vélez et al., 2022*).

## Methods
### UPENN-RAM iEEG recordings

iEEG recordings from 249 patients shared by Kahana and colleagues at the University of Pennsylvania (UPENN) (obtained from the UPENN-RAM public data release) were used for analysis (*Jacobs et al., 2016*). Patients with pharmaco-resistant epilepsy underwent surgery for removal of their seizure onset zones. iEEG recordings of these patients were downloaded from a UPENN-RAM consortium-hosted data-sharing archive (http://memory.psych.upenn.edu/RAM). These data were recorded at eight hospitals: Thomas Jefferson University Hospital; University of Texas Southwestern Medical Center; Emory University Hospital; Dartmouth College Hospital; University of Pennsylvania Hospital; Mayo Clinic; National Institutes of Health; and Columbia University Hospital. Prior to data collection, research protocols and ethical guidelines were approved by the Institutional Review Board at the participating hospitals and informed consent was obtained from the participants and guardians (*Jacobs et al., 2016*).

Details of all the recording sessions and data pre-processing procedures are described by Kahana and colleagues (*Jacobs et al., 2016*). Briefly, iEEG recordings were obtained using subdural grids and strips (contacts placed 10 mm apart) or depth electrodes (contacts spaced 5–10 mm apart) using recording systems at each clinical site. iEEG systems included DeltaMed XlTek (Natus), Grass Telefactor, and Nihon-Kohden EEG systems. Electrodes located in brain lesions or those which corresponded to seizure onset zones or had significant interictal spiking or had broken leads were excluded from analysis.

Anatomical localization of electrode placement was accomplished by co-registering the postoperative computed CTs with the postoperative MRIs using FSL (FMRIB [Functional MRI of the Brain] Software Library), BET (Brain Extraction Tool), and FLIRT (FMRIB Linear Image Registration Tool) software packages. Preoperative MRIs were used when postoperative MRIs were not available. The resulting contact locations were mapped to MNI space using an indirect stereotactic technique and OsiriX Imaging Software DICOM viewer package.

We used the insula atlas by Faillenot and colleagues to demarcate the AI (*Faillenot et al., 2017*), downloaded from http://brain-development.org/brain-atlases/adult-brain-atlases/. This atlas is based on probabilistic analysis of the anatomy of the insula with demarcations of the AI based on three short dorsal gyri and the posterior insula (PI), which encompasses two long gyri. To visualize iEEG electrodes on the insula atlas, we used surface-rendering code (GitHub: https://github.com/ludovicbellier/InsulaWM; *Bellier, 2022*) provided by *Llorens et al., 2023*. We used the Brainnetome atlas (*Fan et al., 2016*) to demarcate the PCC/precuneus, the mPFC, the dPPC, and the MFG. The dorsal anterior cingulate cortex node of the SN was excluded from analysis due to lack of sufficient electrode placement. Out of 249 individuals, data from 177 individuals (aged from 16 to 64, mean age 36.3 ± 11.5, 91 females) were used for subsequent analysis based on electrode placement in the AI and the PCC/precuneus, mPFC, dPPC, and MFG.

Original sampling rates of iEEG signals were 500 Hz, 1000 Hz, 1024 Hz, and 1600 Hz. Hence, iEEG signals were downsampled to 500 Hz, if the original sampling rate was higher, for all subsequent analysis. The two major concerns when analyzing interactions between closely spaced intracranial electrodes are volume conduction and confounding interactions with the reference electrode (*Burke*

*et al., 2013*; *Frauscher et al., 2018*). Hence, bipolar referencing was used to eliminate confounding artifacts and improve the signal-to-noise ratio of the neural signals, consistent with previous studies using UPENN-RAM iEEG data (*Burke et al., 2013*; *Ezzyat et al., 2018*). Signals recorded at individual electrodes were converted to a bipolar montage by computing the difference in signal between adjacent electrode pairs on each strip, grid, and depth electrode and the resulting bipolar signals were treated as new 'virtual' electrodes originating from the midpoint between each contact pair, identical to procedures in previous studies using UPENN-RAM data (*Solomon et al., 2019*). Line noise (60 Hz) and its harmonics were removed from the bipolar signals using band-stop filters at 57–63 Hz, 117–123 Hz, and 177–183 Hz. Finally, each bipolar signal was Z-normalized by removing mean and scaling by the standard deviation. For filtering, we used a fourth-order two-way zero phase lag Butterworth filter throughout the analysis. iEEG signals were filtered in the broad frequency spectrum (0.5–80 Hz) as well as narrowband frequency spectra delta-theta (0.5–8 Hz), alpha (8–12 Hz), beta (12–30 Hz), gamma (30–80 Hz), and high-gamma (80–160 Hz).

## Episodic memory experiments

### VFR task

Patients performed multiple trials of a VFR experiment, where they were presented with a list of words and subsequently asked to recall as many as possible from the original list (*Figure 1a*; *Solomon et al., 2017*; *Solomon et al., 2019*). The task consisted of three periods: encoding, delay, and recall. During encoding, a list of 12 words was visually presented for ~30 s. Words were selected at random, without replacement, from a pool of high-frequency English nouns (http://memory.psych.upenn.edu/Word_Pools). Each word was presented for a duration of 1600 ms, followed by an inter-stimulus interval of 800–1200 ms. After the encoding period, participants engaged in a math distractor task (the delay period in *Figure 1a*), where they were instructed to solve a series of arithmetic problems in the form of $a + b + c = ??$, where $a$, $b$, and $c$ were randomly selected integers ranging from 1 to 9. Mean accuracy across patients in the math task was 90.87% ± 7.22%, indicating that participants performed the math task with a high level of accuracy, similar to our previous studies (*Das and Menon, 2022a*). After a 20 s post-encoding delay, participants were instructed to recall as many words as possible during the 30 s recall period. Average recall accuracy across patients was 25.0% ± 10.6%, similar to prior studies of verbal episodic memory retrieval in neurosurgical patients (*Burke et al., 2014*). We analyzed iEEG epochs from the encoding and recall periods of the VFR task. For the recall periods, iEEG recordings 1600 ms prior to the vocal onset of each word were analyzed (*Solomon et al., 2019*). Data from each trial was analyzed separately and specific measures were averaged across trials.

### CATVFR task

This task was very similar to the VFR task. Here, patients performed multiple trials of a categorized free recall experiment, where they were presented with a list of words with consecutive pairs of words from a specific category (e.g., JEANS-COAT, GRAPE-PEACH, etc.) and subsequently asked to recall as many as possible from the original list (*Figure 1b*; *Qasim et al., 2023*). Similar to the uncategorized VFR task, this task also consisted of three periods: encoding, delay, and recall. During encoding, a list of 12 words was visually presented for ~30 s. Semantic categories were chosen using Amazon Mechanical Turk. Pairs of words from the same semantic category were never presented consecutively. Each word was presented for a duration of 1600 ms, followed by an inter-stimulus interval of 750–1000 ms. After a 20 s post-encoding delay (math) similar to the uncategorized VFR task, participants were instructed to recall as many words as possible during the 30 s recall period. Average accuracy across patients in the math task was 89.46% ± 9.90%. Average recall accuracy across patients was 29.6% ± 13.4%. Analysis of iEEG epochs from the encoding and recall periods of the categorized free recall task was same as the uncategorized VFR task.

### PALVCR task

Patients performed multiple trials of a PALVCR experiment, where they were presented with a list of word-pairs and subsequently asked to recall based on the given word-cue (*Figure 1c*). Similar to the uncategorized VFR task, this task also consisted of three periods: encoding, delay, and recall. During encoding, a list of six word-pairs was visually presented for ~36 s. Similar to the uncategorized VFR task, words were selected at random, without replacement, from a pool of high-frequency English

nouns (http://memory.psych.upenn.edu/Word_Pools). Each word was presented for a duration of 4000 ms, followed by an inter-stimulus interval of 1750–2000 ms. After a 20 s post-encoding delay (math) similar to the uncategorized VFR task, participants were shown a specific word-cue for a duration of 4000 ms and asked to verbally recall the cued word from memory. Each word presentation during recall was followed by an inter-stimulus interval of 1750–2000 ms and the recall period lasted for ~36 s. Average accuracy across patients in the math task was 93.91% ± 4.66%. Average recall accuracy across patients was 33.8% ± 25.9%. For encoding, iEEG recordings corresponding to the 4000 ms encoding period of the task were analyzed. For recall, iEEG recordings 1600 ms prior to the vocal onset of each word were analyzed (*Solomon et al., 2019*). Data from each trial was analyzed separately and specific measures were averaged across trials.

### WMSM task

Patients performed multiple trials of a spatial memory experiment in a virtual navigation paradigm (*Goyal et al., 2018*; *Jacobs et al., 2016*; *Lee et al., 2018*) similar to the Morris water maze (*Morris, 1984*). The environment was rectangular (1.8:1 aspect ratio) and was surrounded by a continuous boundary (*Figure 1d*). There were four distal visual cues (landmarks), one centered on each side of the rectangle, to aid with orienting. Each trial (96 trials per session, 1–3 sessions per subject) started with two 5 s encoding periods, during which subjects were driven to an object from a random starting location. At the beginning of an encoding period, the object appeared and, over the course of 5 s, the subject was automatically driven directly toward it. The 5 s period consisted of three intervals: first, the subject was rotated toward the object (1 s); second, the subject was driven toward the object (3 s); and, finally, the subject paused while at the object location (1 s). After a 5 s delay with a blank screen, the same process was repeated from a different starting location. After both encoding periods for each item, there was a 5 s pause followed by the recall period. The subject was placed in the environment at a random starting location with the object hidden and then asked to freely navigate using a joystick to the location where they thought the object was located. When they reached their chosen location, they pressed a button to record their response. They then received feedback on their performance via an overhead view of the environment showing the actual and reported object locations. Average recall accuracy across patients was 48.1% ± 5.6%.

We analyzed the 5 s iEEG epochs corresponding to the entire encoding and recall periods of the task as has been done previously (*Goyal et al., 2018*; *Jacobs et al., 2016*; *Lee et al., 2018*). Data from each trial was analyzed separately and specific measures were averaged across trials, similar to the verbal tasks.

Out of total 177 participants, 51% (91 out of 177) of participants participated in at least two experiments, 17% (30 out of 177) of participants participated in at least three experiments, and 6% (10 out of 177) of participants participated in all four experiments.

## iEEG analysis of high-gamma power

We first filtered the signals in the high-gamma (80–160 Hz) frequency band (*Canolty et al., 2006*; *Helfrich and Knight, 2016*; *Miller et al., 2009*) using sequential band-pass filters in increments of 10 Hz (i.e., 80–90 Hz, 90–100 Hz, etc.), using a fourth-order two-way zero phase lag Butterworth filter. We used these narrowband filtering processing steps to correct for the 1/f decay of power. We then calculated the amplitude (envelope) of each narrow band signal by taking the absolute value of the analytic signal obtained from the Hilbert transform (*Foster et al., 2015*). Each narrow band amplitude time series was then normalized to its own mean amplitude, expressed as a percentage of the mean. Finally, we calculated the mean of the normalized narrow band amplitude time series, producing a single-amplitude time series. Signals were then smoothed using 0.2 s windows with 90% overlap (*Kwon et al., 2021*) and normalized with respect to 0.2 s pre-stimulus periods by subtracting the pre-stimulus baseline from the post-stimulus signal.

## iEEG analysis of PTE

PTE is a nonlinear measure of the directionality of information flow between time series and can be applied to nonstationary time series (*Das and Menon, 2021*; *Lobier et al., 2014*). Note that the information flow described here relates to signaling between brain areas and does not necessarily reflect the representation or coding of behaviorally relevant variables per se. The PTE measure is in contrast

to the Granger causality measure, which can be applied only to stationary time series (**Barnett and Seth, 2014**). We first carried out a stationarity test of the iEEG recordings (unit root test for stationarity [**Barnett and Seth, 2014**]) and found that the spectral radius of the autoregressive model is very close to one, indicating that the iEEG time series is nonstationary. This precluded the applicability of the Granger causality analysis in our study.

Given two time series $\{x_i\}$ and $\{y_i\}$, where $i = 1, 2, ..., M$, instantaneous phases were first extracted using the Hilbert transform. Let $\{x_i^p\}$ and $\{y_i^p\}$, where $i = 1, 2, ..., M$, denote the corresponding phase time series. If the uncertainty of the target signal $\{y_i^p\}$ at delay $\tau$ is quantified using Shannon entropy, then the PTE from driver signal $\{x_i^p\}$ to target signal $\{y_i^p\}$ can be given by

$$PTE_{x \to y} = \sum_i p\left(y_{i+\tau}^p, y_i^p, x_i^p\right) \log \left( \frac{p\left(y_{i+\tau}^p | y_i^p, x_i^p\right)}{p\left(y_{i+\tau}^p | y_i^p\right)} \right), \tag{1}$$

where the probabilities can be calculated by building histograms of occurrences of singles, pairs, or triplets of instantaneous phase estimates from the phase time series (**Hillebrand et al., 2016**). For our analysis, the number of bins in the histograms was set as $3.49 \times STD \times M^{-1/3}$ and delay $\tau$ was set as $2M/M_\pm$, where $STD$ is average standard deviation of the phase time series $\{x_i^p\}$ and $\{y_i^p\}$ and $M_\pm$ is the number of times the phase changes sign across time and channels (**Hillebrand et al., 2016**). PTE has been shown to be robust against the choice of the delay $\tau$ and the number of bins for forming the histograms (**Hillebrand et al., 2016**). In our analysis, PTE was calculated for the entire encoding and recall periods for each trial and then averaged across trials.

Net outflow was calculated as the difference between the total outgoing information and total incoming information, that is, net outflow = PTE(out) – PTE(in). For example, for calculation of PTE(out) and PTE(in) for the AI electrodes, electrodes in the PCC/precuneus, mPFC, dPPC, and MFG were considered, that is, PTE(out) was calculated as the net PTE from AI electrodes to the PCC/precuneus, mPFC, dPPC, and MFG electrodes, and PTE(in) was calculated as the net PTE from the PCC/precuneus, mPFC, dPPC, and MFG electrodes to AI electrodes. Net outflow for the PCC/precuneus, mPFC, dPPC, and MFG electrodes was calculated similarly.

## iEEG analysis of PLV and phase synchronization

We used PLV to compute phase synchronization between two time series (**Lachaux et al., 1999**). We first calculated the instantaneous phases of the two signals by using the analytical signal approach based on the Hilbert transform (**Bruns, 2004**). Given time series $x(t), t = 1, 2, ..., M$, its complex-valued analytical signal $z(t)$ can be computed as

$$z(t) = x(t) + i\tilde{x}(t) = A_x(t) e^{\Phi_x(t)}, \tag{2}$$

where $i$ denotes the square root of minus one, $\tilde{x}(t)$ is the Hilbert transform of $x(t)$, and $A_x(t)$ and are the instantaneous amplitude and instantaneous phase respectively and can be given by

$$A_x(t) = \sqrt{[x(t)]^2 + [\tilde{x}(t)]^2} \quad \text{and} \quad \phi_x(t) = \arctan\frac{\tilde{x}(t)}{x(t)}. \tag{3}$$

The Hilbert transform of $x(t)$ was computed as

$$\tilde{x}(t) = \frac{1}{\pi} PV \int_{-x}^{\infty} \frac{x(\tau)}{t - \tau} d\tau, \tag{4}$$

where $PV$ denotes the Cauchy principal value. MATLAB function 'hilbert' was used to calculate the Hilbert transform in our analysis. Given two time series $x(t)$ and $y(t)$, where $t = 1, 2, ..., M$, the PLV (zero-lag) can be computed as

$$\text{PLV} \triangleq \left| E\left[ e^{i(\phi_x(t) - \phi_y(t))} \right] \right|, \tag{5}$$

where $\phi_y(t)$ is the instantaneous phase for time series $y(t)$, $|\cdot|$ denotes the absolute value operator, $E[\cdot]$ denotes the expectation operator with respect to time $t$, and $i$ denotes the square root of minus one. PLVs were then averaged across trials to estimate the final PLV for each pair of electrodes.

## Statistical analysis

Statistical analysis was conducted using mixed-effects analysis with the lmerTest package (*Kuznetsova et al., 2017*) implemented in R software (version 4.0.2, R Foundation for Statistical Computing). Because PTE data were not normally distributed, we used BestNormalize (*Peterson and Cavanaugh, 2020*), which contains a suite of transformation-estimating functions that can be used to optimally normalize data. The resulting normally distributed data were subjected to mixed-effects analysis with the following model: *PTE ~ Condition + (1|Subject)*, where *Condition* models the fixed effects (condition differences) and (1|*Subject*) models the random repeated measurements within the same participant, similar to prior iEEG studies (*Das and Menon, 2021*; *Hoy et al., 2021*; *Salamone et al., 2021*). Before running the mixed-effects model, PTE was first averaged across trials for each channel pair. ANOVA was used to test the significance of findings with FDR-corrections for multiple comparisons (p<0.05). Linear mixed-effects models were run for encoding and recall periods separately. Similar mixed-effects statistical analysis procedures were used for comparison of high-gamma power across task conditions, where the mixed-effects analysis was run on each of the 0.2 s windows.

For effect size estimation, we used Cohen's *d* statistics for pairwise comparisons. We used the *lme. dscore*() function in the *EMAtools* package in R for estimating Cohen's *d*.

## Bayesian replication analysis

We used replication BF (*Ly et al., 2019*; *Verhagen and Wagenmakers, 2014*) analysis to estimate the degree of replicability for the direction of information flow for each frequency and task condition and across task domains. Analysis was implemented in R software using the BayesFactor package (*Rouder et al., 2009*). Because PTE data were not normally distributed, as previously, we used BestNormalize (*Peterson and Cavanaugh, 2020*) to optimally normalize data. We calculated the replication BF for pairwise experiments. We compared the BF of the joint model *PTE(task1 + task2) ~ Condition + (1|Subject)* with the BF of individual model as *PTE(task1) ~ Condition + (1|Subject)*, where *task1* denotes the VFR (original) task and *task2* denotes the CATVFR, PALVCR, or WMSM (replication) conditions. We calculated the ratio *BF(task1 + task2)/BF(task1)*, which was used to quantify the degree of replicability. We determined whether the degree of replicability was higher than 3 as BF of at least three indicates evidence for replicability (*Jeffreys, 1998*). A BF of at least 100 is considered '*decisive*' for the degree of replication (*Jeffreys, 1998*). Same analysis procedures were used to estimate the degree of replicability for high-gamma power comparison of DMN and FPN electrodes with the AI electrodes across experiments.

## Acknowledgements

This research was supported by NIH grants NS086085 and MH126518. We are grateful to the members of the UPENN-RAM consortia for generously sharing their unique iEEG data. We thank Dr. Byeong-wook Lee for assistance with the figures. We acknowledge the computational resources and support provided by the Stanford Research Computing Center.

## Additional information

### Funding

| Funder | Grant reference number | Author |
| --- | --- | --- |
| National Institutes of Health | NS086085 | Vinod Menon |
| National Institutes of Health | MH126518 | Vinod Menon |

The funders had no role in study design, data collection and interpretation, or the decision to submit the work for publication.

## Author contributions
Anup Das, Conceptualization, Resources, Software, Formal analysis, Validation, Investigation, Visualization, Methodology, Writing - original draft, Writing - review and editing; Vinod Menon, Conceptualization, Supervision, Funding acquisition, Writing - original draft, Project administration, Writing - review and editing

## Author ORCIDs
Anup Das  https://orcid.org/0000-0002-8897-7021

## Ethics
Human subjects: iEEG recordings from 249 patients shared by Kahana and colleagues at the University of Pennsylvania (UPENN) (obtained from the UPENN-RAM public data release) were used for analysis (Jacobs et al., 2016). Patients with pharmaco-resistant epilepsy underwent surgery for removal of their seizure onset zones. iEEG recordings of these patients were downloaded from a UPENN-RAM consortium hosted data sharing archive (URL: http://memory.psych.upenn.edu/RAM). These data were recorded at eight hospitals: Thomas Jefferson University Hospital; University of Texas Southwestern Medical Center; Emory University Hospital; Dartmouth College Hospital; University of Pennsylvania Hospital; Mayo Clinic; National Institutes of Health; and Columbia University Hospital. Prior to data collection, research protocols and ethical guidelines were approved by the Institutional Review Board at the participating hospitals and informed consent was obtained from the participants and guardians (Jacobs et al., 2016).

Reviewer #1 (Public review): https://doi.org/10.7554/eLife.99018.4.sa1
Author response https://doi.org/10.7554/eLife.99018.4.sa2

---

# Additional files

## Supplementary files
• MDAR checklist

## Data availability
iEEG recordings used in the study can be downloaded from http://memory.psych.upenn.edu/RAM without any restrictions. Scripts used in this study can be downloaded from https://github.com/scsnl/Das_NeuroImage_2022 (*de los Angeles, 2022*) without any restrictions.

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

# Appendix 1

## Appendix figures

**(a) Experiment 1 — VFR**

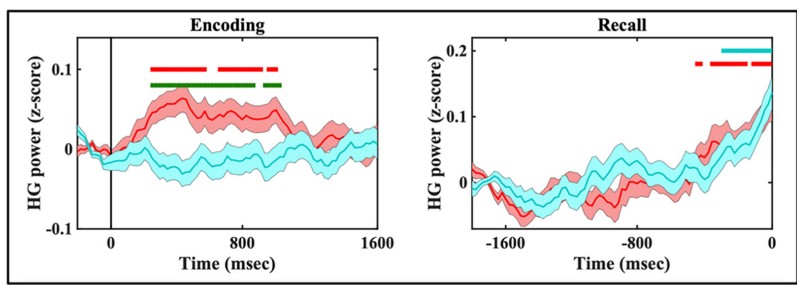

**(b) Experiment 2 — CATVFR**

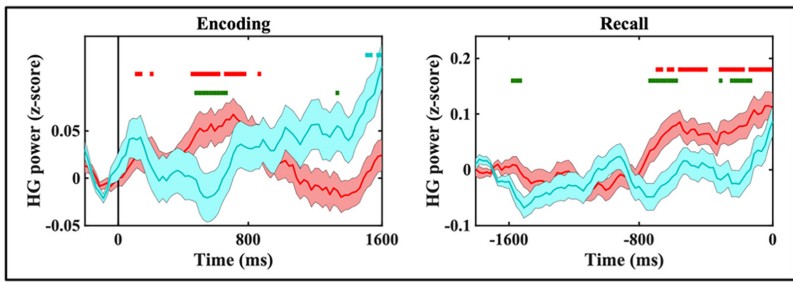

**(c) Experiment 3 — PALVCR**

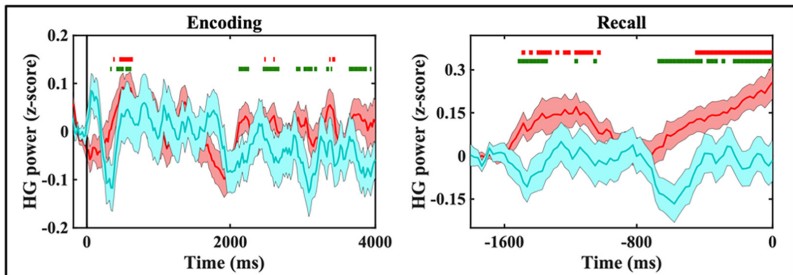

**(d) Experiment 4 — WMSM**

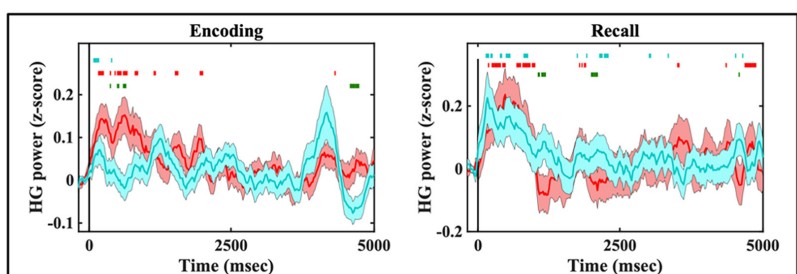

**Appendix 1—figure 1.** Intracranial electroencephalography (iEEG)-evoked response for anterior insula (AI) (red) and medial prefrontal cortex (mPFC) (cyan) in the four experiments. Green horizontal lines denote time periods where high-gamma power between the AI and mPFC was significantly different from each other. Red and cyan horizontal lines denote increase of high-gamma power compared to the resting baseline in the AI and mPFC, respectively.

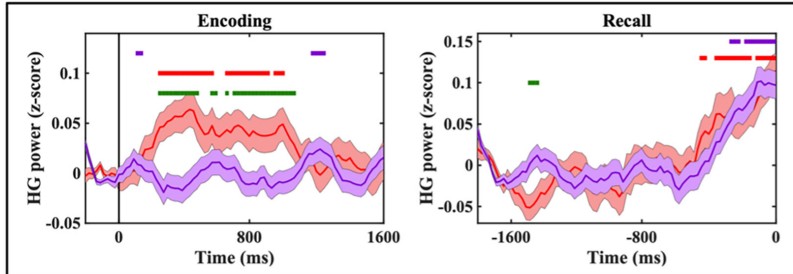

**(a) Experiment 1 — VFR**

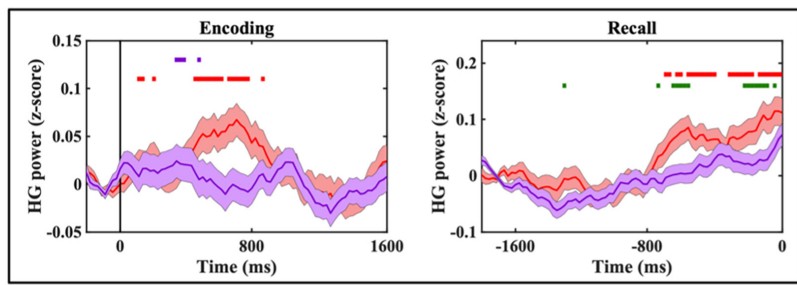

**(b) Experiment 2 — CATVFR**

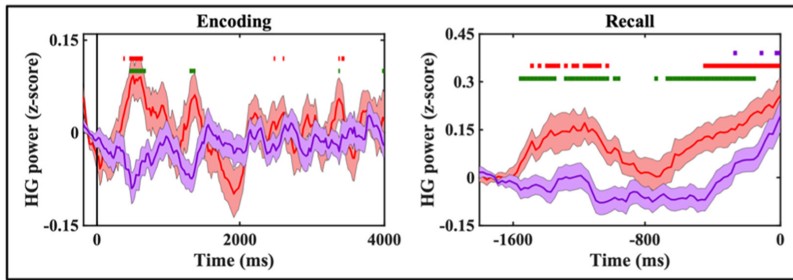

**(c) Experiment 3 — PALVCR**

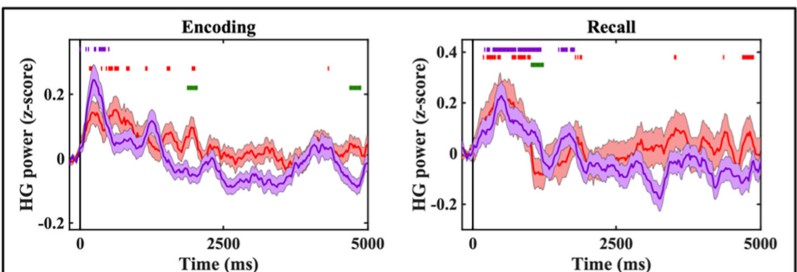

**(d) Experiment 4 — WMSM**

**Appendix 1—figure 2.** Intracranial electroencephalography (iEEG)-evoked response for anterior insula (AI) (red) and dorsal posterior parietal cortex (dPPC) (purple) in the four experiments. Green horizontal lines denote time periods where high-gamma power between the AI and dPPC was significantly different from each other. Red and purple horizontal lines denote increase of high-gamma power compared to the resting baseline in the AI and dPPC, respectively.

**(a) Experiment 1 — VFR**

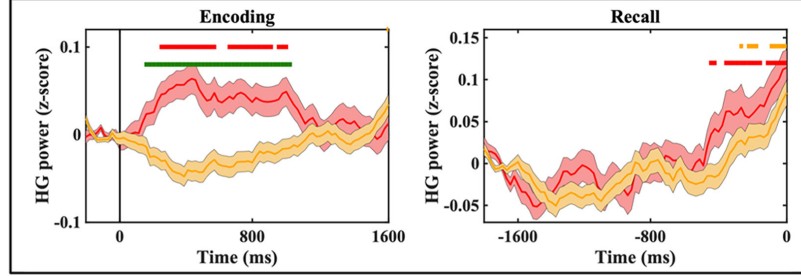

**(b) Experiment 2 — CATVFR**

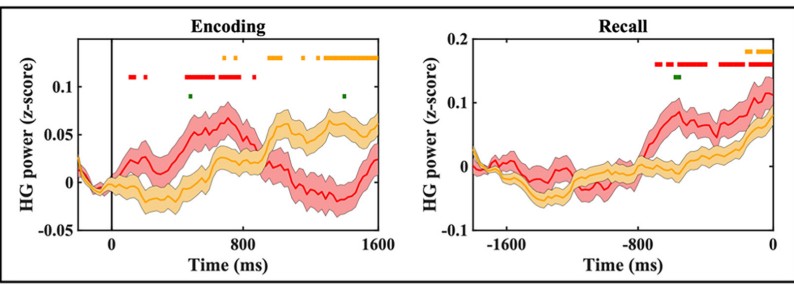

**(c) Experiment 3 — PALVCR**

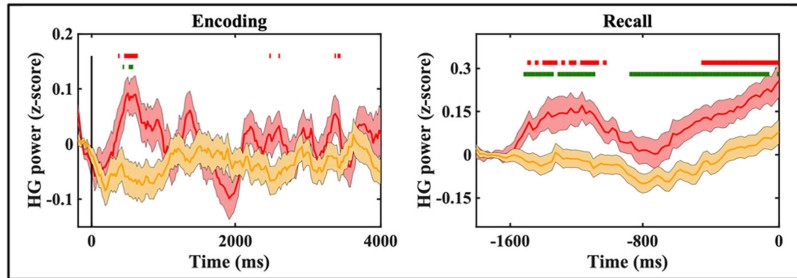

**(d) Experiment 4 — WMSM**

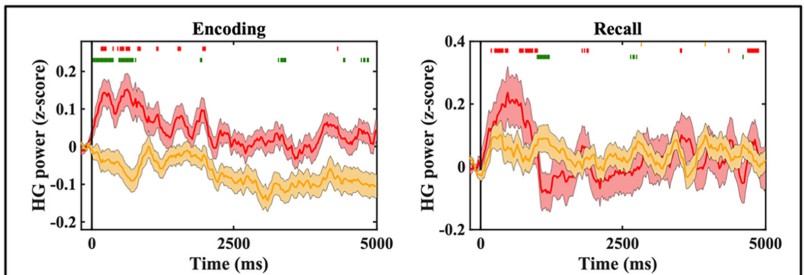

**Appendix 1—figure 3.** Intracranial electroencephalography (iEEG)-evoked response for anterior insula (AI) (red) and middle frontal gyrus (MFG) (orange) in the four experiments. Green horizontal lines denote time periods where high-gamma power between the AI and MFG was significantly different from each other. Red and orange horizontal lines denote increase of high-gamma power compared to the resting baseline in the AI and MFG, respectively.

**(a) Experiment 1 — VFR**

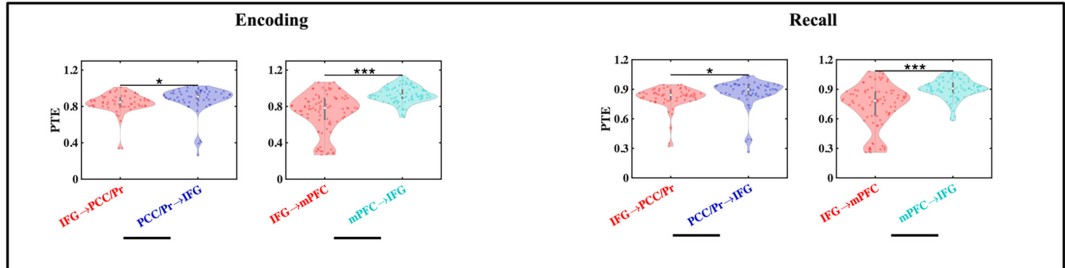

**(b) Experiment 2 — CATVFR**

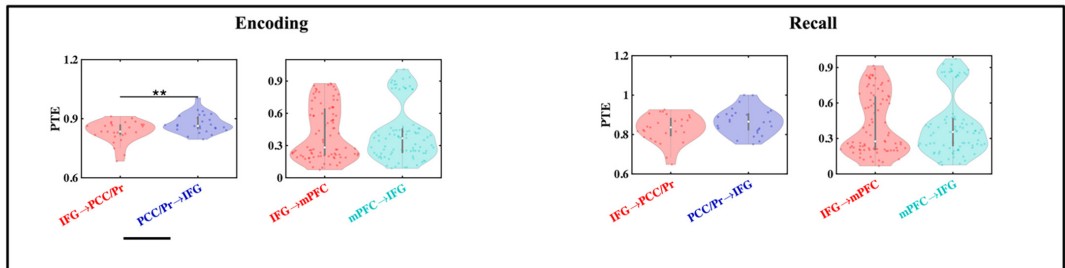

**(c) Experiment 3 — PALVCR**

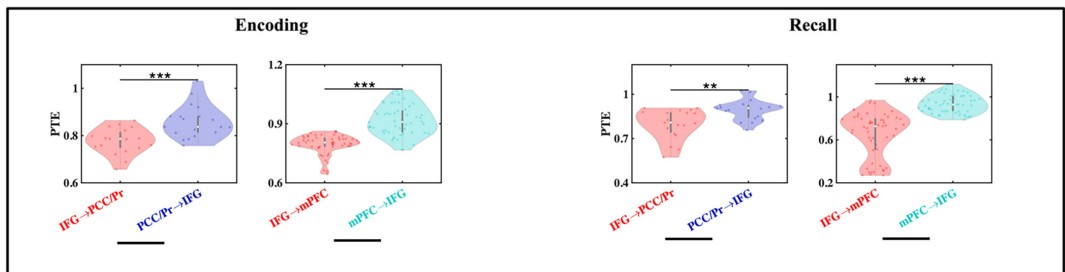

**(d) Experiment 4 — WMSM**

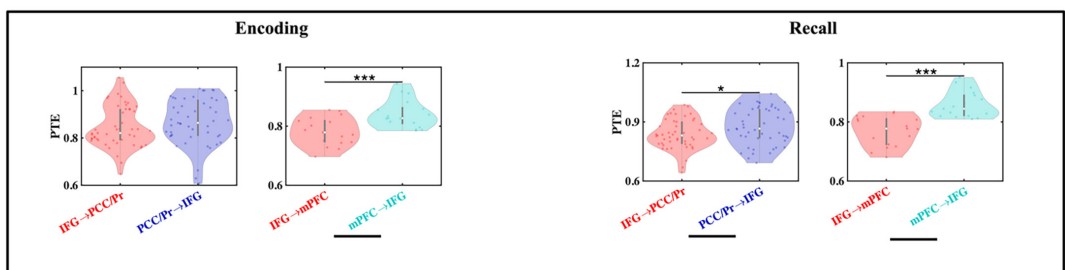

**Appendix 1—figure 4.** Directed information flow from the inferior frontal gyrus (IFG) to the default mode network (DMN) nodes and the reverse in broadband frequencies (0.5–80 Hz). \*\*\*p<0.001, \*\*p<0.01, \*p<0.05.

**(a) Experiment 1 — VFR**

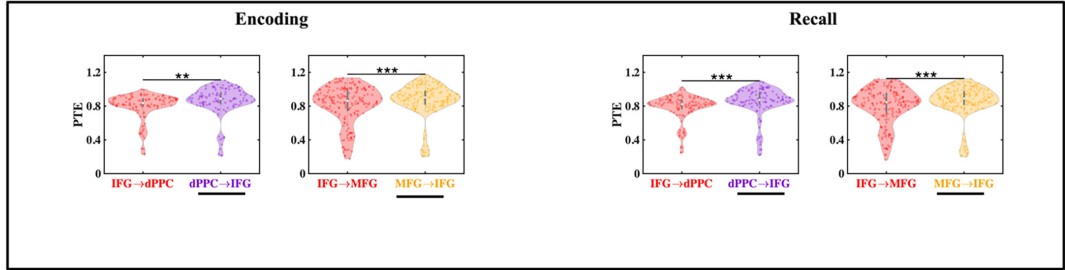

**(b) Experiment 2 — CATVFR**

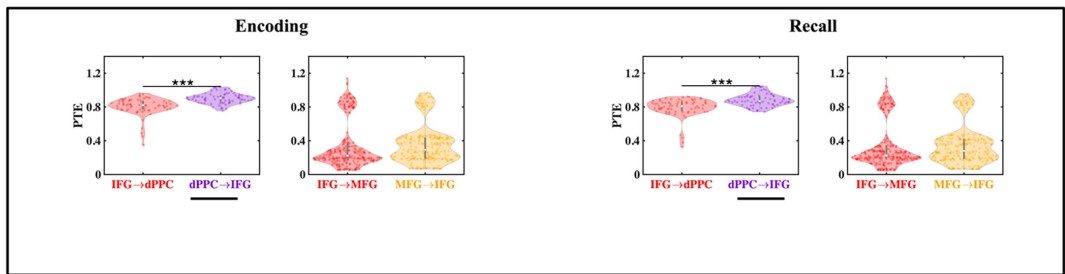

**(c) Experiment 3 — PALVCR**

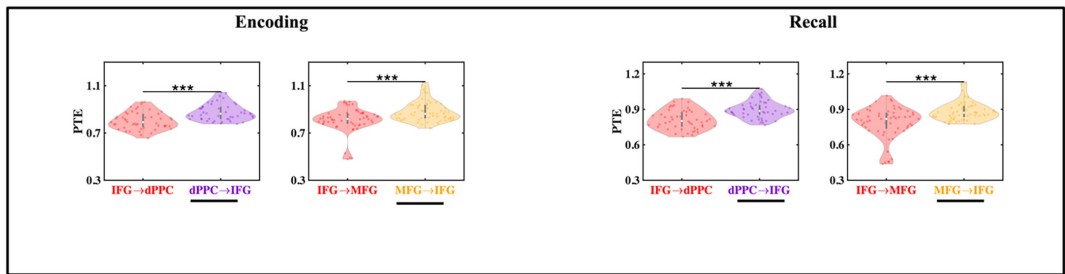

**(d) Experiment 4 — WMSM**

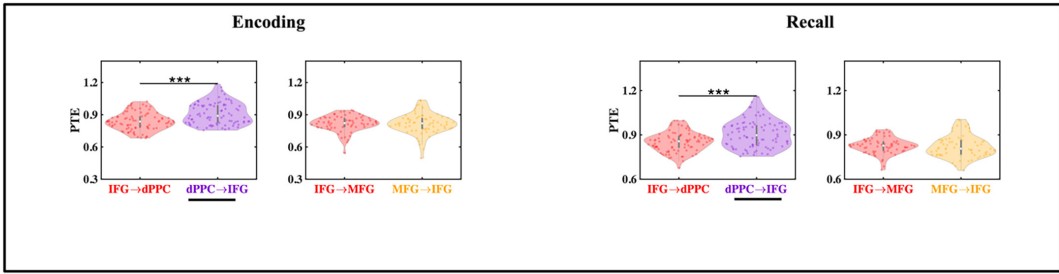

**Appendix 1—figure 5.** Directed information flow from the inferior frontal gyrus (IFG) to the frontoparietal network (FPN) nodes and the reverse in broadband frequencies (0.5–80 Hz). ***p<0.001, **p<0.01, *p<0.05.

**(a) Experiment 1 — VFR**

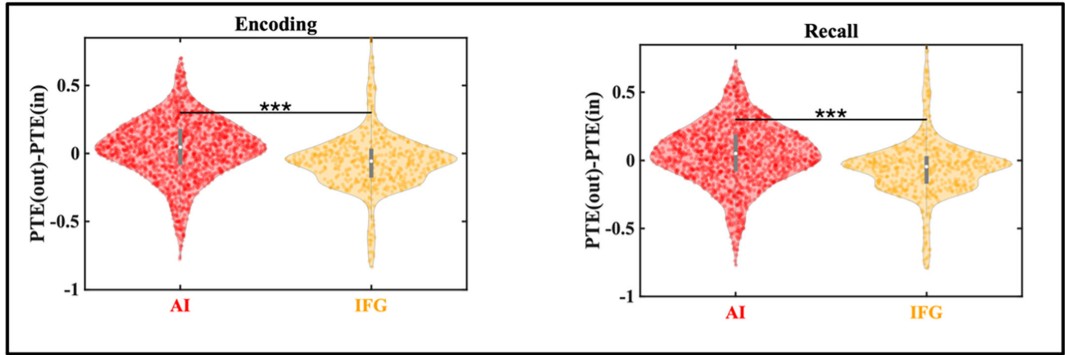

**(b) Experiment 2 — CATVFR**

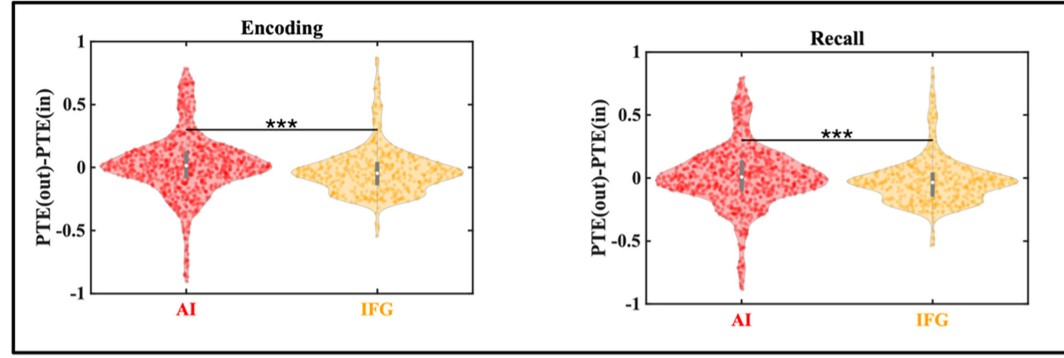

**(c) Experiment 3 — PALVCR**

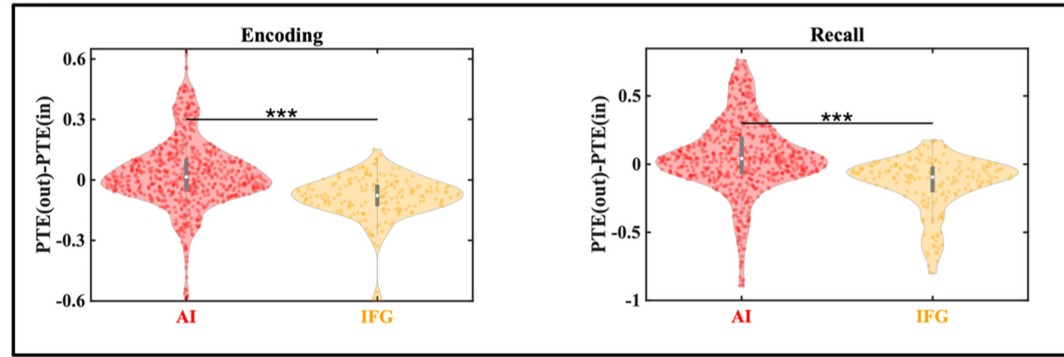

**(d) Experiment 4 — WMSM**

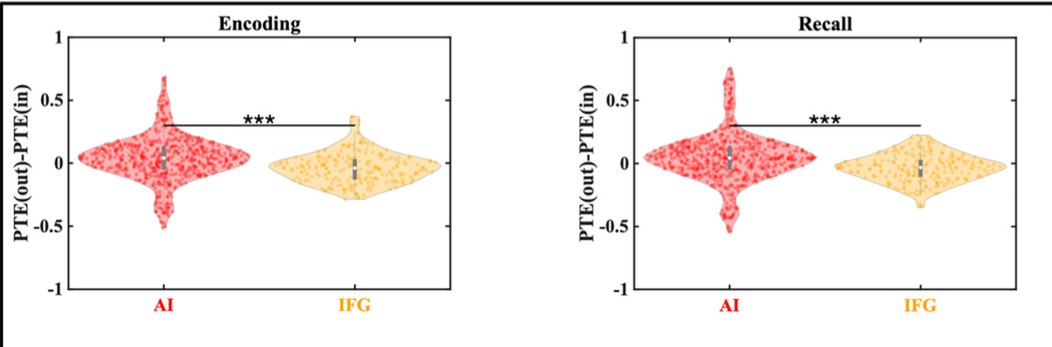

**Appendix 1—figure 6.** Comparison of net outflow for anterior insula (AI) and inferior frontal gyrus (IFG) in broadband frequencies (0.5–80 Hz). ***p<0.001.

**(a) Experiment 1 — VFR**

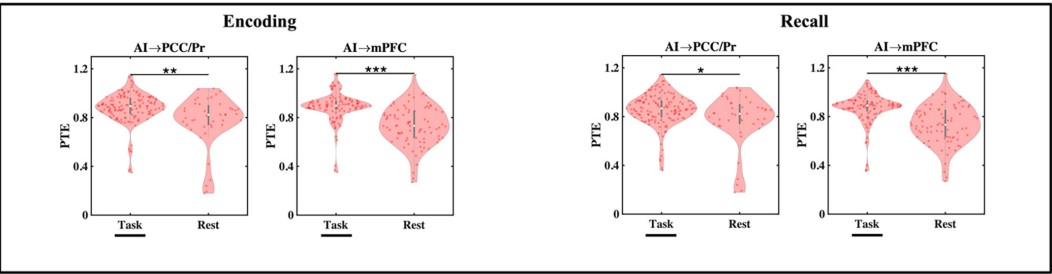

**(b) Experiment 2 — CATVFR**

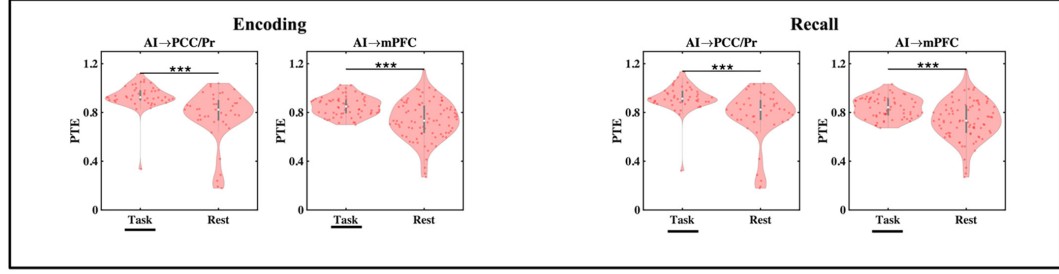

**(c) Experiment 3 — PALVCR**

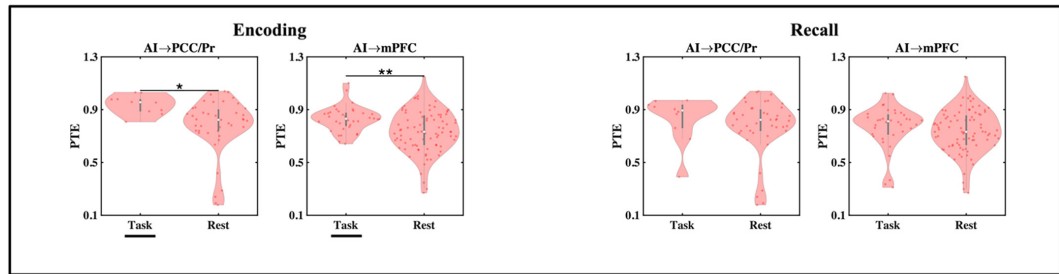

**(d) Experiment 4 — WMSM**

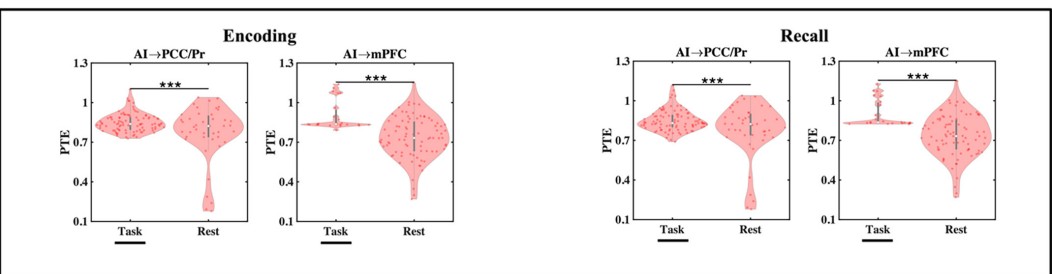

**Appendix 1—figure 7.** Differential directed information flow from the anterior insula to the default mode network (DMN) nodes during task versus resting-state in broadband frequencies (0.5–80 Hz). ***p<0.001, **p<0.01, *p<0.05.

**(a) Experiment 1 — VFR**

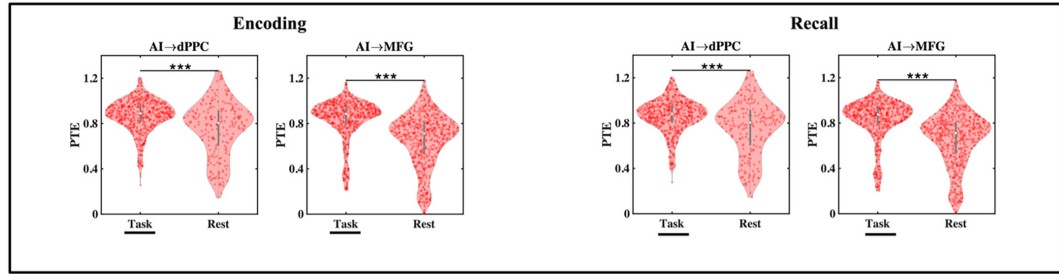

**(b) Experiment 2 — CATVFR**

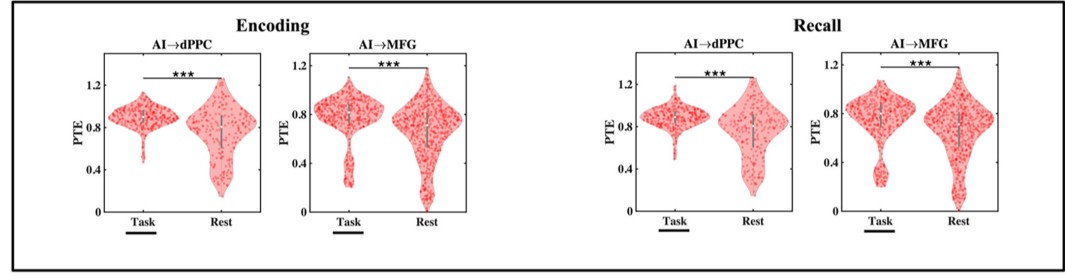

**(c) Experiment 3 — PALVCR**

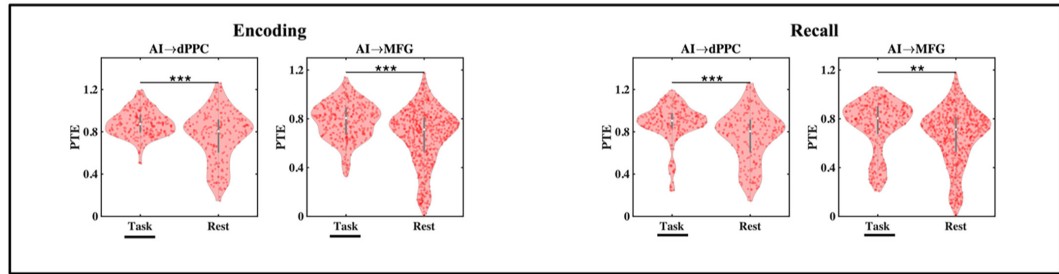

**(d) Experiment 4 — WMSM**

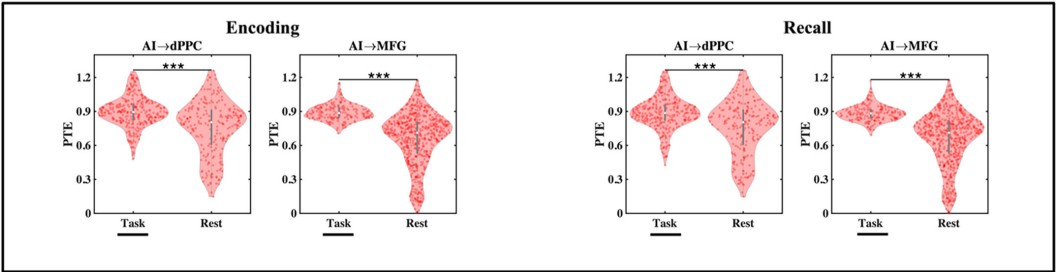

**Appendix 1—figure 8.** Differential directed information flow from the anterior insula to the FPN nodes during task versus resting-state, in broadband frequencies (0.5–80 Hz). *** p<0.001, ** p<0.01, * p<0.05.

**(a) Experiment 1 — VFR**

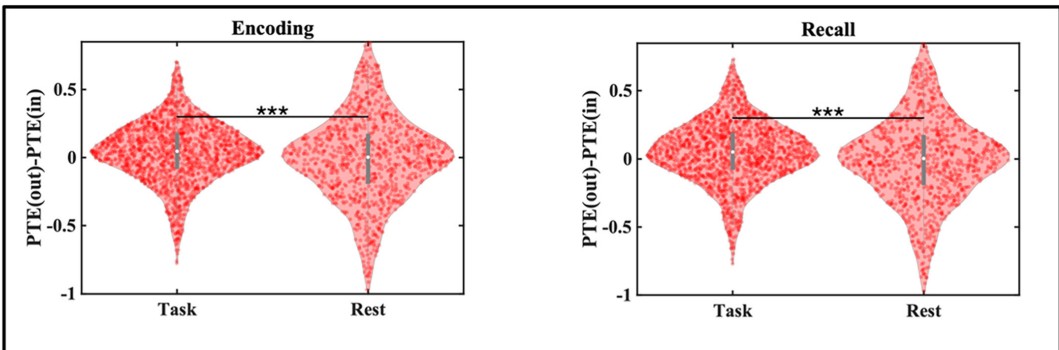

**(b) Experiment 2 — CATVFR**

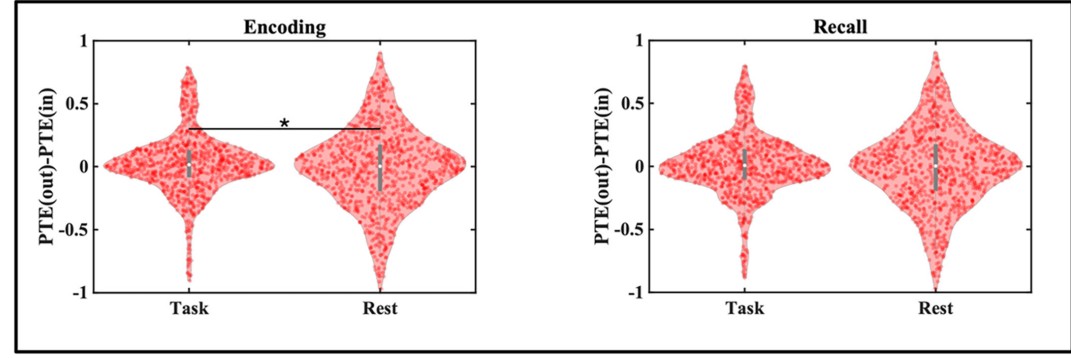

**(c) Experiment 3 — PALVCR**

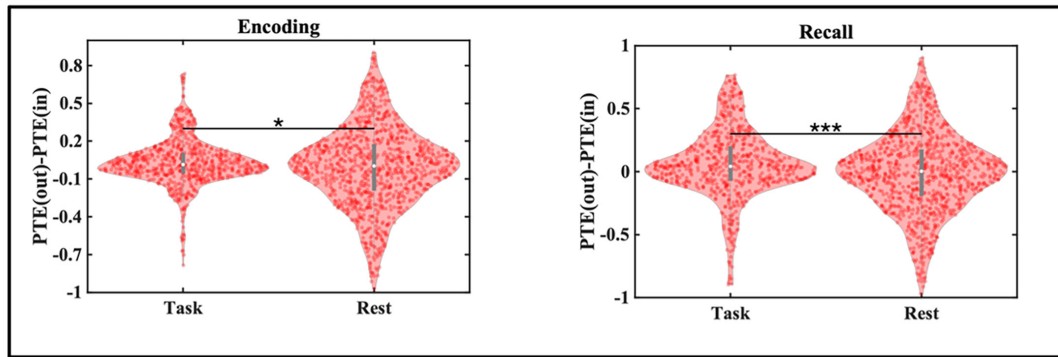

**(d) Experiment 4 — WMSM**

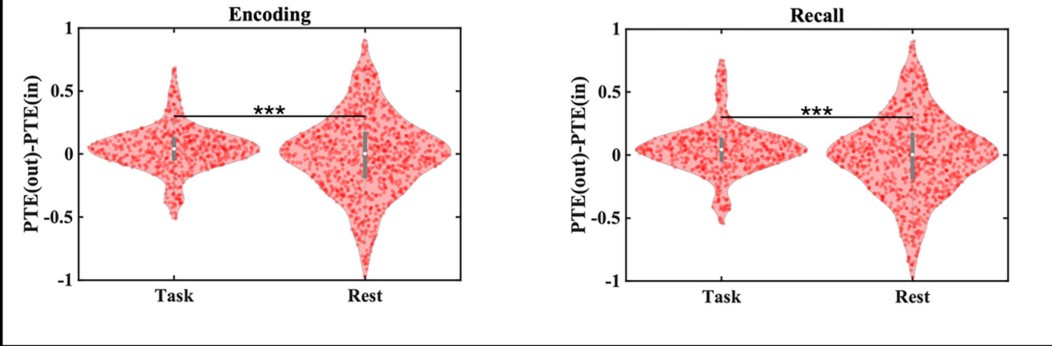

**Appendix 1—figure 9.** Comparison of net outflow for task versus resting state in anterior insula (AI) in broadband frequencies (0.5–80 Hz). ***p<0.001, *p<0.05.

**(a) Experiment 1 — VFR**

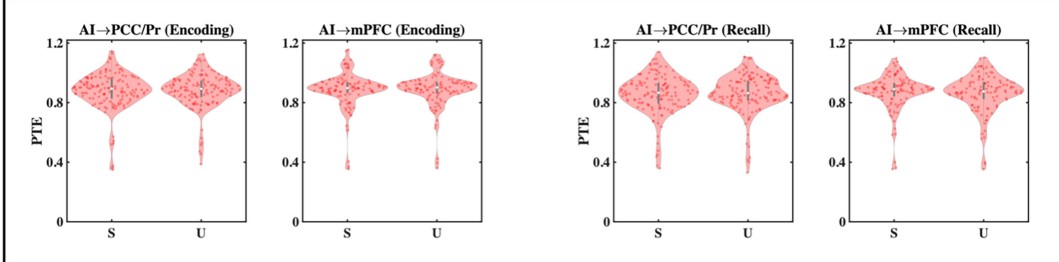

**(b) Experiment 2 — CATVFR**

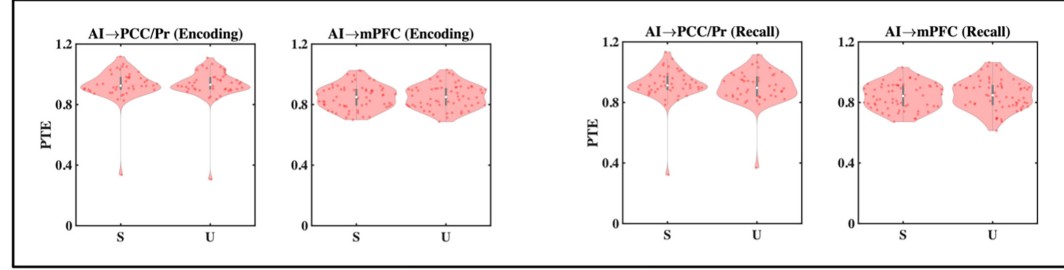

**(c) Experiment 3 — PALVCR**

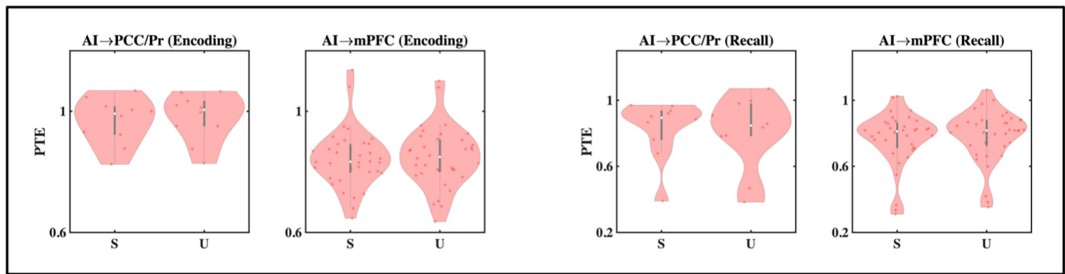

**(d) Experiment 4 — WMSM**

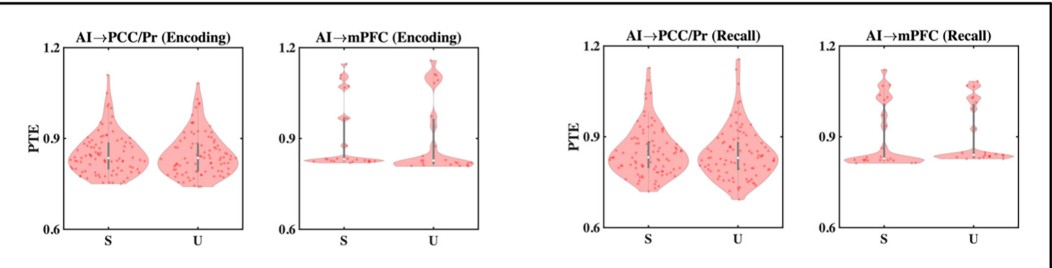

**Appendix 1—figure 10.** Directed information flow from the anterior insula to the default mode network (DMN) nodes during successfully (S) compared to unsuccessfully (U) recalled trials in broadband frequencies (0.5–80 Hz).

**(a) Experiment 1 — VFR**

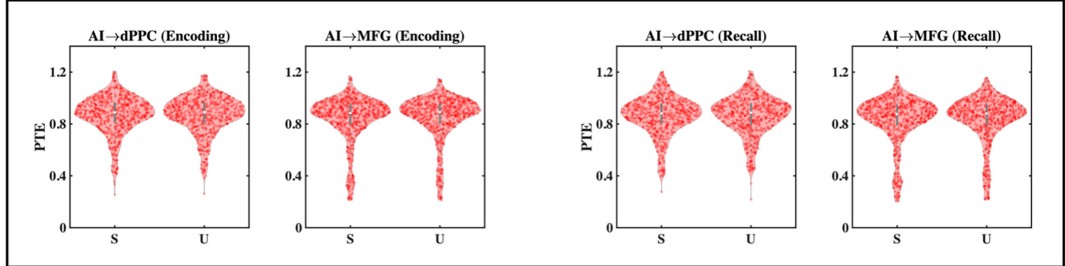

**(b) Experiment 2 — CATVFR**

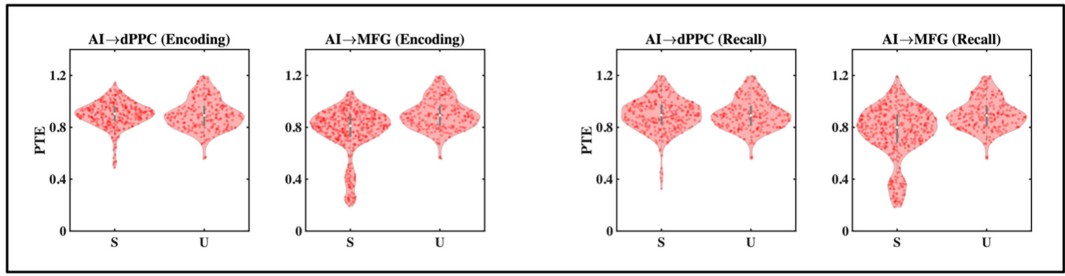

**(c) Experiment 3 — PALVCR**

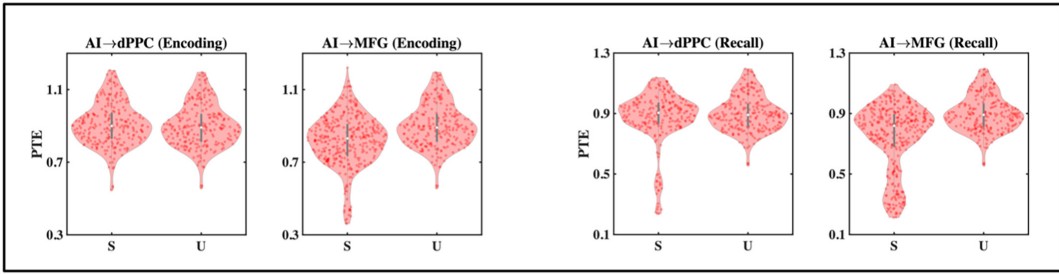

**(d) Experiment 4 — WMSM**

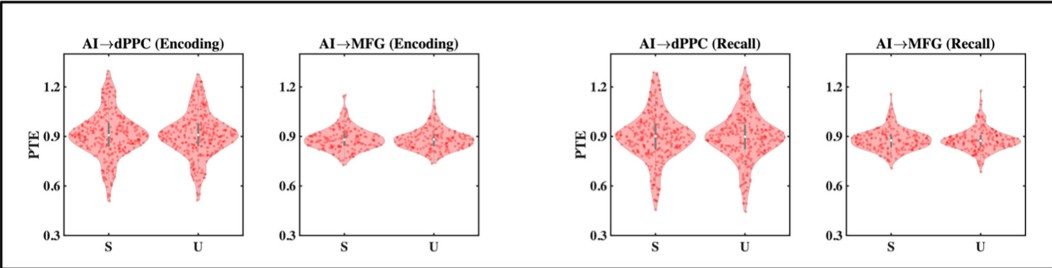

**Appendix 1—figure 11.** Directed information flow from the anterior insula to the frontoparietal network (FPN) nodes during successfully (S) compared to unsuccessfully (U) recalled trials in broadband frequencies (0.5–80 Hz).

**(a) Experiment 1 — VFR**

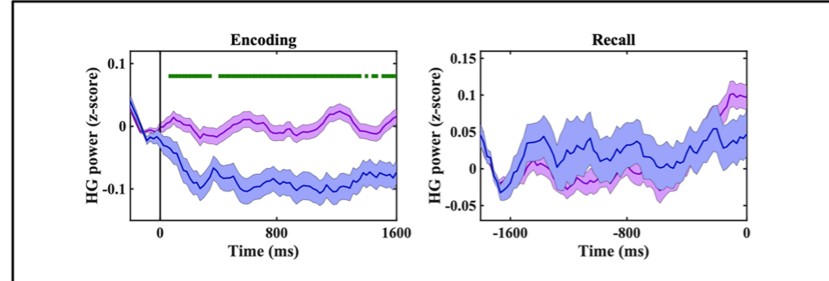

**(b) Experiment 2 — CATVFR**

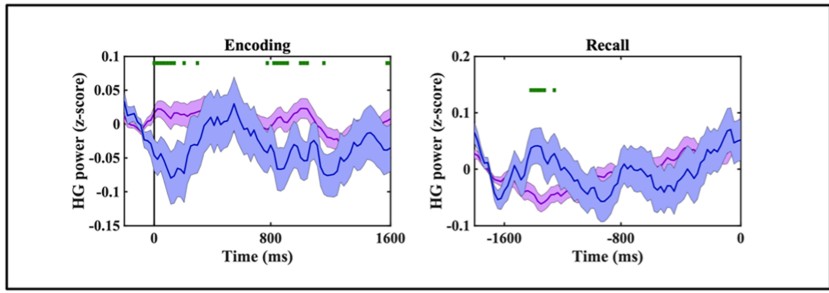

**(c) Experiment 3 — PALVCR**

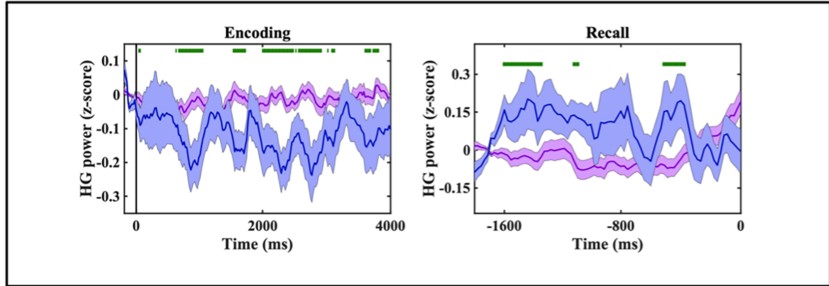

**(d) Experiment 4 — WMSM**

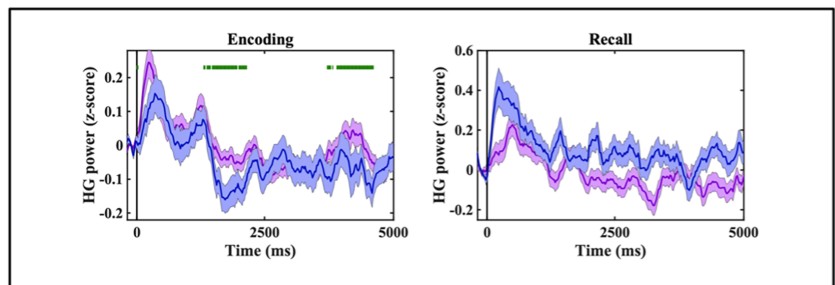

**Appendix 1—figure 12.** Intracranial electroencephalography (iEEG)-evoked response for posterior cingulate cortex (PCC)/precuneus (blue) and dorsal posterior parietal cortex (dPPC) (purple) in the four experiments. Green horizontal lines denote time periods where high-gamma power between the PCC/precuneus and dPPC was significantly different from each other.

**(a) Experiment 1 — VFR**

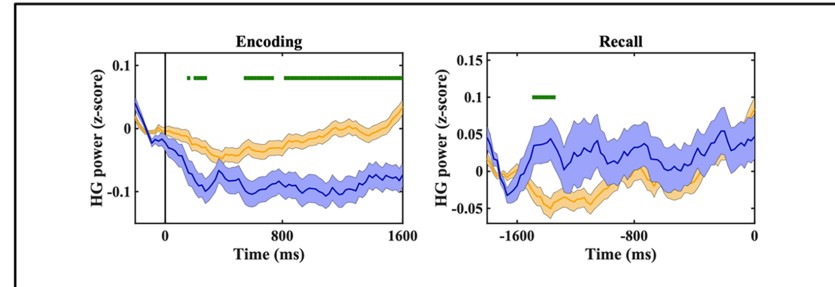

**(b) Experiment 2 — CATVFR**

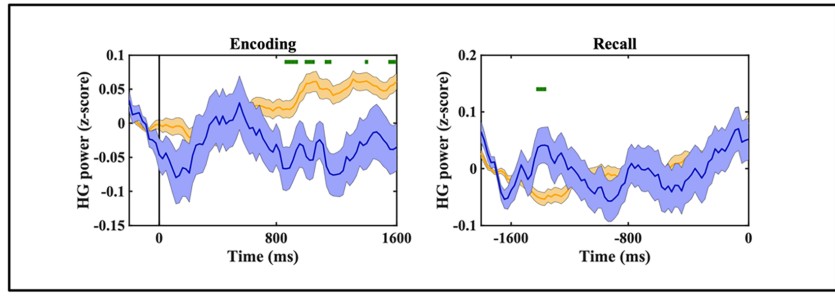

**(c) Experiment 3 — PALVCR**

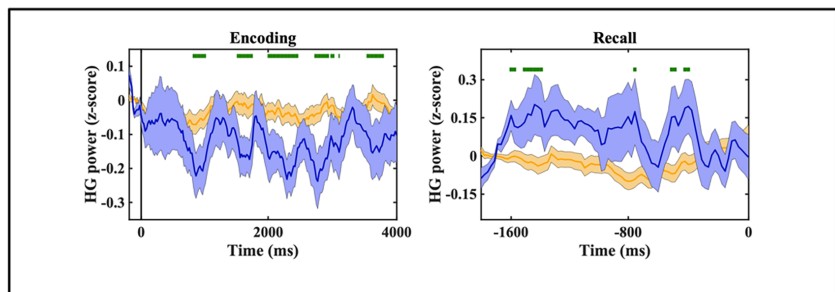

**(d) Experiment 4 — WMSM**

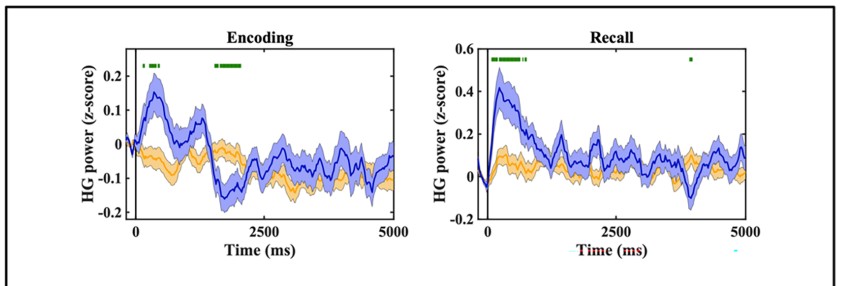

**Appendix 1—figure 13.** Intracranial electroencephalography (iEEG)-evoked response for posterior cingulate cortex (PCC)/precuneus (blue) and middle frontal gyrus (MFG) (orange) in the four experiments. Green horizontal lines denote time periods where high-gamma power between the PCC/precuneus and MFG was significantly different from each other.

## Appendix results

### Directed information flow from the AI to the DMN during encoding and recall in the CATVFR task

#### Encoding

Directed information flow from the AI to the PCC/precuneus was higher than the reverse during memory encoding ($F(1, 84) = 36.18$, p<0.001, Cohen's $d$ = 1.32) (*Figure 4b*).

#### Recall

Directed information flow from the AI to the PCC/precuneus ($F(1, 83) = 29.54$, p<0.001, Cohen's $d$ = 1.19) was higher than the reverse during memory recall (*Figure 4b*).

These results demonstrate that the AI has strong directed information flow to the PCC/precuneus node of the DMN during both the encoding and recall phases of the CATVFR episodic memory task.

### Directed information flow from the AI to the DMN during encoding and recall in the PALVCR task

#### Encoding

Directed information flow from the AI to the PCC/precuneus ($F(1, 17) = 22.19$, p<0.001, Cohen's $d$ = 2.28) was higher than the reverse (*Figure 4c*).

#### Recall

Directed information flow from the AI to the PCC/precuneus was higher than the reverse ($F(1, 17) = 6.45$, p<0.05, Cohen's $d$ = 1.23) (*Figure 4c*).

These results demonstrate that the AI has stronger directed information flow to the PCC/precuneus node of the DMN during both the encoding and recall phases of the PALVCR episodic memory task.

### Directed information flow from the AI to the DMN during encoding and recall in the WMSM task

#### Encoding

Directed information flow from the AI to the PCC/precuneus ($F(1, 176) = 51.14$, p<0.001, Cohen's $d$ = 1.08) and mPFC ($F(1, 41) = 44.53$, p<0.001, Cohen's $d$ = 2.08) was higher than the reverse (*Figure 4d*).

#### Recall

Directed information flow from the AI to the PCC/precuneus ($F(1, 177) = 36.86$, p<0.001, Cohen's $d$ = 0.91) and the mPFC ($F(1, 41) = 39.62$, p<0.001, Cohen's $d$ = 1.96) was also higher than the reverse (*Figure 4d*).

These results demonstrate that the AI has stronger directed information flow to the PCC/precuneus and mPFC nodes of the DMN during both the encoding and recall phases of the WMSM task.

### Directed information flow from AI to FPN nodes in the CATVFR task

We next examined directed information flow between the AI and FPN nodes during the CATVFR task.

#### Encoding

Directed information flow from the AI to the dPPC was higher than the reverse ($F(1, 639) = 27.16$, p<0.001, Cohen's $d$ = 0.41) (*Figure 5b*).

#### Recall

Directed information flow from the AI to the dPPC was higher than the reverse ($F(1, 639) = 20.48$, p<0.001, Cohen's $d$ = 0.36) (*Figure 5b*).

These results demonstrate that the AI has stronger directed information flow to the dPPC node of the FPN during both the encoding and recall phases of the CATVFR episodic memory task.

## Directed information flow from AI to FPN nodes in the PALVCR task

We next examined directed information flow between the AI and FPN nodes during the PALVCR task.

### Encoding

Directed information flow from the AI to the dPPC ($F(1, 476) = 38.25$, p<0.001, Cohen's $d = 0.57$) was higher than the reverse (*Figure 5c*).

### Recall

Directed information flow from the AI to the dPPC ($F(1, 475) = 60.09$, p<0.001, Cohen's $d = 0.71$) and MFG ($F(1, 709) = 9.90$, p<0.01, Cohen's $d = 0.24$) was higher than the reverse (*Figure 5c*).

These results demonstrate that the AI has stronger directed information flow to the dPPC node of the FPN during encoding and both dPPC and MFG nodes of the FPN during the recall phase of the PALVCR episodic memory task.

## Directed information flow from the AI to FPN nodes in the WMSM task

### Encoding

Directed information flow from the AI to the MFG ($F(1, 343) = 74.38$, p<0.001, Cohen's $d = 0.93$) was higher than the reverse (*Figure 5d*).

### Recall

Directed information flow from the AI to the MFG ($F(1, 344) = 102.18$, p<0.001, Cohen's $d = 1.09$) was higher than the reverse (*Figure 5d*).

These results demonstrate that the AI has stronger directed information flow to the MFG node of the FPN during both the encoding and recall phases of the WMSM task.

## Outflow hub during encoding and recall in the CATVFR task

### Encoding

Net outflow from the AI is positive and higher than both PCC/precuneus ($F(1, 2023) = 59.97$, p<0.001, Cohen's $d = 0.34$) and mPFC ($F(1, 2676) = 23.16$, p<0.001, Cohen's $d = 0.19$) during encoding (*Figure 6b*).

We also found that the net outflow from the AI is higher than the MFG during encoding ($F(1, 3974) = 11.61$, p<0.001, Cohen's $d = 0.11$) (*Figure 6b*). However, the net outflow from the AI was lower than the dPPC during encoding ($F(1, 3535) = 6.04$, p<0.05, Cohen's $d = 0.08$) (*Figure 6b*).

### Recall

Net outflow from the AI is positive and higher than both PCC/precuneus ($F(1, 1827) = 33.55$, p<0.001, Cohen's $d = 0.27$) and mPFC ($F(1, 2656) = 29.81$, p<0.001, Cohen's $d = 0.21$) during the recall phase of the CATVFR task (*Figure 6b*).

We also found that the net outflow from the AI is higher than the MFG during recall ($F(1, 3827) = 6.87$, p<0.01, Cohen's $d = 0.08$) (*Figure 6b*).

## Outflow hub during encoding and recall in the PALVCR task

### Encoding

We found similar results for the PALVCR task where net outflow from the AI is positive and higher than both PCC/precuneus ($F(1, 736) = 9.84$, p<0.01, Cohen's $d = 0.23$) and mPFC ($F(1, 1079) = 21.93$, p<0.001, Cohen's $d = 0.29$) during memory encoding (*Figure 6c*).

We also found that the net outflow from the AI is higher than the MFG during encoding ($F(1, 1779) = 14.45$, p<0.001, Cohen's $d = 0.18$) (*Figure 6c*). However, the net outflow from the AI is lower than the dPPC during encoding ($F(1, 1261) = 8.72$, p<0.01, Cohen's $d = 0.17$) (*Figure 6c*).

### Recall

Net outflow from the AI is positive and higher than both PCC/precuneus ($F(1, 530) = 10.96$, p<0.001, Cohen's $d = 0.29$) and mPFC ($F(1, 909) = 8.42$, p<0.01, Cohen's $d = 0.19$) during memory recall (*Figure 6c*).

Net outflow from the AI is higher than both the dPPC ($F$(1, 1041) = 31.15, p<0.001, Cohen's d = 0.35) and MFG ($F$(1, 736) = 70.08, p<0.001, Cohen's $d$ = 0.62) nodes of the FPN during the recall phase of the PALVCR task (**Figure 6c**).

Together, these results demonstrate that the AI is an outflow hub in its interactions with the PCC/precuneus and mPFC nodes of the DMN and the MFG node of the FPN during both verbal memory encoding and recall.

## Outflow hub during encoding and recall in the WMSM task
We next repeated our hub analysis during the encoding and recall phases of the WMSM task.

### Encoding
We found that net outflow from the AI is positive and higher than both the PCC/precuneus ($F$(1, 1669) = 168.5, p<0.001, Cohen's $d$ = 0.64) and mPFC ($F$(1, 1278) = 9.91, p<0.01, Cohen's $d$ = 0.18) nodes of the DMN during encoding (**Figure 6d**).

We also found that net outflow from the AI is higher than both the dPPC ($F$(1, 2501) = 7.10, p<0.01, Cohen's $d$ = 0.11) and MFG ($F$(1, 1977) = 73.49, p<0.001, Cohen's $d$ = 0.39) nodes of the FPN during encoding (**Figure 6d**).

### Recall
Net outflow from the AI is positive and higher than both the PCC/precuneus ($F$(1, 1672) = 166.95, p<0.001, Cohen's $d$ = 0.63) and mPFC ($F$(1, 1270) = 12.75, p<0.001, Cohen's $d$ = 0.20) nodes of the DMN during recall (**Figure 6d**).

Net outflow from the AI is also higher than the MFG ($F$(1, 1985) = 90.81, p<0.001, Cohen's $d$ = 0.43) node of the FPN during recall (**Figure 6d**).

Together, these results demonstrate that the AI is an outflow hub in its interactions with the PCC/precuneus and mPFC nodes of the DMN and also the dPPC and MFG nodes of the FPN, during both spatial memory encoding and recall.

## Appendix tables

**Appendix 1—table 1.** Participant demographic information (total 177 participants).

| Participant ID | Gender | Age |
| --- | --- | --- |
| 001 | F | 48 |
| 002 | F | 49 |
| 003 | F | 39 |
| 006 | F | 20 |
| 010 | F | 30 |
| 014 | F | 47 |
| 015 | F | 54 |
| 018 | M | 47 |
| 019 | F | 34 |
| 020 | F | 48 |
| 021 | M | 38 |
| 022 | M | 24 |
| 023 | M | 32 |
| 024 | F | 36 |
| 025 | F | 19 |
| 026 | F | 24 |

*Appendix 1—table 1 Continued on next page*

*Appendix 1—table 1 Continued*

| Participant ID | Gender | Age |
|---|---|---|
| 027 | M | 48 |
| 028 | F | 27 |
| 029 | F | 33 |
| 030 | M | 23 |
| 032 | F | 19 |
| 033 | F | 31 |
| 034 | F | 29 |
| 035 | F | 45 |
| 036 | M | 49 |
| 039 | F | 28 |
| 041 | M | 34 |
| 042 | F | 28 |
| 044 | M | 58 |
| 045 | M | 51 |
| 049 | F | 52 |
| 050 | M | 20 |
| 051 | F | 24 |
| 052 | F | 19 |
| 053 | F | 39 |
| 054 | M | 23 |
| 056 | M | 34 |
| 057 | M | 53 |
| 059 | F | 44 |
| 060 | F | 36 |
| 062 | F | 23 |
| 063 | M | 23 |
| 064 | M | 56 |
| 065 | F | 34 |
| 066 | M | 39 |
| 067 | F | 45 |
| 068 | F | 39 |
| 069 | M | 26 |
| 070 | F | 40 |
| 074 | M | 24 |
| 075 | M | 50 |
| 076 | M | 29 |
| 077 | F | 47 |
| 078 | F | 22 |

*Appendix 1—table 1 Continued on next page*

*Appendix 1—table 1 Continued*

| Participant ID | Gender | Age |
| --- | --- | --- |
| 080 | F | 43 |
| 081 | F | 33 |
| 082 | M | 39 |
| 084 | M | 25 |
| 087 | M | 51 |
| 089 | M | 36 |
| 090 | F | 52 |
| 091 | M | 28 |
| 092 | M | 44 |
| 093 | M | 24 |
| 094 | M | 47 |
| 095 | F | 35 |
| 097 | M | 34 |
| 098 | F | 38 |
| 100 | F | 43 |
| 101 | F | 26 |
| 102 | M | 34 |
| 105 | M | 25 |
| 106 | M | 26 |
| 107 | M | 25 |
| 108 | F | 23 |
| 109 | F | 43 |
| 111 | M | 20 |
| 114 | F | 31 |
| 115 | M | 47 |
| 118 | M | 33 |
| 119 | F | 26 |
| 120 | F | 33 |
| 121 | M | 34 |
| 123 | F | 29 |
| 124 | F | 40 |
| 125 | F | 44 |
| 127 | F | 40 |
| 128 | M | 26 |
| 129 | F | 34 |
| 130 | M | 57 |
| 131 | M | 24 |
| 134 | M | 64 |

*Appendix 1—table 1 Continued on next page*

*Appendix 1—table 1 Continued*

| Participant ID | Gender | Age |
|---|---|---|
| 135 | M | 47 |
| 136 | F | 16 |
| 137 | F | 21 |
| 138 | M | 41 |
| 141 | F | 44 |
| 142 | F | 43 |
| 144 | M | 53 |
| 147 | M | 47 |
| 148 | F | 59 |
| 149 | F | 28 |
| 150 | F | 49 |
| 151 | M | 36 |
| 153 | M | 38 |
| 155 | M | 37 |
| 156 | M | 27 |
| 157 | M | 22 |
| 158 | F | 45 |
| 159 | F | 42 |
| 161 | F | 53 |
| 162 | F | 30 |
| 163 | M | 45 |
| 164 | M | 37 |
| 166 | M | 38 |
| 167 | M | 33 |
| 168 | M | 24 |
| 171 | M | 36 |
| 172 | F | 22 |
| 173 | F | 18 |
| 174 | M | 29 |
| 175 | M | 34 |
| 176 | F | 41 |
| 177 | F | 23 |
| 178 | M | 40 |
| 180 | F | 21 |
| 181 | M | 22 |
| 184 | M | 42 |
| 186 | M | 27 |
| 187 | F | 51 |

*Appendix 1—table 1 Continued on next page*

*Appendix 1—table 1 Continued*

| Participant ID | Gender | Age |
|---|---|---|
| 189 | M | 22 |
| 190 | F | 57 |
| 193 | M | 37 |
| 195 | M | 44 |
| 196 | M | 18 |
| 200 | M | 25 |
| 202 | F | 29 |
| 203 | F | 36 |
| 204 | F | 25 |
| 207 | F | 39 |
| 212 | M | 46 |
| 221 | M | 57 |
| 222 | F | 20 |
| 223 | F | 42 |
| 227 | M | 32 |
| 228 | F | 58 |
| 230 | F | 56 |
| 232 | M | 27 |
| 234 | M | 25 |
| 236 | F | 51 |
| 238 | M | 27 |
| 239 | M | 27 |
| 240 | F | 37 |
| 245 | M | 30 |
| 247 | F | 61 |
| 251 | M | 31 |
| 260 | F | 57 |
| 263 | M | 30 |
| 264 | F | 52 |
| 268 | F | 32 |
| 271 | M | 37 |
| 274 | F | 44 |
| 275 | M | 41 |
| 276 | M | 28 |
| 279 | F | 57 |
| 283 | F | 29 |
| 284 | F | 32 |
| 286 | F | 57 |

*Appendix 1—table 1 Continued on next page*

*Appendix 1—table 1 Continued*

| Participant ID | Gender | Age |
|---|---|---|
| 292 | F | 39 |
| 297 | M | 24 |
| 298 | F | 24 |
| 299 | M | 43 |
| 302 | M | 48 |
| 303 | F | 62 |
| 304 | F | 33 |
| 310 | M | 20 |
| 312 | M | 21 |

**Appendix 1—table 2.** Number of electrode pairs used in phase transfer entropy (PTE) analysis in the verbal free recall task.

AI: anterior insula, PCC: posterior cingulate cortex, Pr: precuneus, mPFC: medial prefrontal cortex, dPPC: dorsal posterior parietal cortex, MFG: middle frontal gyrus.

| Network pair | Number of electrode pairs (n) | Number of participants | Participant IDs (gender/age) |
|---|---|---|---|
| AI-PCC/Pr | 142 | 18 | 030 (M/23), 049 (F/52), 054 (M/23), 057 (M/53), 062 (F/23), 114 (F/31), 115 (M/47), 134 (M/64), 153 (M/38), 158 (F/45), 168 (M/24), 193 (M/37), 196 (M/18), 204 (F/25), 236 (F/51), 240 (F/37), 286 (F/57), 299 (M/43) |
| AI-mPFC | 112 | 20 | 026 (F/24), 027 (M/48), 049 (F/52), 057 (M/53), 062 (F/23), 114 (F/31), 115 (M/47), 123 (F/29), 153 (M/38), 163 (M/45), 168 (M/24), 189 (M/22), 193 (M/37), 196 (M/18), 204 (F/25), 223 (F/42), 228 (F/58), 247 (F/61), 274 (F/44), 299 (M/43) |
| AI-dPPC | 586 | 28 | 030 (M/23), 032 (F/19), 033 (F/31), 049 (F/52), 054 (M/23), 057 (M/53), 062 (F/23), 065 (F/34), 080 (F/43), 114 (F/31), 115 (M/47), 128 (M/26), 134 (M/64), 153 (M/38), 158 (F/45), 163 (M/45), 168 (M/24), 173 (F/18), 189 (M/22), 193 (M/37), 196 (M/18), 204 (F/25), 232 (M/27), 236 (F/51), 240 (F/37), 247 (F/61), 286 (F/57), 299 (M/43) |
| AI-MFG | 642 | 36 | 026 (F/24), 030 (M/23), 032 (F/19), 033 (F/31), 049 (F/52), 054 (M/23), 057 (M/53), 062 (F/23), 063 (M/23), 065 (F/34), 114 (F/31), 115 (M/47), 153 (M/38), 158 (F/45), 163 (M/45), 166 (M/38), 168 (M/24), 178 (M/40), 189 (M/22), 193 (M/37), 196 (M/18), 204 (F/25), 207 (F/39), 223 (F/42), 228 (F/58), 230 (F/56), 232 (M/27), 240 (F/37), 247 (F/61), 264 (F/52), 274 (F/44), 283 (F/29), 286 (F/57), 298 (F/24), 299 (M/43), 310 (M/20) |

**Appendix 1—table 3.** Number of electrode pairs used in phase transfer entropy (PTE) analysis in the categorized verbal free recall task.

AI: anterior insula, PCC: posterior cingulate cortex, Pr: precuneus, mPFC: medial prefrontal cortex, dPPC: dorsal posterior parietal cortex, MFG: middle frontal gyrus.

| Network pair | Number of electrode pairs (n) | Number of participants | Participant IDs (gender/age) |
|---|---|---|---|
| AI-PCC/Pr | 46 | 7 | 114 (F/31), 141 (F/44), 158 (F/45), 204 (F/25), 240 (F/37), 245 (M/30), 286 (F/57) |
| AI-mPFC | 64 | 12 | 026 (F/24), 114 (F/31), 141 (F/44), 163 (M/45), 189 (M/22), 204 (F/25), 228 (F/58), 245 (M/30), 247 (F/61), 271 (M/37), 274 (F/44), 303 (F/62) |
| AI-dPPC | 327 | 14 | 028 (F/27), 032 (F/19), 065 (F/34), 114 (F/31), 141 (F/44), 158 (F/45), 163 (M/45), 189 (M/22), 204 (F/25), 240 (F/37), 245 (M/30), 247 (F/61), 271 (M/37), 286 (F/57) |
| AI-MFG | 462 | 22 | 026 (F/24), 032 (F/19), 065 (F/34), 114 (F/31), 141 (F/44), 158 (F/45), 163 (M/45), 178 (M/40), 189 (M/22), 204 (F/25), 207 (F/39), 228 (F/58), 230 (F/56), 240 (F/37), 245 (M/30), 247 (F/61), 264 (F/52), 271 (M/37), 274 (F/44), 286 (F/57), 303 (F/62), 310 (M/20) |

**Appendix 1—table 4.** Number of electrode pairs used in phase transfer entropy (PTE) analysis in the paired associates learning verbal cued recall task.

AI: anterior insula, PCC: posterior cingulate cortex, Pr: precuneus, mPFC: medial prefrontal cortex, dPPC: dorsal posterior parietal cortex, MFG: middle frontal gyrus.

| Network pair | Number of electrode pairs (n) | Number of participants | Participant IDs (gender/age) |
|---|---|---|---|
| AI-PCC/Pr | 10 | 2 | 141 (F/44), 196 (M/18) |
| AI-mPFC | 36 | 5 | 141 (F/44), 196 (M/18), 223 (F/42), 228 (F/58), 303 (F/62) |
| AI-dPPC | 242 | 9 | 028 (F/27), 065 (F/34), 090 (F/52), 091 (M/28), 141 (F/44), 196 (M/18), 232 (M/27), 238 (M/27), 312 (M/21) |
| AI-MFG | 362 | 14 | 065 (F/34), 090 (F/52), 091 (M/28), 141 (F/44), 196 (M/18), 207 (F/39), 223 (F/42), 228 (F/58), 230 (F/56), 232 (M/27), 238 (M/27), 283 (F/29), 303 (F/62), 312 (M/21) |

**Appendix 1—table 5.** Number of electrode pairs used in phase transfer entropy (PTE) analysis in the water maze spatial memory task.

AI: anterior insula, PCC: posterior cingulate cortex, Pr: precuneus, mPFC: medial prefrontal cortex, dPPC: dorsal posterior parietal cortex, MFG: middle frontal gyrus.

| Network pair | Number of electrode pairs (n) | Number of participants | Participant IDs (gender/age) |
|---|---|---|---|
| AI-PCC/Pr | 91 | 6 | 030 (M/23), 049 (F/52), 054 (M/23), 062 (F/23), 114 (F/31), 124 (F/40) |
| AI-mPFC | 23 | 5 | 026 (F/24), 049 (F/52), 052 (F/19), 062 (F/23), 114 (F/31) |
| AI-dPPC | 302 | 10 | 030 (M/23), 032 (F/19), 033 (F/31), 049 (F/52), 052 (F/19), 054 (M/23), 062 (F/23), 065 (F/34), 114 (F/31), 124 (F/40) |
| AI-MFG | 177 | 10 | 026 (F/24), 030 (M/23), 032 (F/19), 033 (F/31), 049 (F/52), 052 (F/19), 054 (M/23), 062 (F/23), 065 (F/34), 114 (F/31) |

**Appendix 1—table 6.** Number of electrodes in each node used in high-gamma power analysis in the verbal free recall task.

AI: anterior insula, PCC: posterior cingulate cortex, Pr: precuneus, mPFC: medial prefrontal cortex, dPPC: dorsal posterior parietal cortex, MFG: middle frontal gyrus.

| Brain regions | Number of electrodes (n) | Number of participants | Participant IDs (gender/age) |
|---|---|---|---|
| AI | 148 | 44 | 026 (F/24), 027 (M/48), 030 (M/23), 032 (F/19), 033 (F/31), 049 (F/52), 054 (M/23), 057 (M/53), 062 (F/23), 063 (M/23), 065 (F/34), 080 (F/43), 114 (F/31), 115 (M/47), 123 (F/29), 128 (M/26), 134 (M/64), 150 (F/49), 153 (M/38), 158 (F/45), 163 (M/45), 166 (M/38), 168 (M/24), 173 (F/18), 178 (M/40), 189 (M/22), 193 (M/37), 196 (M/18), 204 (F/25), 207 (F/39), 223 (F/42), 228 (F/58), 230 (F/56), 232 (M/27), 236 (F/51), 240 (F/37), 247 (F/61), 264 (F/52), 274 (F/44), 283 (F/29), 286 (F/57), 298 (F/24), 299 (M/43), 310 (M/20) |
| PCC/Pr | 143 | 47 | 006 (F/20), 010 (F/30), 015 (F/54), 018 (M/47), 023 (M/32), 030 (M/23), 034 (F/29), 039 (F/28), 044 (M/58), 049 (F/52), 051 (F/24), 054 (M/23), 057 (M/53), 062 (F/23), 070 (F/40), 074 (M/24), 076 (M/29), 077 (F/47), 081 (F/33), 084 (M/25), 094 (M/47), 101 (F/26), 105 (M/25), 106 (M/26), 114 (F/31), 115 (M/47), 134 (M/64), 135 (M/47), 138 (M/41), 153 (M/38), 155 (M/37), 158 (F/45), 162 (F/30), 168 (M/24), 175 (M/34), 186 (M/27), 193 (M/37), 196 (M/18), 203 (F/36), 204 (F/25), 236 (F/51), 240 (F/37), 268 (F/32), 275 (M/41), 286 (F/57), 297 (M/24), 299 (M/43) |
| mPFC | 312 | 55 | 018 (M/47), 022 (M/24), 026 (F/24), 027 (M/48), 034 (F/29), 036 (M/49), 039 (F/28), 049 (F/52), 051 (F/24), 053 (F/39), 056 (M/34), 057 (M/53), 059 (F/44), 060 (F/36), 062 (F/23), 070 (F/40), 074 (M/24), 075 (M/50), 077 (F/47), 081 (F/33), 084 (M/25), 098 (F/38), 106 (M/26), 114 (F/31), 115 (M/47), 121 (M/34), 123 (F/29), 129 (F/34), 130 (M/57), 131 (M/24), 142 (F/43), 151 (M/36), 153 (M/38), 155 (M/37), 156 (M/27), 163 (M/45), 167 (M/33), 168 (M/24), 175 (M/34), 187 (F/51), 189 (M/22), 193 (M/37), 196 (M/18), 200 (M/25), 202 (F/29), 203 (F/36), 204 (F/25), 222 (F/20), 223 (F/42), 228 (F/58), 247 (F/61), 274 (F/44), 275 (M/41), 299 (M/43), 304 (F/33) |

*Appendix 1—table 6 Continued on next page*

*Appendix 1—table 6 Continued*

| Brain regions | Number of electrodes (n) | Number of participants | Participant IDs (gender/age) |
|---|---|---|---|
| dPPC | 537 | 89 | 001 (F/48), 003 (F/39), 006 (F/20), 010 (F/30), 015 (F/54), 018 (M/47), 020 (F/48), 023 (M/32), 030 (M/23), 032 (F/19), 033 (F/31), 035 (F/45), 036 (M/49), 039 (F/28), 042 (F/28), 044 (M/58), 049 (F/52), 050 (M/20), 053 (F/39), 054 (M/23), 056 (M/34), 057 (M/53), 059 (F/44), 062 (F/23), 065 (F/34), 066 (M/39), 067 (F/45), 068 (F/39), 069 (M/26), 070 (F/40), 074 (M/24), 075 (M/50), 077 (F/47), 080 (F/43), 084 (M/25), 089 (M/36), 094 (M/47), 101 (F/26), 102 (M/34), 105 (M/25), 106 (M/26), 111 (M/20), 114 (F/31), 115 (M/47), 120 (F/33), 121 (M/34), 125 (F/44), 128 (M/26), 130 (M/57), 134 (M/64), 135 (M/47), 138 (M/41), 147 (M/47), 151 (M/36), 153 (M/38), 156 (M/27), 158 (F/45), 161 (F/53), 162 (F/30), 163 (M/45), 164 (M/37), 168 (M/24), 171 (M/36), 173 (F/18), 174 (M/29), 175 (M/34), 176 (F/41), 177 (F/23), 184 (M/42), 186 (M/27), 189 (M/22), 193 (M/37), 195 (M/44), 196 (M/18), 203 (F/36), 204 (F/25), 232 (M/27), 234 (M/25), 236 (F/51), 240 (F/37), 247 (F/61), 251 (M/31), 260 (F/57), 268 (F/32), 275 (M/41), 286 (F/57), 292 (F/39), 297 (M/24), 299 (M/43) |
| MFG | 538 | 97 | 002 (F/49), 003 (F/39), 006 (F/20), 015 (F/54), 020 (F/48), 022 (M/24), 023 (M/32), 026 (F/24), 030 (M/23), 032 (F/19), 033 (F/31), 034 (F/29), 036 (M/49), 039 (F/28), 042 (F/28), 045 (M/51), 049 (F/52), 051 (F/24), 053 (F/39), 054 (M/23), 056 (M/34), 057 (M/53), 059 (F/44), 060 (F/36), 062 (F/23), 063 (M/23), 065 (F/34), 066 (M/39), 067 (F/45), 069 (M/26), 070 (F/40), 074 (M/24), 075 (M/50), 076 (M/29), 077 (F/47), 081 (F/33), 084 (M/25), 089 (M/36), 098 (F/38), 102 (M/34), 105 (M/25), 106 (M/26), 114 (F/31), 115 (M/47), 121 (M/34), 127 (F/40), 129 (F/34), 130 (M/57), 131 (M/24), 135 (M/47), 136 (F/16), 137 (F/21), 142 (F/43), 147 (M/47), 148 (F/59), 149 (F/28), 151 (M/36), 153 (M/38), 155 (M/37), 156 (M/27), 158 (F/45), 159 (F/42), 162 (F/30), 163 (M/45), 164 (M/37), 166 (M/38), 168 (M/24), 172 (F/22), 175 (M/34), 177 (F/23), 178 (M/40), 186 (M/27), 189 (M/22), 193 (M/37), 195 (M/44), 196 (M/18), 200 (M/25), 203 (F/36), 204 (F/25), 207 (F/39), 222 (F/20), 223 (F/42), 228 (F/58), 230 (F/56), 232 (M/27), 240 (F/37), 247 (F/61), 260 (F/57), 264 (F/52), 274 (F/44), 275 (M/41), 283 (F/29), 286 (F/57), 298 (F/24), 299 (M/43), 304 (F/33), 310 (M/20) |

**Appendix 1—table 7.** Number of electrodes in each node used in high-gamma power analysis in the categorized verbal free recall task.

AI: anterior insula, PCC: posterior cingulate cortex, Pr: precuneus, mPFC: medial prefrontal cortex, dPPC: dorsal posterior parietal cortex, MFG: middle frontal gyrus.

| Brain regions | Number of electrodes (n) | Number of participants | Participant IDs (gender/age) |
|---|---|---|---|
| AI | 107 | 25 | 026 (F/24), 028 (F/27), 032 (F/19), 065 (F/34), 114 (F/31), 141 (F/44), 158 (F/45), 163 (M/45), 178 (M/40), 189 (M/22), 204 (F/25), 207 (F/39), 228 (F/58), 230 (F/56), 236 (F/51), 239 (M/27), 240 (F/37), 245 (M/30), 247 (F/61), 264 (F/52), 271 (M/37), 274 (F/44), 286 (F/57), 303 (F/62), 310 (M/20) |
| PCC/Pr | 74 | 21 | 015 (F/54), 039 (F/28), 041 (M/34), 044 (M/58), 074 (M/24), 094 (M/47), 105 (M/25), 106 (M/26), 114 (F/31), 135 (M/47), 141 (F/44), 157 (M/22), 158 (F/45), 186 (M/27), 204 (F/25), 227 (M/32), 236 (F/51), 240 (F/37), 245 (M/30), 275 (M/41), 286 (F/57) |
| mPFC | 116 | 33 | 026 (F/24), 029 (F/33), 036 (M/49), 039 (F/28), 041 (M/34), 056 (M/34), 060 (F/36), 074 (M/24), 075 (M/50), 106 (M/26), 107 (M/25), 114 (F/31), 119 (F/26), 130 (M/57), 131 (M/24), 141 (F/44), 163 (M/45), 167 (M/33), 180 (F/21), 181 (M/22), 187 (F/51), 189 (M/22), 202 (F/29), 204 (F/25), 212 (M/46), 222 (F/20), 228 (F/58), 245 (M/30), 247 (F/61), 271 (M/37), 274 (F/44), 275 (M/41), 303 (F/62) |
| dPPC | 357 | 57 | 015 (F/54), 028 (F/27), 032 (F/19), 035 (F/45), 036 (M/49), 039 (F/28), 042 (F/28), 044 (M/58), 050 (M/20), 056 (M/34), 065 (F/34), 066 (M/39), 067 (F/45), 069 (M/26), 074 (M/24), 075 (M/50), 089 (M/36), 092 (M/44), 094 (M/47), 102 (M/34), 105 (M/25), 106 (M/26), 108 (F/23), 111 (M/20), 114 (F/31), 119 (F/26), 130 (M/57), 135 (M/47), 141 (F/44), 144 (M/53), 147 (M/47), 157 (M/22), 158 (F/45), 163 (M/45), 171 (M/36), 174 (M/29), 176 (F/41), 181 (M/22), 184 (M/42), 186 (M/27), 189 (M/22), 190 (F/57), 204 (F/25), 212 (M/46), 221 (M/57), 227 (M/32), 236 (F/51), 240 (F/37), 245 (M/30), 247 (F/61), 251 (M/31), 260 (F/57), 271 (M/37), 275 (M/41), 279 (F/57), 286 (F/57), 302 (M/48) |
| MFG | 375 | 58 | 015 (F/54), 021 (M/38), 026 (F/24), 029 (F/33), 032 (F/19), 036 (M/49), 039 (F/28), 041 (M/34), 042 (F/28), 045 (M/51), 056 (M/34), 060 (F/36), 065 (F/34), 066 (M/39), 067 (F/45), 069 (M/26), 074 (M/24), 075 (M/50), 089 (M/36), 092 (M/44), 093 (M/24), 102 (M/34), 105 (M/25), 106 (M/26), 107 (M/25), 108 (F/23), 114 (F/31), 119 (F/26), 130 (M/57), 131 (M/24), 135 (M/47), 141 (F/44), 147 (M/47), 157 (M/22), 158 (F/45), 163 (M/45), 178 (M/40), 181 (M/22), 186 (M/27), 189 (M/22), 204 (F/25), 207 (F/39), 212 (M/46), 221 (M/57), 222 (F/20), 228 (F/58), 230 (F/56), 240 (F/37), 245 (M/30), 247 (F/61), 260 (F/57), 264 (F/52), 271 (M/37), 274 (F/44), 275 (M/41), 286 (F/57), 303 (F/62), 310 (M/20) |

**Appendix 1—table 8.** Number of electrodes in each node used in high-gamma power analysis in the paired associates learning verbal cued recall task.

AI: anterior insula, PCC: posterior cingulate cortex, Pr: precuneus, mPFC: medial prefrontal cortex, dPPC: dorsal posterior parietal cortex, MFG: middle frontal gyrus.

| Brain regions | Number of electrodes (n) | Number of participants | Participant IDs (gender/age) |
|---|---|---|---|
| AI | 84 | 15 | 028 (F/27), 065 (F/34), 090 (F/52), 091 (M/28), 141 (F/44), 196 (M/18), 207 (F/39), 223 (F/42), 228 (F/58), 230 (F/56), 232 (M/27), 238 (M/27), 283 (F/29), 303 (F/62), 312 (M/21) |
| PCC/Pr | 28 | 10 | 023 (M/32), 074 (M/24), 078 (F/2), 106 (M/26), 141 (F/44), 162 (F/30), 175 (M/34), 196 (M/18), 284 (F/32), 297 (M/24) |
| mPFC | 78 | 20 | 036 (M/49), 056 (M/34), 060 (F/36), 074 (M/24), 082 (M/39), 097 (M/34), 106 (M/26), 121 (M/34), 130 (M/57), 131 (M/24), 141 (F/43), 175 (M/34), 196 (M/18), 202 (F/29), 212 (M/46), 223 (F/42), 228 (F/58), 263 (M/30), 303 (F/62) |
| dPPC | 192 | 39 | 001 (F/48), 003 (F/39), 023 (M/32), 028 (F/27), 035 (F/45), 036 (M/49), 042 (F/28), 050 (M/20), 056 (M/34), 065 (F/34), 066 (M/39), 069 (M/26), 074 (M/24), 078 (F/22), 082 (M/39), 087 (M/51), 089 (M/36), 090 (F/52), 091 (M/28), 095 (F/35), 097 (M/34), 102 (M/34), 106 (M/26), 109 (F/43), 111 (M/20), 118 (M/33), 121 (M/34), 130 (M/57), 141 (F/44), 162 (F/30), 175 (M/34), 196 (M/18), 212 (M/46), 232 (M/27), 238 (M/27), 276 (M/28), 284 (F/32), 297 (M/24), 312 (M/21) |
| MFG | 204 | 44 | 002 (F/49), 003 (F/39), 023 (M/32), 036 (M/49), 042 (F/28), 056 (M/34), 060 (F/36), 065 (F/34), 066 (M/39), 069 (M/26), 074 (M/24), 078 (F/22), 082 (M/39), 089 (M/36), 090 (F/52), 091 (M/28), 095 (F/35), 097 (M/34), 100 (F/43), 102 (M/34), 106 (M/26), 118 (M/33), 121 (M/34), 130 (M/57), 131 (M/24), 136 (F/16), 141 (F/44), 142 (F/43), 149 (F/28), 162 (F/30), 175 (M/34), 196 (M/18), 207 (F/39), 212 (M/46), 223 (F/42), 228 (F/58), 230 (F/56), 232 (M/27), 238 (M/27), 263 (M/30), 276 (M/28), 283 (F/29), 303 (F/62), 312 (M/21) |

**Appendix 1—table 9.** Number of electrodes in each node used in power spectral density (PSD) analysis in the water maze spatial memory task.

AI: anterior insula, PCC: posterior cingulate cortex, Pr: precuneus, mPFC: medial prefrontal cortex, dPPC: dorsal posterior parietal cortex, MFG: middle frontal gyrus.

| Brain regions | Number of electrodes (n) | Number of participants | Participant IDs (gender/age) |
|---|---|---|---|
| AI | 59 | 11 | 026 (F/24), 030 (M/23), 032 (F/19), 033 (F/31), 049 (F/52), 052 (F/19), 054 (M/23), 062 (F/23), 065 (F/34), 114 (F/31), 124 (F/40) |
| PCC/Pr | 89 | 21 | 006 (F/20), 010 (F/30), 015 (F/54), 018 (M/47), 023 (M/32), 024 (F/36), 030 (M/23), 034 (F/29), 041 (M/34), 044 (M/58), 049 (F/52), 051 (F/24), 054 (M/23), 062 (F/23), 064 (M/56), 074 (M/24), 077 (F/47), 101 (F/26), 106 (M/26), 114 (F/31), 124 (F/40) |
| mPFC | 77 | 17 | 014 (F/47), 018 (M/47), 025 (F/19), 026 (F/24), 034 (F/29), 041 (M/34), 049 (F/52), 051 (F/24), 052 (F/19), 056 (M/34), 060 (F/36), 062 (F/23), 074 (M/24), 075 (M/50), 077 (F/47), 106 (M/26), 114 (F/31) |
| dPPC | 226 | 36 | 001 (F/48), 006 (F/20), 010 (F/30), 014 (F/47), 015 (F/54), 018 (M/47), 019 (F/34), 023 (M/32), 024 (F/36), 025 (F/19), 030 (M/23), 032 (F/19), 033 (F/31), 042 (F/28), 044 (M/58), 049 (F/52), 050 (M/20), 052 (F/19), 054 (M/23), 056 (M/34), 062 (F/23), 064 (M/56), 065 (F/34), 066 (M/39), 067 (F/45), 068 (F/39), 069 (M/26), 074 (M/24), 075 (M/50), 077 (F/47), 089 (M/36), 101 (F/26), 106 (M/26), 114 (F/31), 124 (F/40), 177 (F/23) |
| MFG | 147 | 33 | 006 (F/20), 014 (F/47), 015 (F/54), 019 (F/34), 021 (M/38), 023 (M/32), 025 (F/19), 026 (F/24), 030 (M/23), 032 (F/19), 033 (F/31), 034 (F/29), 041 (M/34), 042 (F/28), 045 (M/51), 049 (F/52), 051 (F/24), 052 (F/19), 054 (M/23), 056 (M/34), 060 (F/36), 062 (F/23), 065 (F/34), 066 (M/39), 067 (F/45), 069 (M/26), 074 (M/24), 075 (M/50), 077 (F/47), 089 (M/36), 106 (M/26), 114 (F/31), 177 (F/23) |

