## [Editor Report · eLife Assessment]

In this article, the authors present **valuable** findings on the apparent role of a salience network anterior insula node in directing frontoparietal and default mode network activity within a tripartite network during control of memory, drawn from an impressive invasive human neurophysiological dataset. Overall, the authors have presented a **convincing** set of analyses. We also commend the use of a large intracranial EEG dataset to approach this question.

---

## [Referee Report · Reviewer #1 (Public review)]

Summary

Das and Menon describe an analysis of a large open-source iEEG dataset (UPENN-RAM). From encoding and recall phases of memory tasks, they analyzed power and phase-transfer entropy as a measure of directed information flow in regions across a hypothesized tripartite network system. The anterior insula (AI) was found to have heightened high gamma power during encoding and retrieval, which corresponded to suppression of high gamma power in medial prefrontal cortex (mPFC) and posterior cingulate cortex (PCC) during encoding but not recall. In contrast, directed information flow from (but not to) AI to mPFC and PCC is high during both time periods when PTE is analyzed with broadband but not narrowband activity. They claim that these findings significantly advance an understanding of how network communication facilitates cognitive operations during memory tasks, and that the AI of the salience network (SN) is responsible for influencing both the frontoparietal network (FPN) and default-mode network (DMN) during memory encoding and retrieval.

I find this question interesting and important and agree with the authors that iEEG presents a unique opportunity to investigate the temporal dynamics within network nodes. Their findings convey intriguing information about the structure and order of communication between network regions during on-task cognition in general (though, perhaps not specific to memory - see Weaknesses), with the AI of the SN ostensibly playing an important role in possibly influencing the DMN and FPN.

Strengths

- The authors present results from an impressively-sized iEEG sample. For reader context, this type of invasive human data is difficult and time-consuming to collect and many similar studies in high-level journals include 5-20 participants, typically not all of whom have electrodes in all regions of interest. It is excellent that they have been able to leverage open-source data in this way.

- Preprocessing of iEEG data also seems sensible and appropriate based on field standards.

- The authors tackle the replication issues inherent in much of the literature by replicating findings across task contexts, demonstrating that the principles of network communication evidenced by their results generalize in multiple task memory contexts. Again, the number of iEEG patients who have multiple tasks' worth of data is impressive.

- Though the revised manuscript presents a broader and more novel investigation of the tripartite network's role in memory encoding and retrieval (as opposed to cognitive control of memory) the authors now thoroughly review the literature motivating this investigation of open-source data.

Weaknesses

- As the authors discuss, it is currently unclear if the directed information flow from AI to DMN and FPN nodes truly arises from memory-associated processes as opposed to more general attentional and cognitive demands, especially given that information flow does not relate meaningfully to task performance (whether memory retrieval is successful or not). I also note this is a concern because - though the authors have now demonstrated that information flow is increased compared to an off-task baseline - influences of AI on DMN or FPN were not increased relative to baseline epochs during the task in the original preprint version, again suggesting these effects may not be specific to the memory component of the analyzed tasks. The authors have thoughtfully noted in the Discussion several ways that experimental design can be improved in future studies to address this limitation.

---

## [Author Response]

The following is the authors’ response to the previous reviews.

Removing claims of causality: To avoid confusion, we have now removed claims of causality from our manuscript and also changed the title of the manuscript accordingly

"Electrophysiological dynamics of salience, default mode, and frontoparietal networks during episodic memory formation and recall: A multi-experiment iEEG replication".

Control analyses directly comparing AI and IFG: As per the reviewer’s suggestion, we have carried out additional control analyses by directly comparing the net inward/outward balance between the AI and the IFG. Our analysis revealed that the net outflow for the AI is significantly higher compared to the IFG during both encoding and recall phases, a pattern that was replicated across all four experiments.

These findings further highlight the unique role of the AI as a key hub in coordinating network interactions during episodic memory formation and retrieval, distinguishing it from a key anatomically adjacent prefrontal region implicated in cognitive control.

We have incorporated these results into the manuscript (see new Figure S6 and updated Results section).

Control analyses directly comparing task with resting state: As per the reviewer’s suggestion, we compared the AI's net outflow during task periods to resting state, finding significantly higher outflow during both encoding and recall across all experiments (*ps* < 0.05). These results provide further evidence for enhanced role of AI net directed information flow to the DMN and FPN during memory processing compared to the resting state.

We have incorporated these results into the manuscript (see new Figure S9 and updated Results section).

Control analysis using every region of the brain outside the considered networks: We appreciate the reviewer's suggestion to conduct additional control analyses. However, we have concerns about implementing this approach for several reasons:

(1) Hypothesis-driven research: Our study was designed based on a strong hypothesis derived from prior fMRI studies, which have consistently shown that the salience network (SN), anchored by the anterior insula (AI), plays a critical role in regulating the engagement and disengagement of the default mode network (DMN) and frontoparietal network (FPN) across diverse cognitive tasks.

(2) Risk of p-hacking: Running analyses on a large number of brain regions outside our networks of interest without a priori hypotheses could lead to p-hacking, a practice strongly criticized in the scientific community, including by eLife editors (Makin & Orban de Xivry, 2019). Such an approach could potentially yield spurious results and undermine the validity of our findings.

(3) Principled control region selection: Our choice of the inferior frontal gyrus (IFG) as a control region was hypothesis-driven, based on its: (a) Anatomical adjacency to the AI (b) Involvement in cognitive control functions, including response inhibition (c) Frequent coactivation with the AI in fMRI studies.

(4) Robustness of current findings: Our PTE analysis involving the IFG, along with the additional control analyses requested by the reviewer (comparing the task-related net balance of the AI with the IFG and with resting state, see response to reviewer comment 2.1), strongly support a key role for the AI in orchestrating large-scale network dynamics during memory processes.

(5) Specificity of findings: The contrast between AI and IFG results demonstrates that our observed patterns are not general to all task-active regions but are specific to the AI's role in network coordination.

We believe that our current analyses, including the additional controls, provide a comprehensive and rigorous examination of the AI's role in memory-related network dynamics. Adding analyses of numerous additional regions without clear hypotheses could potentially dilute the focus and interpretability of our results.

However, we acknowledge the importance of considering broader network interactions. In future studies, we could explore the role of other key regions in a hypothesis-driven manner, potentially expanding our understanding of the complex interactions between multiple brain networks during memory processes.

These revisions, combined with our rigorous methodologies and comprehensive analyses, provide compelling support for the central claims of our manuscript. We believe these changes significantly enhance the scientific contribution of our work.

Our point-by-point responses to the reviewers' comments are provided below.

**Reviewer 1:**
(1.1) Because phase-transfer entropy is referenced as a "causal" analysis in this investigation (PTE), I believe it is important to highlight for readers recent discussions surrounding the description of "causal mechanisms" in neuroscience (see "Confusion about causation" section from Ross and Bassett, 2024, Nature Neuroscience). A large proportion of neuroscientists (myself included) use "causal" only to refer to a mechanism whose modulation or removal (with direct manipulation, such as by lesion or stimulation) is known to change or control a given outcome (such as a successful behavior). As Ross and Bassett highlight, it is debatable whether such mechanistic causality is captured by Granger "causality" (a.k.a. Granger prediction) or the parametric PTE, and imprecise use of "causation" may be confusing. The authors have defined in the revised Introduction what their definition of "causality" is within the context of this investigation.

We appreciate the reviewer's feedback in terms of the terminology used in our manuscript. To avoid confusion, we have now removed claims of causality from our manuscript and also changed the title of the manuscript accordingly.

**Reviewer 2:**
(2.1) Clarifying the new control analyses. The authors have been responsive to our feedback and implemented several new analyses. The use of a pre-task baseline period and a control brain region (IFG) definitively help to contextualize their results, and the findings shown in the revision do suggest that (1) relative to a pre-task baseline, directed interactions from the AI are stronger and (2) relative to a nearby region, the IFG, the AI exhibits greater outward-directed influence.However, it is difficult to draw strong quantitative conclusions from the analyses as presented, because they do not directly statistically contrast the effect in question (directed interactions with the FPN and DMN) between two conditions (e.g. during baseline vs. during memory encoding/retrieval). As I understand it, in their main figures the authors ask, "Is there statistically greater influence from the AI to the DMN/FPN in one direction versus another?" And in the AI they show greater "outward" PTE than "inward" PTE from other networks during encoding/retrieval. The balance of directed information favors an outward influence from the AI to DMN/FPN.But in their new analyses, they simply show that the degree of "outward" PTE is greater during task relative to baseline in (almost) all tasks. I believe a more appropriately matched analysis would be to quantify the inward/outward balance during task states, quantify the inward/outward balance during rest states, and then directly statistically compare the two. It could be that the relative balance of directed information flow is nonsignificantly changed between task and rest states, which would be important to know.

We thank the reviewer for this suggestion. We have now run additional analysis by directly comparing the inward/outward balance during the task versus the rest states. To calculate the net inward/outward balance, we calculated the net outflow as the difference between the total outgoing information and total incoming information (PTE(out)–PTE(in)). This analysis revealed that net outflow during task periods is significantly higher compared to rest, during both encoding and recall, and across the four experiments (*ps* < 0.05). These results provide further evidence for enhanced role of AI net directed information flow to the DMN and FPN during memory processing compared to the resting state. These new results have now been included in the revised manuscript (page 12).

Likewise, a similar principle applies to their IFG analysis. They show that the IFG tends to have an "inward" balance of influence from the DMN/FPN (the opposite of the AIs effect), but this does not directly answer whether the AI occupies a statistically unique position in terms of the magnitude of its influence on other regions. More appropriate, as I suggest above, would be to quantify the relative balance inward/outward influence, both for the IFG and the AI, and then directly compare those two quantities. (Given the inversion of the direction of effect, this is likely to be a significant result, but I think it deserves a careful approach regardless.)

We appreciate the reviewer's suggestion. As per the reviewer’s suggestion, we directly compared the net inward/outward balance between the AI and the IFG. Specifically, we compared the net outflow (PTE(out)–PTE(in)) for the AI with the IFG. This analysis revealed that the net outflow for the AI is significantly higher compared to the IFG during both encoding and recall, and across the four experiments. These findings further highlight a key role for the AI in orchestrating large-scale network dynamics during memory processes. The AI's pattern of directed information flow stands in contrast to that of the IFG, despite their anatomical proximity and shared involvement in cognitive control processes. This dissociation underscores the specificity of the AI's function in coordinating network interactions during memory formation and retrieval. These new results have now been included in our revised manuscript (page 11).

(2.2) Consider additional control regions. The authors justify their choice of IFG as a control region very well. In my original comments, I perhaps should have been more clear that the most compelling control analyses here would be to subject every region of the brain outside these networks (with good coverage) to the same analysis, quantify the degree of inward/outward balance, and then see how the magnitude of the AI effect stacks up against all possible other options. If the assertion is that the AI plays a uniquely important role in these memory processes, showing how its influence stacks up against all possible "competitors" would be a very compelling demonstration of their argument.

We thank the reviewer for this suggestion. However, please note that running a large number of random analysis by including a large number of brain regions (every region of the brain outside these networks) and comparing their dynamics to the AI without a hypothesis or solid principle amounts to *p-hacking*, which has been previously strongly criticized by the eLife editors (Makin & Orban de Xivry, 2019). Our study was strongly driven by a solid hypothesis based on prior fMRI studies that have shown that the SN, anchored by the anterior insula (AI), plays a critical role in regulating the engagement and disengagement of the DMN and FPN across diverse cognitive tasks (Bressler & Menon, 2010; Cai et al., 2016; Cai, Ryali, Pasumarthy, Talasila, & Menon, 2021; Chen, Cai, Ryali, Supekar, & Menon, 2016; Kronemer et al., 2022; Raichle et al., 2001; Seeley et al., 2007; Sridharan, Levitin, & Menon, 2008). Moreover, our selection of the IFG as a control region for comparison was also very strongly hypothesis driven, due to its anatomical adjacency to the AI, its involvement in a wide range of cognitive control functions including response inhibition (Cai, Ryali, Chen, Li, & Menon, 2014), and its frequent co-activation with the AI in fMRI studies. Furthermore, the IFG has been associated with controlled retrieval of memory (Badre, Poldrack, Paré-Blagoev, Insler, & Wagner, 2005; Badre & Wagner, 2007; Wagner, Paré-Blagoev, Clark, & Poldrack, 2001), making it a compelling region for comparison. Our findings related to the PTE analysis involving the IFG and also the additional control analyses requested by the reviewer (directly comparing the task-related net balance of the AI with the IFG and also to resting state, please see response to reviewer comment 2.1) strongly highlight a key role of the AI in orchestrating large-scale network dynamics during memory processes.

We believe that our current analyses, including the additional controls, provide a comprehensive and rigorous examination of the AI's role in memory-related network dynamics. Adding analyses of numerous additional regions without clear hypotheses could potentially dilute the focus and interpretability of our results.

However, we acknowledge the importance of considering broader network interactions. In future studies, we could explore the role of other key regions in a hypothesis-driven manner, potentially expanding our understanding of the complex interactions between multiple brain networks during memory processes.

(2.3) Reporting of successful vs. unsuccessful memory results. I apologize if I was not clear in my original comment (2.7, pg. 13 of the response document) regarding successful vs. unsuccessful memory. The fact that no significant difference was found in PTE between successful/unsuccessful memory is a very important finding that adds valuable context to the rest of the manuscript. I believe it deserves a figure, at least in the Supplement, so that readers can visualize the extent of the effect in successful/unsuccessful trials. This is especially important now that the manuscript has been reframed to focus more directly on claims regarding episodic memory processing; if that is indeed the focus, and their central analysis does not show a significant effect conditionalized on the success of memory encoding/retrieval, it is important that readers can see these data directly.

As per the reviewer’s suggestion, we have now included a Figure related to the results for the successful versus unsuccessful comparison in the Supplementary materials of the revised manuscript (Figures S10, S11).

(2.4) Claims regarding causal relationships in the brain. I understand that the authors have defined "causal" in a specific way in the context of their manuscript; I do believe that as a matter of clear and transparent scientific communication, the authors nonetheless bear a responsibility to appreciate how this word may be erroneously interpreted/overinterpreted and I would urge further review of the manuscript to tone down claims of causality. Reflective of this, I was very surprised that even as both reviewers remarked on the need to use the word "causal" with extreme caution, the authors added it to the title in their revised manuscript.

We thank the reviewer for this suggestion. To avoid confusion, we have now removed claims of causality from our manuscript and also changed the title of the manuscript accordingly.

References

Badre, D., Poldrack, R. A., Paré-Blagoev, E. J., Insler, R. Z., & Wagner, A. D. (2005). Dissociable controlled retrieval and generalized selection mechanisms in ventrolateral prefrontal cortex. *Neuron, 47*(6), 907-918. doi:10.1016/j.neuron.2005.07.023

Badre, D., & Wagner, A. D. (2007). Left ventrolateral prefrontal cortex and the cognitive control of memory. *Neuropsychologia, 45*(13), 2883-2901. doi:10.1016/j.neuropsychologia.2007.06.015

Bressler, S. L., & Menon, V. (2010). Large-scale brain networks in cognition: emerging methods and principles. *Trends in Cognitive Sciences, 14*(6), 277-290. doi:10.1016/j.tics.2010.04.004

Cai, W., Chen, T., Ryali, S., Kochalka, J., Li, C. S., & Menon, V. (2016). Causal Interactions Within a Frontal-Cingulate-Parietal Network During Cognitive Control: Convergent Evidence from a Multisite-Multitask Investigation. *Cereb Cortex, 26*(5), 2140-2153. doi:10.1093/cercor/bhv046

Cai, W., Ryali, S., Chen, T., Li, C. S., & Menon, V. (2014). Dissociable roles of right inferior frontal cortex and anterior insula in inhibitory control: evidence from intrinsic and taskrelated functional parcellation, connectivity, and response profile analyses across multiple datasets. *J Neurosci, 34*(44), 14652-14667. doi:10.1523/jneurosci.3048-14.2014

Cai, W., Ryali, S., Pasumarthy, R., Talasila, V., & Menon, V. (2021). Dynamic causal brain circuits during working memory and their functional controllability. *Nat Commun, 12*(1), 3314. doi:10.1038/s41467-021-23509-x

Chen, T., Cai, W., Ryali, S., Supekar, K., & Menon, V. (2016). Distinct Global Brain Dynamics and Spatiotemporal Organization of the Salience Network. *PLOS Biology, 14*(6), e1002469. doi:10.1371/journal.pbio.1002469

Kronemer, S. I., Aksen, M., Ding, J. Z., Ryu, J. H., Xin, Q., Ding, Z., . . . Blumenfeld, H. (2022). Human visual consciousness involves large scale cortical and subcortical networks independent of task report and eye movement activity. *Nat Commun, 13*(1), 7342. doi:10.1038/s41467-022-35117-4

Makin, T. R., & Orban de Xivry, J. J. (2019). Ten common statistical mistakes to watch out for when writing or reviewing a manuscript. Elife, 8. doi:10.7554/eLife.48175

Raichle, M. E., MacLeod, A. M., Snyder, A. Z., Powers, W. J., Gusnard, D. A., & Shulman, G. L. (2001). A default mode of brain function. Proc Natl Acad Sci U S A, 98(2), 676-682. doi:10.1073/pnas.98.2.676

Seeley, W. W., Menon, V., Schatzberg, A. F., Keller, J., Glover, G. H., Kenna, H., . . . Greicius, M. D. (2007). Dissociable Intrinsic Connectivity Networks for Salience Processing and Executive Control. Journal of Neuroscience, 27(9), 2349-2356. doi:10.1523/JNEUROSCI.5587-06.2007

Sridharan, D., Levitin, D. J., & Menon, V. (2008). A critical role for the right fronto-insular cortex in switching between central-executive and default-mode networks. Proceedings of the National Academy of Sciences, 105(34), 12569-12574. doi:10.1073/pnas.0800005105

Wagner, A. D., Paré-Blagoev, E. J., Clark, J., & Poldrack, R. A. (2001). Recovering meaning: left prefrontal cortex guides controlled semantic retrieval. *Neuron, 31*(2), 329-338. doi:10.1016/s0896-6273(01)00359-2